# Discrepancies are Virtue: Weak-to-Strong Generalization through Lens of Intrinsic Dimension

Yijun Dong [1]  Yicheng Li [1]  Yunai Li [2]  Jason D. Lee [3]  Qi Lei [1]

## Abstract

Weak-to-strong (W2S) generalization is a type of finetuning (FT) where a strong (large) student model is trained on pseudo-labels generated by a weak teacher. Surprisingly, W2S FT often outperforms the weak teacher. We seek to understand this phenomenon through the observation that FT often occurs in intrinsically low-dimensional spaces. Leveraging the low intrinsic dimensionality of FT, we analyze W2S in the ridgeless regression setting from a variance reduction perspective. For a strong student-weak teacher pair with sufficiently expressive low-dimensional feature subspaces $\mathcal{V}_s, \mathcal{V}_w$, we provide an exact characterization of the variance that dominates the generalization error of W2S. This unveils a virtue of discrepancy between the strong and weak models in W2S: the variance of the weak teacher is inherited by the strong student in $\mathcal{V}_s \cap \mathcal{V}_w$, while reduced by a factor of $\dim(\mathcal{V}_s)/N$ in the subspace of discrepancy $\mathcal{V}_w \setminus \mathcal{V}_s$ with $N$ pseudo-labels for W2S. Our analysis further casts light on the sample complexities and the scaling of performance gap recovery in W2S. The analysis is supported by experiments on synthetic regression problems, as well as real vision and NLP tasks.

## 1. Introduction

As the capabilities of modern machine learning models grow and exceed human performance in many domains, an emerging problem is whether it would be possible to align the strong superhuman models with weaker supervisors such as human feedback. The weak-to-strong (W2S) framework introduced in (Burns et al., 2024) is a feasible analogy for this problem, inquiring how much capacity of a strong student model can be evoked under the supervision of a weak teacher model. W2S is related to various learning paradigms like co-training (Blum & Mitchell, 1998), self-training (Scudder, 1965), knowledge distillation (Hinton, 2015), and self-distillation (Zhang et al., 2019; 2021), yet being critically dissimilar.

Formalizing the *discrepancy* between the student and the teacher in their *model capacities* is essential for understanding W2S. Most existing theories for W2S treat model capacity as an absolute notion with respect to the downstream task, *e.g.* the weak teacher lacks the robustness to perturbation (Lang et al., 2024; Shin et al., 2024) or the ability to fit the target function (Ildiz et al., 2024; Wu & Sahai, 2024). Nevertheless, empirical observations suggest W2S models also surpass weak models' performance on less challenging tasks (Burns et al., 2024), where the weak teacher has sufficient capacity to achieve good performance. This gap of understanding motivates some natural questions:

*Why W2S happens when both the teacher and student have sufficient capacities for the downstream task?*
*What affects W2S generalization beyond the absolute notion of model capacity?*

To answer the above questions, we analyze W2S generalization through the lens of intrinsic dimension beyond the absolute notion of model capacity. We develop a theoretical framework that incorporates student-teacher correlation, providing a more nuanced explanation of when and why W2S model surpasses the weak teacher's performance.

Our framework is built on two inspiring observations on finetuning (FT): (i) FT tends to fall in the kernel regime (Jacot et al., 2018; Wei et al., 2022; Malladi et al., 2023); and (ii) for a downstream task, relevant features in a stronger pretrained model tend to concentrate in a subspace of lower dimension, known as the *intrinsic dimension*, even when FT is highly overparametrized (Aghajanyan et al., 2021). Leveraging these properties, we cast FT as a ridgeless regression problem over subgaussian features. In particular, we consider two subspaces $\mathcal{V}_s, \mathcal{V}_w \subset \mathbb{R}^d$ of low dimensions $d_s, d_w \ll d$ that encapsulate relevant features in the strong student and weak teacher, respectively. The "absolute" model capacities are measured from two aspects: (i) the intrinsic dimensions $d_s, d_w$ that quantify the representation

[1]New York University [2]Shanghai Jiaotong University [3]Princeton University. Correspondence to: Yijun Dong <yd1319@nyu.edu>, Qi Lei <ql518@nyu.edu>.

*Proceedings of the 42nd International Conference on Machine Learning*, Vancouver, Canada. PMLR 267, 2025. Copyright 2025 by the author(s).

"complexity" and (ii) the approximation errors $\rho_s < \rho_w$ that quantify the representation "accuracy" of the strong and weak models, respectively. In addition, the student-teacher correlation is measured by alignment between the strong and weak feature subspaces through their canonical angles (see Appendix D), $d_{s \wedge w} = \sum \cos(\angle(\mathcal{V}_s, \mathcal{V}_w))$ such that $d_{s \wedge w} \in [0, \min\{d_s, d_w\}]$.

This framework reveals the roles of low intrinsic dimensions and student-teacher correlation in W2S. Decomposing the W2S generalization error into variance and bias, the bias is due to the approximation errors, $\rho_s, \rho_w$, which are low when both student and teacher have sufficient capabilities; whereas the variance comes from noise in the labeled samples for finetuning the weak teacher. When finetuning the strong student with $N \gtrsim d_s$ pseudo-labels generated by a weak teacher supervisedly finetuned with $n \gtrsim d_w$ noisy labels, the variance of W2S is proportional to:

$$\boxed{\frac{d_{s \wedge w}}{n}} \quad + \quad \boxed{\frac{d_s}{N}} \boxed{\frac{d_w - d_{s \wedge w}}{n}} \ .$$

$$\text{Var. in } \mathcal{V}_s \cap \mathcal{V}_w \qquad \text{W2S} \quad \text{Var. in } \mathcal{V}_w \setminus \mathcal{V}_s$$

Specifically, the student mimics variance of the weak teacher in the overlapped feature subspace $\mathcal{V}_s \cap V_w$ but reduces the variance by a factor of $d_s/N$ in the discrepancy between $\mathcal{V}_w$ and $\mathcal{V}_s$. Compared to the weak teacher variance that scales as $d_w/n$, W2S happens (*i.e.* the student outperforms its weak teacher) with sufficient sample sizes $n, N$ when: (i) the strong student has a lower intrinsic dimension, $d_s < d_w$ (as empirically observed in (Aghajanyan et al., 2021) on NLP tasks), or (ii) the student-teacher correlation is low, $d_{s \wedge w} < d_w$. This unveils the benefit of discrepancy between the teacher and student features for W2S:

*In the variance-dominated regime, W2S comes from variance reduction in the discrepancy of weak teacher features from strong student features.*

To provide intuitions for such variance reduction, let's consider $\mathcal{V}_s$ and $\mathcal{V}_w$ with large discrepancy as two distinct aspects of a downstream task that both provide sufficient information. For example, to classify the brand of a car in an image, one can use either the simple information in the logo (strong features $\mathcal{V}_s$ with a lower intrinsic dimension $d_s$) or the complex information in the design (weak features $\mathcal{V}_w$ with a higher intrinsic dimension $d_w$). In a high-dimensional feature space, $\mathcal{V}_s$ and $\mathcal{V}_w$ that encode irrelevant information are likely almost orthogonal, leading to a small $d_{s \wedge w}$. Then, the error of weak teacher induced by noise in the $n$ labeled samples is only correlated to the design features in $\mathcal{V}_w$ but almost independent of the logo features in $\mathcal{V}_s$. Therefore, the error in weak supervision can be viewed as independent label noise for the student. With an intrinsic dimension of $d_s$, the generalization error of student induced by such independent noise vanishes at a rate of $O(d_s/N)$.

Our main contributions are summarized as follows:

- We introduce a theoretical framework for W2S based on the low intrinsic dimensions of FT, where we characterize model capacities from three aspects: approximation errors for "accuracy", intrinsic dimensions for "complexity", and student-teacher correlation for "alignment" (Section 2).

- We provide a generalization analysis for W2S with an exact characterization of the variance under a Gaussian feature assumption, unveiling the virtue of discrepancy between the student and teacher in W2S (Section 3.1).

- We investigate the relative W2S performance in terms of performance gap recovery (PGR) (Burns et al., 2024) and outperforming ratio (OPR) compared to the strong baseline model supervisedly finetuned with $n$ labels. A case study provides insights into the scaling of PGR and OPR with respect to the sample sizes $n, N$ and sample complexities in W2S (Section 3.2).

### 1.1. Related works

In this section, we review literature directly related to W2S and intrinsic dimension, while deferring detailed discussions on other related topics to Appendix A.

**W2S alignment: emergence & growing influence.** W2S generalization was first introduced by (Burns et al., 2024), offering a promising avenue for aligning superhuman models. A rapidly expanding body of work has empirically validated this phenomenon across diverse tasks in vision and language models since then. Guo et al. (2024); Liu & Alahi (2024) propose loss functions and multi-teacher algorithms. Guo & Yang (2024); Yang et al. (2024b) refine training data to improve W2S alignment, while Li et al. (2024); Sun et al. (2024) use weak models for data filtering and reranking. In contrast, Yang et al. (2024a) highlight the issue of W2S deception, where strong models superficially align with weak teachers but fail in new or conflicting cases. This calls for theoretical understanding of the mechanism behind W2S generalization and better strategies to mitigate misalignment. Coinciding with our theoretical findings, the strong negative correlation between model similarity and W2S performance is empirically observed in the concurrent work (Goel et al., 2025) in extensive experiments.

**Theoretical perspectives on W2S generalization.** Existing theories on W2S interpret the difference between strong and weak models in terms of the quality of their representations (from the bias perspective in our context). Lang et al. (2024) study W2S in classification through the lens of neighborhood expansion (Wei et al., 2020; Cai et al., 2021) where model capacity is interpreted as the robustness to perturbation. Within this framework, Shin et al. (2024) highlights the importance of data selection in W2S while

proposing metrics and algorithms for data selection in W2S. In the same classification setting, Somerstep et al. (2024) takes a transfer learning perspective and highlights the limitation of naive FT in W2S. Wu & Sahai (2024) take a benign overfitting (Bartlett et al., 2020; Muthukumar et al., 2021) perspective and show the asymptotic transition between W2S generalization and random guessing. For regression tasks, Charikar et al. (2024) reveals the connection between W2S gain and misfit error of the strong student on weak pseudo-labels. Ildiz et al. (2024) treats W2S as a special case of knowledge distillation, showing its limitation in terms of improving the data scaling law (Spigler et al., 2020; Bahri et al., 2024). We consider a similar ridgeless regression setting as Ildiz et al. (2024) but from a fundamentally different aspect – variance reduction. This offers a fresh take on the roles of intrinsic dimension and student-teacher correlation in W2S. Parallel to this work, Medvedev et al. (2025); Yao et al. (2025) analyze W2S in the context of early stopping and loss functions, respectively.

**Intrinsic dimension.** There has been prevailing empirical and theoretical evidence that natural high-dimensional systems often exhibit low-dimensional structures (Udell & Townsend, 2019). The concept of intrinsic dimension has been widely studied in manifold learning (Tenenbaum et al., 2000), dimensionality reduction (Van der Maaten & Hinton, 2008), and representation learning (Bengio et al., 2013). In the context of neural network training, Li et al. (2018) propose a method to measure the intrinsic dimension of the objective landscape based on the Johnson-Lindenstrauss-type transforms (Johnson, 1984). This offers a structural perspective on task complexity, which is largely absent from prior W2S studies. Aghajanyan et al. (2021) investigate the intrinsic dimensions of FT, showing that FT over large models usually has surprisingly low intrinsic dimensions, while good pretraining tends to reduce the intrinsic dimension. Our work extends these insights by linking the intrinsic dimension to W2S, decomposing generalization error into bias and variance, and building upon findings from Yang et al. (2020); Amari et al. (2020) on variance-dominated risks in learning from noisy labels.

### 1.2. Notations
Given any $n \in \mathbb{Z}_+$, we denote $[n] = \{1, \cdots, n\}$. Let $\mathbf{e}_n$ be the $n$-th canonical basis of conformable dimension; $\mathbf{I}_n$ is the $n \times n$ identity matrix; and $\mathbf{0}_n, \mathbf{1}_n \in \mathbb{R}^n$ are vectors with all zeroes and ones. For any distribution $p$ and $n \in \mathbb{Z}_+$, let $p^n \triangleq \bigotimes_{i=1}^n p$ as the $n$-fold product distribution of $p$, sampling which yields $n$ *i.i.d.* samples from $p$. For any matrix $\mathbf{A} \in \mathbb{R}^{n \times d}$, let $\mathbf{A}^\dagger$ be the Moore-Penrose pseudoinverse. We adapt the standard asymptotic notations: for any functions $f, g : \mathbb{R}_+ \to \mathbb{R}_+$, we write $f = O(g)$ or $f \lesssim g$ if there exists some constant $C > 0$ such that $f(x) \leqslant Cg(x)$ for all $x \in \mathbb{R}_+$; $f = \Omega(g)$ or $f \gtrsim g$ if $g = O(f)$; $f \asymp g$

if $f = O(g)$ and $f = \Omega(g)$. Also, we denote $f = o(g)$ or $f/g = o_x(1)$ if $\lim_{x \to \infty} f(x)/g(x) = 0$.

## 2. Problem setup
In this section, we cast FT as a ridgeless regression problem. The setup is introduced in three parts: model capacity, FT algorithms, and metrics for W2S performance.

Consider the problem of learning an unknown data distribution $\mathcal{D}(f_*) : \mathcal{X} \times \mathcal{Y} \to [0, 1]$ (where $\mathcal{X}$ is a set and $\mathcal{Y} \subseteq \mathbb{R}$) associated with a downstream task characterized by an unknown ground truth function $f_* : \mathcal{X} \to \mathbb{R}$. Every sample $(\mathbf{x}, y) \sim \mathcal{D}(f_*)$ satisfies $y = f_*(\mathbf{x}) + z$ where $z \sim \mathcal{N}(0, \sigma^2)$ is an independent Gaussian label noise. Let $\mathcal{D} : \mathcal{X} \to [0, 1]$ be the marginal distribution over $\mathcal{X}$. We assume that $f_*$ is bounded: $|f_*(\mathbf{x})| \leqslant 1$ for $\mathbf{x} \sim \mathcal{D}$ almost surely (under normalization without loss of generality).

### 2.1. Measures for model capacity
Model capacity is a key notion in W2S that distinguishes the weak and strong models. Intuitively, a stronger model is capable of representing a downstream task $\mathcal{D}(f_*)$ more accurately and efficiently. We formalize such "accuracy" and "complexity" through the notions of intrinsic dimensions and FT approximation errors, as introduced below.

Consider two pretrained models, a weak model $\phi_w$ and a strong model $\phi_s$, that output features $\mathcal{X} \to \mathbb{R}^d$:

**Assumption 2.1** (Sub-gaussian features). For $\mathbf{x} \sim \mathcal{D}$, assume both $\phi_w(\mathbf{x})$ and $\phi_s(\mathbf{x})$ are zero-mean sub-gaussian random vectors with $\mathbb{E}[\phi_w(\mathbf{x})] = \mathbb{E}[\phi_s(\mathbf{x})] = \mathbf{0}_d$, and $\mathbb{E}[\phi_w(\mathbf{x})\phi_w(\mathbf{x})^\top] = \mathbf{\Sigma}_w, \mathbb{E}[\phi_s(\mathbf{x})\phi_s(\mathbf{x})^\top] = \mathbf{\Sigma}_s$.

Approximation errors measure the model capacity from the "accuracy" perspective: how accurately can the downstream task $\mathcal{D}(f_*)$ be represented by the pretrained features of $\phi_s$ and $\phi_w$ over the population.

**Definition 2.2** (FT approximation error). Given $\mathcal{D}(f_*)$, let the FT approximation errors of $\phi_s$ and $\phi_w$ be

$$\rho_s = \min_{\boldsymbol{\theta} \in \mathbb{R}^d} \mathbb{E}_{\mathbf{x} \sim \mathcal{D}} \left[ (\phi_s(\mathbf{x})^\top \boldsymbol{\theta} - f_*(\mathbf{x}))^2 \right],$$
$$\rho_w = \min_{\boldsymbol{\theta} \in \mathbb{R}^d} \mathbb{E}_{\mathbf{x} \sim \mathcal{D}} \left[ (\phi_w(\mathbf{x})^\top \boldsymbol{\theta} - f_*(\mathbf{x}))^2 \right],$$

such that $\rho_s, \rho_w \in [0, 1]$ (given $\Pr_{\mathbf{x} \sim \mathcal{D}}[|f_*(\mathbf{x})| \leqslant 1] = 1$ by assumption). We assume both $\rho_s$ and $\rho_w$ are small compared to label noise: $\rho_s + \rho_w \ll \sigma^2$; while the stronger model $\phi_s$ has a lower FT approximation error: $\rho_s < \rho_w$.

Notice that FT approximation error is different from approximation error of the full model. Precisely, Definition 2.2 quantifies the approximation error of *finetuning the pretrained model*, whose dynamics (Wei et al., 2022; Malladi et al., 2023) fall in the kernel regime (Jacot et al., 2018).

Since feature learning is limited in the kernel regime (Woodworth et al., 2020), a low FT approximation error requires the pretrained features $\phi_s$ and $\phi_w$ to provide an expressive set of features for the downstream task $\mathcal{D}(f_*)$.

In addition to "accuracy", a strong model ought to be able to represent a downstream task concisely. We quantify such "complexity" through intrinsic dimension – the minimum dimension of a feature subspace that can represent the downstream task $\mathcal{D}(f_*)$ accurately. In light of the ubiquitous observations on low intrinsic dimensions of FT (Aghajanyan et al., 2021), we introduce a common assumption for FT (Xia et al., 2024; Dong et al., 2024c) that the pretrained features of $\phi_s, \phi_w$ are concentrated in low-dimensional subspaces, as formalized below.

**Definition 2.3** (Intrinsic dimensions). Let $d_s = \mathrm{rank}(\boldsymbol{\Sigma}_s)$ and $d_w = \mathrm{rank}(\boldsymbol{\Sigma}_w)$ be the **intrinsic dimensions** of $\phi_s$ and $\phi_w$. Assume low intrinsic dimensions[1]: $d_s, d_w \ll d$.

Moreover, Aghajanyan et al. (2021) observed that the stronger pretrained models tend to have lower intrinsic dimensions, *i.e.* we often have $d_s < d_w$ in practice.

Beyond the absolute notion of model capacity in terms of intrinsic dimensions and FT approximation errors, we introduce a relative measure for the similarity between weak and strong models – the correlation dimension characterizing the overlap between feature subspaces of $\phi_s$ and $\phi_w$.

**Definition 2.4** (Correlation dimension). Consider spectral decompositions $\boldsymbol{\Sigma}_s = \mathbf{V}_s \boldsymbol{\Lambda}_s \mathbf{V}_s^\top$ and $\boldsymbol{\Sigma}_w = \mathbf{V}_w \boldsymbol{\Lambda}_w \mathbf{V}_w^\top$, where $\boldsymbol{\Lambda}_s \in \mathbb{R}^{d_s \times d_s}$ and $\boldsymbol{\Lambda}_w \in \mathbb{R}^{d_w \times d_w}$ are diagonal matrices with positive eigenvalues in decreasing order; while $\mathbf{V}_s \in \mathbb{R}^{d \times d_s}$ and $\mathbf{V}_w \in \mathbb{R}^{d \times d_w}$ consist of the corresponding orthonormal eigenvectors. Let $d_{s \wedge w} = \left\| \mathbf{V}_s^\top \mathbf{V}_w \right\|_F^2$ be the **correlation dimension** between $\phi_s$ and $\phi_w$ such that $0 \leqslant d_{s \wedge w} \leqslant \min\{d_s, d_w\}$.

*Remark* 2.5 (Extension to general FT). While we focus on learning $\mathcal{D}(f_*)$ via linear probing over $\phi_w$ and $\phi_s$, since the finetuning dynamics fall approximately in the kernel regime (Wei et al., 2022; Malladi et al., 2023), the linear probing analysis naturally extends to general FT. Precisely, let $f_w(\cdot|\mathbf{0}_d) : \mathcal{X} \to \mathbb{R}$ and $f_s(\cdot|\mathbf{0}_d) : \mathcal{X} \to \mathbb{R}$ be the pretrained weak and strong models, where $d$ is the number of finetunable parameters. By denoting $\phi_w(\mathbf{x}) = \nabla_{\boldsymbol{\theta}} f_w(\mathbf{x}|\mathbf{0}_d)$ and $\phi_s(\mathbf{x}) = \nabla_{\boldsymbol{\theta}} f_s(\mathbf{x}|\mathbf{0}_d)$, the general FT process effectively reduces to linear probing over $\phi_w$ and $\phi_s$.

---

[1] In practice, $\boldsymbol{\Sigma}_s$ and $\boldsymbol{\Sigma}_w$ usually admit fast-decaying eigenvalues, but not exactly low-rank. In this more realistic case, ridge regression with suitable choices of regularization hyperparameters intuitively performs "soft" truncation of the small singular values, effectively leading to low intrinsic dimensions $d_s, d_w \ll d$. For conciseness of the main message, we focus on the ideal case of exactly low-rank $\boldsymbol{\Sigma}_s$ and $\boldsymbol{\Sigma}_w$ in the main text, while deferring the ridge regression analysis for general $\boldsymbol{\Sigma}_s$ and $\boldsymbol{\Sigma}_w$ to Appendix C.

## 2.2. W2S and supervised finetuning

With the task $\mathcal{D}(f_*)$ and models $\phi_s, \phi_w$ specified, we are ready to formalize the data and algorithms for FT.

We consider two sample sets drawn *i.i.d.* from $\mathcal{D}(f_*)$: a small labeled set $\widetilde{\mathcal{S}} = \{(\widetilde{\mathbf{x}}_i, \widetilde{y}_i) | i \in [n]\} \sim \mathcal{D}(f_*)^n$ and a large sample set $\mathcal{S} = \{(\mathbf{x}_i, y_i) | i \in [N]\} \sim \mathcal{D}(f_*)^N$ where the labels $y_i$ are inaccessible, denoting the unlabeled part as $\mathcal{S}_x = \{\mathbf{x}_i | i \in [N]\}$. The goal is to learn a function $f : \mathcal{X} \to \mathbb{R}$ using $\widetilde{\mathcal{S}}$ and $\mathcal{S}_x$ that generalizes well to $\mathcal{D}(f_*)$.

For $\widetilde{\mathcal{S}}$, let $\widetilde{\boldsymbol{\Phi}}_w = [\phi_w(\widetilde{\mathbf{x}}_1), ..., \phi_w(\widetilde{\mathbf{x}}_n)]^\top$, $\widetilde{\boldsymbol{\Phi}}_s = [\phi_s(\widetilde{\mathbf{x}}_1), ..., \phi_s(\widetilde{\mathbf{x}}_n)]^\top \in \mathbb{R}^{n \times d}$ be the weak and strong features with associated labels $\widetilde{\mathbf{y}} = [\widetilde{y}_1, \ldots, \widetilde{y}_n]^\top \in \mathbb{R}^n$. Analogously for $\mathcal{S}$, let $\boldsymbol{\Phi}_w = [\phi_w(\mathbf{x}_1), \ldots, \phi_w(\mathbf{x}_N)]^\top$, $\boldsymbol{\Phi}_s = [\phi_s(\mathbf{x}_1), \ldots, \phi_s(\mathbf{x}_N)]^\top \in \mathbb{R}^{N \times d}$ be the weak and strong features with unknown labels $\mathbf{y} = [y_1, \ldots, y_N]^\top \in \mathbb{R}^N$. For conciseness of notations, we introduce a mild regularity assumption on the ranks of these feature matrices.

**Assumption 2.6** (Sufficient finetuning data). Assume $\widetilde{\mathcal{S}}$ and $\mathcal{S}$ are sufficiently large such that $\mathrm{rank}(\widetilde{\boldsymbol{\Phi}}_w) = \mathrm{rank}(\boldsymbol{\Phi}_w) = d_w$ and $\mathrm{rank}(\widetilde{\boldsymbol{\Phi}}_s) = \mathrm{rank}(\boldsymbol{\Phi}_s) = d_s$ almost surely[2].

Given regularization hyperparameters $\alpha_w, \alpha_{\mathrm{w2s}}, \alpha_s, \alpha_c > 0$, we consider the following FT algorithms:

(a) **Weak teacher model** $f_w(\mathbf{x}) = \phi_w(\mathbf{x})^\top \boldsymbol{\theta}_w$ is supervisedly finetuned over $\widetilde{\mathcal{S}}$:

$$\boldsymbol{\theta}_w = \operatorname*{argmin}_{\boldsymbol{\theta} \in \mathbb{R}^d} \frac{1}{n} \left\| \widetilde{\boldsymbol{\Phi}}_w \boldsymbol{\theta} - \widetilde{\mathbf{y}} \right\|_2^2 + \alpha_w \left\| \boldsymbol{\theta} \right\|_2^2. \quad (1)$$

(b) **W2S model** $f_{\mathrm{w2s}}(\mathbf{x}) = \phi_s(\mathbf{x})^\top \boldsymbol{\theta}_{\mathrm{w2s}}$ is finetuned over the strong feature $\phi_s$ through $\mathcal{S}_x$ and their pseudo-labels generated by the weak teacher model:

$$\boldsymbol{\theta}_{\mathrm{w2s}} = \operatorname*{argmin}_{\boldsymbol{\theta} \in \mathbb{R}^d} \frac{1}{N} \left\| \boldsymbol{\Phi}_s \boldsymbol{\theta} - \boldsymbol{\Phi}_w \boldsymbol{\theta}_w \right\|_2^2 + \alpha_{\mathrm{w2s}} \left\| \boldsymbol{\theta} \right\|_2^2$$

$$(2)$$

(c) **Strong SFT model** $f_s(\mathbf{x}) = \phi_s(\mathbf{x})^\top \boldsymbol{\theta}_s$ is a strong baseline where the strong feature $\phi_s$ is supervisedly finetuned over the small labeled set $\widetilde{\mathcal{S}}$ directly:

$$\boldsymbol{\theta}_s = \operatorname*{argmin}_{\boldsymbol{\theta} \in \mathbb{R}^d} \frac{1}{n} \left\| \widetilde{\boldsymbol{\Phi}}_s \boldsymbol{\theta} - \widetilde{\mathbf{y}} \right\|_2^2 + \alpha_s \left\| \boldsymbol{\theta} \right\|_2^2. \quad (3)$$

(d) **Strong ceiling model** $f_c(\mathbf{x}) = \phi_s(\mathbf{x})^\top \boldsymbol{\theta}_c$ is a reference for the ceiling performance where $\phi_s$ is supervisedly finetuned over $\mathcal{S} \cup \widetilde{\mathcal{S}}$, assuming access to the

---

[2] Assuming distributions of $\phi_w(\mathbf{x})$ and $\phi_s(\mathbf{x})$ are absolutely continuous with respect to the Lebesgue measure, for any $n \geqslant d_w$ and $n \geqslant d_s$, $\mathrm{rank}(\widetilde{\boldsymbol{\Phi}}_w) = \mathrm{rank}(\boldsymbol{\Phi}_w) = d_w$ and $\mathrm{rank}(\widetilde{\boldsymbol{\Phi}}_s) = \mathrm{rank}(\boldsymbol{\Phi}_s) = d_s$ almost surely (Vershynin, 2018, §3.3.1).

unknown labels $\mathbf{y} = [y_1, \ldots, y_N]^\top$:

$$\boldsymbol{\theta}_c = \underset{\boldsymbol{\theta} \in \mathbb{R}^d}{\operatorname{argmin}} \frac{1}{n+N} \left\| \begin{bmatrix} \widetilde{\boldsymbol{\Phi}}_s \\ \boldsymbol{\Phi}_s \end{bmatrix} \boldsymbol{\theta} - \begin{bmatrix} \widetilde{\mathbf{y}} \\ \mathbf{y} \end{bmatrix} \right\|_2^2 + \alpha_c \|\boldsymbol{\theta}\|_2^2. \quad (4)$$

For any $f$ with randomness from its training samples $\mathcal{S}_f \sim \mathcal{D}(f_*)^{|\mathcal{S}_f|}$, let $\mathbf{X}_f$ be the unlabeled part of $\mathcal{S}_f$, and let $\mathcal{S}_t \sim \mathcal{D}(f_*)^{|\mathcal{S}_t|}$ be a test set. We measure the generalization error via the expected excess risk of $f$ over $\mathcal{S}_f$ and $\mathcal{S}_t$:

$$\mathbf{ER}(f) = \mathbb{E}_{\mathcal{S}_f, \mathcal{S}_t} \left[ \frac{1}{|\mathcal{S}_t|} \sum_{\mathbf{x} \in \mathcal{S}_t} (f(\mathbf{x}) - f_*(\mathbf{x}))^2 \right].$$

Notice that $\mathbf{ER}(f) = \mathbf{Var}(f) + \mathbf{Bias}(f)$ can be decomposed into variance and bias, where

$$\mathbf{Var}(f) = \mathbb{E}_{\mathcal{S}_f, \mathcal{S}_t} \left[ \frac{1}{|\mathcal{S}_t|} \sum_{\mathbf{x} \in \mathcal{S}_t} (f(\mathbf{x}) - \mathbb{E}_{\mathcal{S}_f | \mathbf{X}_f} [f(\mathbf{x})])^2 \right]$$

$$\mathbf{Bias}(f) = \mathbb{E}_{\mathbf{X}_f, \mathcal{S}_t} \left[ \frac{1}{|\mathcal{S}_t|} \sum_{\mathbf{x} \in \mathcal{S}_t} (\mathbb{E}_{\mathcal{S}_f | \mathbf{X}_f} [f(\mathbf{x})] - f_*(\mathbf{x}))^2 \right]$$

For clarity of the main message, we set $\mathcal{S}_t = \mathcal{S}$ for $f_w$ in (1), $f_{\text{w2s}}$ in (2), $f_s$ in (3) for fair comparison, and $\mathcal{S}_t = \widetilde{\mathcal{S}} \cup \mathcal{S}$ for $f_c$ in (4) for simplicity[3].

*Remark* 2.7 (Regularization prevents W2S from overfitting). As pointed out in (Burns et al., 2024), suitable regularization is crucial to prevent W2S from overfitting the weak teacher. For overparametrized problems[4], even without explicit regularization, gradient descent implicitly biases toward the minimum $\ell_2$-norm solutions in the kernel regime (Woodworth et al., 2020), equivalent to solving Equations (1) to (4) with $\alpha_w, \alpha_{\text{w2s}}, \alpha_s, \alpha_c \to 0$. Therefore, we focus on ridgeless regression here under the idealized intrinsic dimension assumption in Definition 2.3. In Appendix C, we extend our analysis to the more general scenario: when $\boldsymbol{\Sigma}_s, \boldsymbol{\Sigma}_w$ are not exactly low-rank, a careful choice of $\alpha_w, \alpha_{\text{w2s}} > 0$ brings a W2S generalization bound, Theorem C.4, that conveys the same message as Theorem 3.1 in the ridgeless case.

### 2.3. Metrics for W2S performance

In addition to the absolute generalization error of W2S, $\mathbf{ER}(f_{\text{w2s}})$, we quantify the W2S performance of $f_{\text{w2s}}$ relative to $f_w$, $f_s$, and $f_c$ through the following metrics:

(a) **Performance gap recovery (PGR)** introduced in (Burns et al., 2024) measures the ratio between excess risk reductions from the weak teacher $f_w$ of the

---

W2S model $f_{\text{w2s}}$ and the strong ceiling model $f_c$:

$$\mathbf{PGR} = \frac{\mathbf{ER}(f_w) - \mathbf{ER}(f_{\text{w2s}})}{\mathbf{ER}(f_w) - \mathbf{ER}(f_c)}. \quad (5)$$

In practice, $\mathbf{ER}(f_{\text{w2s}})$ typically falls between $\mathbf{ER}(f_c)$ and $\mathbf{ER}(f_w)$ (Burns et al., 2024). Therefore, it usually holds that $0 \leqslant \mathbf{PGR} \leqslant 1$. A higher $\mathbf{PGR}$ indicates better W2S generalization: the W2S model $f_{\text{w2s}}$ can recover more of the excess risk gap between the weak teacher $f_w$ and the strong ceiling model $f_c$.

(b) **Outperforming ratio (OPR)** compares excess risks of the strong baseline $f_s$ and the W2S model $f_{\text{w2s}}$:

$$\mathbf{OPR} = \mathbf{ER}(f_s)/\mathbf{ER}(f_{\text{w2s}}). \quad (6)$$

A higher $\mathbf{OPR}$ implies better W2S generalization: $f_{\text{w2s}}$ outperforms $f_s$ when $\mathbf{OPR} > 1$. This metric could be of interest in practice when the labeled samples $\widetilde{\mathcal{S}}$ are limited – if $\mathbf{OPR} < 1$, SFT the strong model over $\widetilde{\mathcal{S}}$ would be a better choice than W2S both in terms of generalization and computational efficiency.

## 3. Main results

In this section, we first analyze the generalization errors of W2S and its reference models in Section 3.1. Then in Section 3.2, we conduct a case study on the W2S performance in terms of the metrics introduced in Section 2.3.

### 3.1. Generalization errors

We start with the W2S model $f_{\text{w2s}}(\mathbf{x}) = \phi_s(\mathbf{x})^\top \boldsymbol{\theta}_{\text{w2s}}$ fine-tuned as in (1), (2) with both $\alpha_w, \alpha_{\text{w2s}} \to 0$. For demonstration purposes, we consider an idealized Gaussian feature case in the main text, where the variance of $f_{\text{w2s}}$ can be exactly characterized (instead of upper bounded)[5].

**Theorem 3.1** (W2S model (formally in B.1)). *Assuming Assumptions 2.1 and 2.6 and $\phi_w(\mathbf{x}) \sim \mathcal{N}(\boldsymbol{0}_d, \boldsymbol{\Sigma}_w)$, for $n > d_w + 1$, $\mathbf{ER}(f_{\text{w2s}}) = \mathbf{Var}(f_{\text{w2s}}) + \mathbf{Bias}(f_{\text{w2s}})$ satisfies*

$$\mathbf{Var}(f_{\text{w2s}}) = \frac{\sigma^2}{n - d_w - 1} \left( d_{s \wedge w} + \frac{d_s}{N} (d_w - d_{s \wedge w}) \right),$$

$$\mathbf{Bias}(f_{\text{w2s}}) \leqslant \mathbf{Bias}(f_w) + \rho_s \leqslant O(\rho_w) + \rho_s.$$

*where the inequality for $\mathbf{Bias}(f_{\text{w2s}})$ is strict if $\mathbf{Bias}(f_w) > 0$ and $d_s < d_w$.*

*Remark* 3.2 (Discrepancy is virtue). Notice that $\mathbf{Var}(f_{\text{w2s}})$ consists of two terms. (a) In the overlapped subspace $\text{Range}(\boldsymbol{\Sigma}_s) \cap \text{Range}(\boldsymbol{\Sigma}_w)$ with correlation dimension $d_{s \wedge w}$, the variance $\sigma^2 d_{s \wedge w}/(n - d_w - 1)$ mimics that of the weak teacher, where more pseudo-labels $N$ fail to reduce the variance. (b) Whereas variance in the subspace

---

[3]The strong ceiling performance $\mathbf{ER}(f_c)$ only serves as a reference in (5), irrelevant of the rest of the analysis.

[4]While the feature dimension $d$ can be either larger (overparametrized) or smaller (underparametrized) than the sample sizes $n, N, n + N$, with the low intrinsic dimensions $d_s, d_w \ll d$, Equations (1) to (4) are always underdetermined.

[5] The analogous generalization bound holds up to constants for sub-gaussian features in Assumption 2.1, see Theorem B.2.

of discrepancy $\mathrm{Range}(\boldsymbol{\Sigma}_w) \setminus \mathrm{Range}(\boldsymbol{\Sigma}_s)$ takes the form $\sigma^2 (d_s/N)(d_w - d_{s \wedge w})/(n - d_w - 1)$, reduced by a factor of $d_s/N$ and vanishing as $N$ grows.

As a reference, we also look into the weak teacher model $f_w(\mathbf{x}) = \phi_w(\mathbf{x})^\top \boldsymbol{\theta}_w$ in (1) with $\alpha_w \to 0$:

**Proposition 3.3** (Weak teacher (B.2)). *Under the setting of Theorem 3.1, $\mathbf{ER}(f_w) = \mathbf{Var}(f_w) + \mathbf{Bias}(f_w)$ satisfies*

$$\mathbf{Var}(f_w) = \frac{\sigma^2 d_w}{n - d_w - 1}, \quad \mathbf{Bias}(f_w) \lesssim \rho_w,$$

*when $\phi_w(\mathbf{x}) \sim \mathcal{N}(\boldsymbol{0}_d, \boldsymbol{\Sigma}_w)$; $\mathbf{Var}(f_w) \lesssim \frac{\sigma^2 d_w}{n}$ otherwise.*

To measure the W2S performance in a relative sense, another two necessary references are the strong SFT baseline $f_s(\mathbf{x}) = \phi_s(\mathbf{x})^\top \boldsymbol{\theta}_s$ in (3) and strong ceiling model $f_c(\mathbf{x}) = \phi_s(\mathbf{x})^\top \boldsymbol{\theta}_c$ in (4), with both $\alpha_s, \alpha_c \to 0$:

**Corollary 3.4** (Strong SFT and ceiling). *Under the setting of Theorem 3.1, $\mathbf{ER}(f_s) = \mathbf{Var}(f_s) + \mathbf{Bias}(f_s)$ satisfies*

$$\mathbf{Var}(f_s) = \frac{\sigma^2 d_s}{n - d_s - 1}, \quad \mathbf{Bias}(f_s) \lesssim \rho_s,$$

*when $\phi_s(\mathbf{x}) \sim \mathcal{N}(\boldsymbol{0}_d, \boldsymbol{\Sigma}_s)$; and $\mathbf{Var}(f_s) \lesssim \frac{\sigma^2 d_s}{n}$ otherwise. $\mathbf{ER}(f_c) = \mathbf{Var}(f_c) + \mathbf{Bias}(f_c)$ satisfies*

$$\mathbf{Var}(f_c) = \frac{\sigma^2 d_s}{N + n}, \quad \mathbf{Bias}(f_c) \leqslant \rho_s.$$

**W2S in variance.** Assuming $\rho_s + \rho_w \ll \sigma^2$ (Definition 2.2), variance dominates the generalization error. Theorem 3.1 and Proposition 3.3 suggest that W2S generalization generally occurs in variance, *i.e.* $\mathbf{Var}(f_{\mathrm{w2s}}) < \mathbf{Var}(f_w)$, as long as the W2S FT sample size is reasonably large, $N > d_s$. Meanwhile, with a low correlation dimension $d_{s \wedge w}$, W2S in variance is more pronounced, especially when $N$ is much larger than $d_s$.

**W2S in bias.** When the strong student has zero FT approximation error, $\rho_s = 0$, as long as $\mathbf{Bias}(f_w) > 0$, and the strong student has a lower intrinsic dimension, $d_s < d_w$, Theorem 3.1 and Proposition 3.3 further suggest that W2S also enjoys a strictly lower bias than the weak teacher: $\mathbf{Bias}(f_{\mathrm{w2s}}) < \mathbf{Bias}(f_w)$[6].

### 3.2. W2S performance: a case study

With the generalization analysis, we are ready to take a closer look at the W2S performance in terms of **PGR** and **OPR** defined in Section 2.3.

---

[6] Quantifying the advantage of W2S in bias requires further assumptions on the downstream task $\mathcal{D}(f_*)$ and the covariance matrices $\boldsymbol{\Sigma}_w, \boldsymbol{\Sigma}_s$, analogous to the settings in (Ildiz et al., 2024; Wu & Sahai, 2024), which is deviating from our focus on variance but could be an interesting future direction.

**Proposition 3.5** (**PGR** and **OPR** lower bounds (B.3)). *Given $f_w, f_{\mathrm{w2s}}, f_c$, and $f_s$ as in Theorem 3.1, Proposition 3.3, and Corollary 3.4, under Assumptions 2.1 and 2.6, assuming $\phi_w(\mathbf{x}) \sim \mathcal{N}(\boldsymbol{0}_d, \boldsymbol{\Sigma}_w)$ and $\phi_s(\mathbf{x}) \sim \mathcal{N}(\boldsymbol{0}_d, \boldsymbol{\Sigma}_s)$[5], with $n = d_w + q + 1$ for some constant $q \in \mathbb{N}$, we have*

$$\mathbf{PGR} \geqslant 1 - \frac{d_{s \wedge w}}{d_w} - \frac{d_s}{N} \frac{d_w - d_{s \wedge w}}{d_w} - \frac{q}{d_w} \frac{O(\rho_w) + \rho_s}{\sigma^2},$$

*and* $\mathbf{OPR} \geqslant$

$$\left( \frac{n}{q} \frac{d_{s \wedge w} + (d_w - d_{s \wedge w})d_s/N}{d_s} + \frac{n}{d_s} \frac{O(\rho_w) + \rho_s}{\sigma^2} \right)^{-1}.$$

We recall from Section 2.3 that the larger **PGR** and **OPR** imply better W2S generalization. Then, a natural question hinted by Proposition 3.5 is *how do the sample sizes $n, N$ affect the W2S performance?* The concrete answers to this question depend on the relative magnitude of the FT approximation errors and label noise, $(\rho_w + \rho_s)/\sigma^2$.

**Case I: negligible FT approximation error.** In the ideal case where the FT approximation errors are negligible compared to label noise, $(\rho_w + \rho_s)/\sigma^2 \to 0$, Proposition 3.5 suggests better lower bounds for **PGR**, **OPR** as $n, N$ increase:

$$\mathbf{PGR} \geqslant 1 - \frac{d_{s \wedge w} + (d_w - d_{s \wedge w})d_s/N}{d_w},$$

$$\mathbf{OPR} \geqslant \frac{n - d_w - 1}{n} \frac{d_s}{d_{s \wedge w} + (d_w - d_{s \wedge w})d_s/N}.$$

Depending on $d_{s \wedge w}$, we have the following cases:

(a) When $d_{s \wedge w} > 0$, with sample sizes $n \gtrsim d_w$ and $N \gtrsim (d_w/d_{s \wedge w} - 1)d_s$, $\mathbf{PGR} \geqslant 1 - O(d_{s \wedge w}/d_w)$ and $\mathbf{OPR} \geqslant \Omega(d_s/d_{s \wedge w})$ imply good W2S performance if $d_{s \wedge w} \ll \min\{d_s, d_w\}$.

(b) When $d_{s \wedge w} = 0$, a labeled sample size of $n \gtrsim d_w$ leads to $\mathbf{PGR} \geqslant 1 - O(d_s/N)$ and $\mathbf{OPR} \geqslant \Omega(N/d_w)$, implying good W2S performance when $N \gg \max\{d_w, d_s\}$.

**Case II: small non-negligible FT approximation error.** In a more realistic scenario where $0 < (\rho_s + \rho_w)/\sigma^2 \ll 1$ is small but non-negligible, the trade-off between variance and bias brings about a non-monotonic scaling of the **PGR** and **OPR** lower bounds with respect to $n$:

**Corollary 3.6** (Non-monotonic scaling *w.r.t.* $n$ (B.4)). *For conciseness, denote $d_{\mathrm{w2s}}(N) = d_{s \wedge w} + (d_w - d_{s \wedge w})d_s/N$ and $\varrho = (O(\rho_w) + \rho_s)/\sigma^2$. Under the same setting as Proposition 3.5, with $n = d_w + q + 1$ for some $q \in \mathbb{N}$, the lower bound of $\mathbf{PGR}$ is maximized when $q = 0$, where*

$$\mathbf{PGR} \geqslant 1 - d_{\mathrm{w2s}}(N)/d_w;$$

*while the lower bound of* **OPR** *is maximized when* $q = \sqrt{(d_w + 1)\, d_{\text{w2s}}(N)/\varrho}$*, where*

$$\mathbf{OPR} \geqslant d_s \left( \sqrt{d_{\text{w2s}}(N)} + \sqrt{\varrho\,(d_w + 1)} \right)^{-2} .$$

Such non-monotonic scaling for **PGR** with respect to $n$ coincides with some empirical observations in (Burns et al., 2024) on NLP tasks. While the variance of $f_{\text{w2s}}$ in Theorem 3.1 decreases monotonically as $n$ grows, so do those of the reference models $f_w$, $f_s$, and $f_c$. With non-negligible FT approximation errors, as $n$ increases, the **PGR** and **OPR** lower bounds decrease with the improvements in bias but increase with the improvements in variance. Therefore, the optimal $n$ for the lower bounds of **PGR** and **OPR** is determined by the trade-off between variance and bias.

Assuming $\rho_s + \rho_w \ll \sigma^2$ in Definition 2.2, we have $\varrho \ll 1$. Again, consider two cases depending on $d_{s \wedge w}$:

(a) If $d_{s \wedge w} > 0$, we have $d_{\text{w2s}}(N) \lesssim d_{s \wedge w}$ when $N \gtrsim (d_w/d_{s \wedge w} - 1)d_s$, implying large **PGR** and **OPR** when $d_{s \wedge w} \ll \min\{d_s, d_w\}$.

(b) If $d_{s \wedge w} = 0$, we have $d_{\text{w2s}}(N) = d_w d_s/N$, implying large **PGR** and **OPR** when $N \gg \max\{d_w, d_s\}$.

## 4. Experiments

We conduct experiments to validate the theoretical findings on both synthetic and real tasks. In this section, we focus on two illustrative settings: synthetic regression (Section 4.1) and real-world image regression (Section 4.2). For brevity, we defer more experiments on image and sentiment classification tasks to Appendices E.2 and E.3, respectively.

### 4.1. Synthetic regression

We start by grounding the theoretical framework introduced in Section 2 with synthetic regression tasks.

**Setup.** We concretize the downstream task $\mathcal{D}(f_*)$ as a regression problem over Gaussian features. Let $f_* : \mathbb{R}^d \to \mathbb{R}$ be a linear function in a high-dimensional feature space $d = 20,000$ of form $f_*(\mathbf{x}) = \mathbf{x}^\top \Lambda_*^{1/2} \boldsymbol{\theta}_*$ where $\Lambda_* = \text{diag}(\lambda_1^*, \cdots, \lambda_d^*)$ is a diagonal matrix with a low rank $d_* = 300$ such that $\lambda_i^* = i^{-1}$ for $i \leqslant d_*$ and $\lambda_i^* = 0$ otherwise; and $\boldsymbol{\theta}_* \in \mathbb{R}^d$ is a random unit vector. Every sample $(\mathbf{x}, y) \sim \mathcal{D}(f_*)$ is generated by $\mathbf{x} \sim \mathcal{N}(\mathbf{0}_d, \mathbf{I}_d)$ and $y = f_*(\mathbf{x}) + z$ with $z \sim \mathcal{N}(0, \sigma^2)$. Given $\mathbf{x}$, the associated strong and weak features in Assumption 2.1 are generated by $\phi_s(\mathbf{x}) = \Sigma_s^{1/2}\mathbf{x}$ and $\phi_w(\mathbf{x}) = \Sigma_w^{1/2}\mathbf{x}$, with intrinsic dimensions $d_s = 100$ and $d_w = 200$ such that $\Sigma_s = \sum_{i=1}^{d_s} \lambda_i^* \mathbf{e}_i \mathbf{e}_i^\top$ and $\Sigma_w = \sum_{i=d_s-d_{s \wedge w}+1}^{d_w+d_s-d_{s \wedge w}} \lambda_i^* \mathbf{e}_i \mathbf{e}_i^\top$. For all synthetic experiments, we have $\rho_s + \rho_w < 0.0004$.

In the experiments, we vary $d_{s \wedge w}$ to control the student-teacher correlation and $\sigma^2$ to control the dominance of vari-

ance over bias (characterized by $\rho_s, \rho_w$). Each error bar reflects the standard deviation over 40 runs.

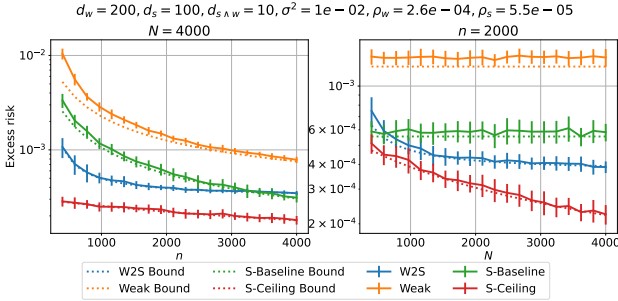

*Figure 1.* Scaling for excess risks on the synthetic regression task in a *variance-dominated regime* with a *low correlation dimension*.

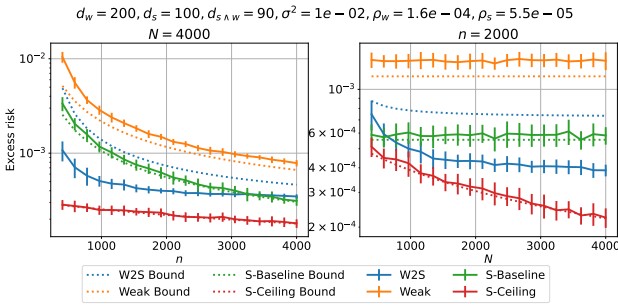

*Figure 2.* Scaling for excess risks on the synthetic regression task in a *variance-dominated regime* with a *high correlation dimension*.

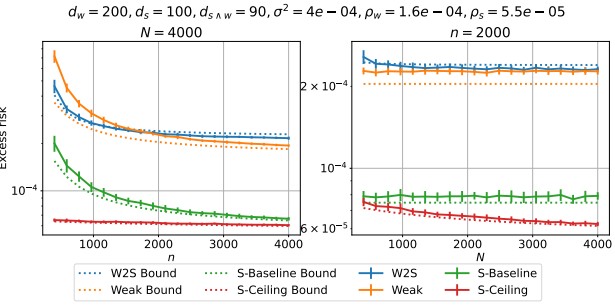

*Figure 3.* Scaling for excess risks on the synthetic regression task when *the variance is not dominant*, $\sigma^2 \approx \rho_s + \rho_w$.

**Scaling for generalization errors.** Figures 1 to 3 show scaling for $\mathbf{ER}(f_{\text{w2s}})$ (W2S), $\mathbf{ER}(f_w)$ (Weak), $\mathbf{ER}(f_s)$ (S-Baseline), and $\mathbf{ER}(f_c)$ (S-Ceiling) with respect to the sample sizes $n, N$. The dashes show theoretical predictions in Theorem 3.1, Proposition 3.3, and Corollary 3.4, consistent with the empirical measurements shown in the solid lines. In particular, we consider three cases:

- Figure 1: When variance dominates ($\sigma^2 = 0.01 \gg \rho_w + \rho_s$), with a low correlation dimension $d_{s \wedge w} = 10$,

$f_{\text{w2s}}$ outperforms both $f_w$ and $f_s$ for a moderate $n$ and a large enough $N$. However, larger sample sizes do not necessarily lead to better W2S generalization in a relative sense. For example, when $n$ keeps increasing, the strong baseline $f_s$ eventually outperforms $f_{\text{w2s}}$.

- Figure 2: When variance dominates, with a high correlation dimension $d_{s\wedge w} = 90$, $f_{\text{w2s}}$ still generalizes better than $f_w$ but fails to outperform the strong baseline $f_s$.

- Figure 3: When the variance is low (not dominant, *e.g.* $\sigma^2 = 0.0004 \approx \rho_s + \rho_w$), $f_{\text{w2s}}$ can fail to outperform $f_w$. This suggests that variance reduction is a key advantage of W2S over supervised FT.

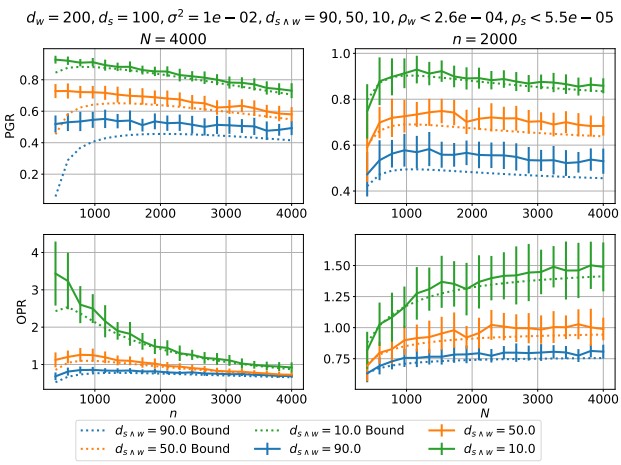

*Figure 4.* Scaling for **PGR** and **OPR** under different $d_{s\wedge w}$ on the synthetic regression task in a *variance-dominated regime*.

**Scaling for PGR and OPR.** Figure 4 show the scaling for **PGR** and **OPR** with respect to sample sizes $n, N$ in the variance-dominated regime (with small non-negligible FT approximation errors), at three different correlation dimensions $d_{s\wedge w} = 90, 50, 10$. The solid and dashed lines represent the empirical measurements and lower bounds in (11), (12), respectively.

- Coinciding with the theoretical predictions in Corollary 3.6 and the performance gaps between W2S and the references in Figure 1, we observe that the relative W2S performance in terms of **PGR** and **OPR** can degenerate as $n$ increases, while the larger $N$ generally leads to better W2S generalization in the relative sense.

- The lower correlation dimension $d_{s\wedge w}$ leads to higher **PGR** and **OPR**, *i.e.* larger discrepancy between the strong and weak features improves W2S generalization.

### 4.2. UTKFace regression

Beyond the synthetic regression, we investigate W2S on a real-world image regression task – age estimation on the

UTKFace dataset (Zhang et al., 2017). Each error bar in this section reflects standard deviation of 10 runs.

**Dataset.** UTKFace (Aligned & Cropped) (Zhang et al., 2017) consists of $23,708$ face images with age labels ranging from 0 to 116. We preprocess the images to $224 \times 224$ pixels and split the dataset into training and testing sets of sizes $20,000$ and $3,708$. Generalization errors are estimated with the mean squared error (MSE) over the test set.

**Linear probing over pretrained features.** We fix the strong student as CLIP ViT-B/32 (Radford et al., 2021) (`CLIP-B32`) and vary the weak teacher among the ResNet series (He et al., 2016) (`ResNet18`, `ResNet34`, `ResNet50`, `ResNet101`, `ResNet152`). We treat the backbones of these models (excluding the classification layers) as $\phi_s, \phi_w$ and finetune them via linear probing. We use ridge regression with a small fixed regularization hyperparameter $\alpha_w, \alpha_{\text{w2s}}, \alpha_s, \alpha_c = 10^{-6}$, close to the machine epsilon of single precision floating point numbers.

**Intrinsic dimension.** The intrinsic dimensions $d_w, d_s$ are measured based on the empirical covariance matrices $\mathbf{\Sigma}_w, \mathbf{\Sigma}_s$ of the weak and strong features over the entire dataset (including training and testing). As mentioned in Footnote 1, these covariances generally have fast-decaying eigenvalues (but not exactly low-rank) in practice, effectively leading to low intrinsic dimensions under ridge regression. We estimate such low intrinsic dimensions as the minimum rank for the best low-rank approximation of $\mathbf{\Sigma}_w, \mathbf{\Sigma}_s$ with a relative error in trace less than $\tau = 0.01$.

**Correlation dimension.** The pretrained feature dimensions (or the finetunable parameter counts) of the weak and strong models can be different in practice (see Appendix E.1, Table 1). We introduce an estimation for $d_{s\wedge w}$ in this case. Consider the (truncated) spectral decompositions $[\![\mathbf{\Sigma}_s]\!]_{d_s} = \mathbf{V}_s \mathbf{\Lambda}_s \mathbf{V}_s^\top$ and $[\![\mathbf{\Sigma}_w]\!]_{d_w} = \mathbf{V}_w \mathbf{\Lambda}_w \mathbf{V}_w^\top$ of two empirical covariances with different feature dimensions $D_s, D_w$ such that $\mathbf{V}_s \in \mathbb{R}^{D_s \times d_s}$ and $\mathbf{V}_w \in \mathbb{R}^{D_w \times d_w}$ consists of the top $d_s, d_w$ orthonormal eigenvectors, respectively. We estimate the correlation dimension $d_{s\wedge w}$ under different feature dimensions $D_s \neq D_w$ by matching the dimensions through a random unitary matrix (Vershynin, 2018) $\mathbf{\Gamma} \in \mathbb{R}^{D_s \times D_w}$: $d_{s\wedge w} = \|\mathbf{V}_s^\top \mathbf{\Gamma} \mathbf{V}_w\|_F^2$. This provides a good estimation for $d_{s\wedge w}$ because with low intrinsic dimensions $\max\{d_s, d_w\} \ll D_s, D_w$ in practice, mild dimension reduction through $\mathbf{\Gamma}$ well preserves the essential information in $\mathbf{V}_s, \mathbf{V}_w$.

**Discrepancies lead to better W2S.** Figure 5 shows the scaling of **PGR** and **OPR** with respect to the sample sizes $n, N$ for different weak teachers in the ResNet series with respect to a fixed student, `CLIP-B32`. We first observe that

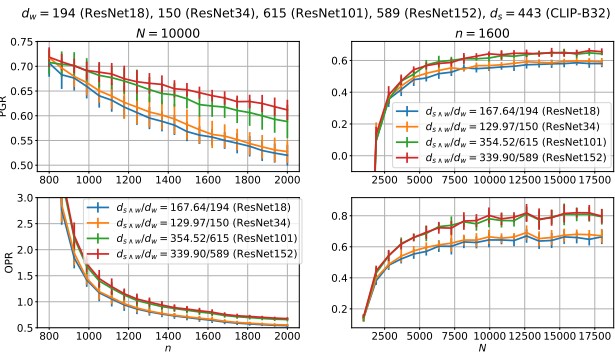

*Figure 5.* Scaling for **PGR** and **OPR** of different weak teachers with a fixed strong student on UTKFace. The legends show the comparison between $d_{s \wedge w}$ and $d_w$.

the relative W2S performance in terms of **PGR** and **OPR** is closely related to the correlation dimension $d_{s \wedge w}$ and the intrinsic dimensions $d_s, d_w$.

- When the strong student has a lower intrinsic dimension than the weak teacher (as widely observed in practice (Aghajanyan et al., 2021)), *i.e.* $d_s < d_w$, the relative W2S performance tends to be better than when $d_s > d_w$.

- The relative W2S performance tends to be better when $d_{s \wedge w}/d_w$ is lower, *i.e.* the larger discrepancy between weak and strong features leads to better W2S generalization.

Meanwhile, both **PGR** and **OPR** scale inversely with the labeled sample size $n$ and exhibit diminishing return with respect to the increasing pseudolabel size $N$, consistent with the theoretical predictions in Corollary 3.6 and the synthetic experiments in Figure 4.

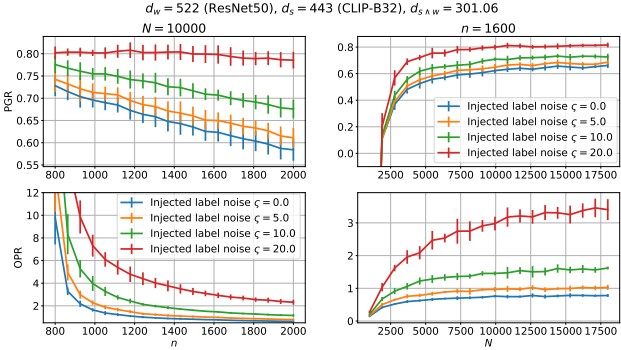

*Figure 6.* Scaling for **PGR** and **OPR** on UTKFace with injected label noise: $y_i \leftarrow y_i + \zeta_i$ where $\zeta_i \sim \mathcal{N}(0, \varsigma^2)$ *i.i.d.*.

**Variance reduction is a key advantage of W2S.** To investigate the impact of variance on W2S generalization, we inject noise to the training label by $y_i \leftarrow y_i + \zeta_i$ where $\zeta_i \sim \mathcal{N}(0, \varsigma^2)$ *i.i.d.*, and $\varsigma$ controls the injected labels noise

level. In Figure 6, we show the scaling for **PGR** and **OPR** with respect to the sample sizes $n, N$ under different noise levels $\varsigma$. We observe that the relative W2S performance in terms of **PGR** and **OPR** improves as the noise level $\varsigma$ increases. This provides empirical evidence that variance reduction is a key advantage of W2S over supervised FT, highlighting the importance of understanding the mechanisms of W2S in the variance-dominated regime.

## 5. Limitations and future directions

In this work, we introduce a theoretical framework for understanding the mechanism of weak-to-strong (W2S) generalization in the variance-dominated regime where both the student and teacher have sufficient capacities for the downstream task. Leveraging the low intrinsic dimensionality of finetuning (FT), we characterize model capacities from three perspectives: FT approximation errors for "accuracy", intrinsic dimensions for "complexity", and student-teacher correlation for "alignment". Our analysis shows that W2S generalization is driven by variance reduction in the discrepancy between the weak teacher and strong student features. This generalization analysis is followed by a case study on the relative W2S performance in terms of performance gap recovery (PGR) and outperforming ratio (OPR). We show that while larger sample sizes imply better W2S generalization in an absolute sense, the relative W2S performance can degenerate as the sample size increases. Our results provide theoretical insights into the choice of weak teachers and sample sizes in W2S pipelines.

An interesting implication of our analysis is that the mechanism of W2S may differ as the balance between variance and bias shifts. In the variance-dominated regime studied in this work, W2S can benefit from a lower intrinsic dimension of the strong student due to the resulting variance reduction in the subspace of discrepancy from the weak teacher. In contrast, in the bias-dominated regime, the lower approximation error of the strong student is generally brought by the larger "capacity" of the strong model corresponding to a higher intrinsic dimension (Ildiz et al., 2024; Wu & Sahai, 2024). This calls for future studies on unified views and transitions between the two regimes, which will provide a more comprehensive understanding of W2S. Toward this goal, a limitation of our analysis is the quantification of the advantage of W2S in bias (see Footnote 6), which could be a promising next step.

## Acknowledgements

The authors would like to thank Denny Wu for insightful discussions. The experiments are supported by the PLI computing cluster. YD acknowledges support of NYU Courant Instructorship. JDL acknowledges support of Open Philanthropy, NSF IIS 2107304, NSF CCF 2212262, NSF CA-

REER Award 2144994, and NSF CCF 2019844. This material is based upon work supported by the U.S. Department of Energy, Office of Science Energy Earthshot Initiative as part of the project "Learning reduced models under extreme data conditions for design and rapid decision-making in complex systems" under Award #DE-SC0024721.

## Impact Statement

This paper presents work whose goal is to advance the field of machine learning. There are many potential societal consequences of our work, none of which we feel must be specifically highlighted here.

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

# Appendix

## Contents

## A. Additional related works

**Knowledge distillation.** Knowledge distillation (KD) (Hinton, 2015; Gou et al., 2021) is closely connected to W2S generalization regarding the teacher-student setup, while W2S reverts the capacities of teacher and student in KD. In KD, a strong teacher model guides a weak student model to learn the teacher's knowledge. In contrast, W2S generalization occurs when a strong student model surpasses a weak teacher model under weak supervision. Phuong & Lampert (2019); Stanton et al. (2021); Ojha et al. (2023); Nagarajan et al. (2023); Dong et al. (2024b); Ildiz et al. (2024) conducted rigorous statistical analyses for the student's generalization from knowledge distillation. From the analysis perspective, a key difference between KD and W2S is that W2S is usually analyzed in the context of finetuning since the notions of "weak" and "strong" are built upon pretraining. This finetuning perspective introduces distinct angles from KD for examining intrinsic dimension (Li et al., 2018) and student-teacher correlation in W2S.

**Self-distillation and self-training.** In contrast to W2S, which considers distinct student and teacher models, self-distillation (Zhang et al., 2019; 2021) and related paradigms such as Born-Again Networks (Furlanello et al., 2018) use the same or progressively refined architectures to iteratively distill knowledge from a "previous version" of the model. There have been extensive theoretical analyses toward understanding the mechanism behind self-distillation (Mobahi et al., 2020; Das & Sanghavi, 2023; Borup & Andersen, 2023; Pareek et al., 2024).

Self-training (Scudder, 1965; Lee et al., 2013) is a closely related method to self-distillation that takes a single model's confident predictions to create pseudo-labels for unlabeled data and refines that model iteratively. Wei et al. (2020); Oymak & Gulcu (2021); Frei et al. (2022) provide theoretical insights into the generalization of self-training. In particular, Wei et al. (2020) introduced a theoretical framework based on neighborhood expansion, which was later on extended to various settings of weakly supervised learning, including domain adaptation (Cai et al., 2021), contrastive learning (Shen et al., 2022; Huang et al., 2021), consistency regularization (Yang et al., 2023; Dong et al., 2023), and recently weak-to-strong generalization (Lang et al., 2024; Shin et al., 2024).

## B. Proofs in Section 3

With respect to any sample size $n \in \mathbb{N}$, let

$$\rho_s(n) = \mathbb{E}_{\mathbf{X} \sim \mathcal{D}^n}[\|\phi_s(\mathbf{X})\phi_s(\mathbf{X})^\dagger f_*(\mathbf{X}) - f_*(\mathbf{X})\|_2^2],$$
$$\rho_w(n) = \mathbb{E}_{\mathbf{X} \sim \mathcal{D}^n}[\|\phi_w(\mathbf{X})\phi_w(\mathbf{X})^\dagger f_*(\mathbf{X}) - f_*(\mathbf{X})\|_2^2],$$

where $\phi_s(\mathbf{X})$ and $\phi_w(\mathbf{X})$ are $n \times d$ feature matrices; and $f_*(\mathbf{X}) \in \mathbb{R}^n$ is a vector of the noiseless ground truth labels.

**Lemma B.1.** *Given the FT approximation errors $\rho_s$ and $\rho_w$ in Definition 2.2, we have*

$$\rho_s(n) \leqslant n\rho_s \quad and \quad \rho_w(n) \leqslant n\rho_w \quad \forall\, n \in \mathbb{N}.$$

*Proof of Lemma B.1.* Let $\boldsymbol{\theta}_* = \operatorname{argmin}_{\boldsymbol{\theta} \in \mathbb{R}^d} \mathbb{E}_{\mathbf{x} \sim \mathcal{D}}[(\phi_w(\mathbf{x})^\top \boldsymbol{\theta} - f_*(\mathbf{x}))^2]$ such that

$$\mathbb{E}_{\mathbf{x} \sim \mathcal{D}}[(\phi_w(\mathbf{x})^\top \boldsymbol{\theta}_* - f_*(\mathbf{x}))^2] = \rho_w.$$

Then, by observing that conditioned on $\mathbf{X}$,

$$\phi_w(\mathbf{X})^\dagger f_*(\mathbf{X}) = \operatorname*{argmin}_{\boldsymbol{\theta} \in \mathbb{R}^d} \|\phi_w(\mathbf{X})\boldsymbol{\theta} - f_*(\mathbf{X})\|_2^2,$$

we have

$$\begin{aligned}
\rho_w(n) &= \mathbb{E}_{\mathbf{X} \sim \mathcal{D}^n}\left[\|\phi_w(\mathbf{X})\phi_w(\mathbf{X})^\dagger f_*(\mathbf{X}) - f_*(\mathbf{X})\|_2^2\right] \\
&\leqslant \mathbb{E}_{\mathbf{X} \sim \mathcal{D}^n}\left[\|\phi_w(\mathbf{X})\boldsymbol{\theta}_* - f_*(\mathbf{X})\|_2^2\right] \\
&= n\, \mathbb{E}_{\mathbf{X} \sim \mathcal{D}^n}\left[\frac{1}{n}\|\phi_w(\mathbf{X})\boldsymbol{\theta}_* - f_*(\mathbf{X})\|_2^2\right] \\
&= n\, \mathbb{E}_{\mathbf{x} \sim \mathcal{D}}\left[(\phi_w(\mathbf{x})^\top \boldsymbol{\theta}_* - f_*(\mathbf{x}))^2\right] \\
&= n\, \rho_w.
\end{aligned}$$

The proof for $\rho_s(n)$ follows analogously. $\qquad\square$

### B.1. Proof of Theorem 3.1

**Theorem B.2** (Formal restatement of Theorem 3.1). *Consider $f_{\mathrm{w2s}}(\mathbf{x}) = \phi_s(\mathbf{x})^\top \boldsymbol{\theta}_{\mathrm{w2s}}$ finetuned as in* (1), (2) *with both $\alpha_w, \alpha_{\mathrm{w2s}} \to 0$. Under Assumptions 2.1 and 2.6, when $n \geqslant \Omega(d_w)$, the excess risk $\mathbf{ER}(f_{\mathrm{w2s}}) = \mathbf{Var}(f_{\mathrm{w2s}}) + \mathbf{Bias}(f_{\mathrm{w2s}})$ satisfies*

$$\mathbf{Var}(f_{\mathrm{w2s}}) \lesssim \frac{\sigma^2}{n}\left(d_{s \wedge w} + \frac{d_s}{N}(d_w - d_{s \wedge w})\right),$$

$$\mathbf{Bias}(f_{\mathrm{w2s}}) \leqslant \mathbf{Bias}(f_w) + \rho_s \leqslant O(\rho_w) + \rho_s.$$

*Moreover, when $\phi_w(\mathbf{x}) \sim \mathcal{N}(\mathbf{0}_d, \boldsymbol{\Sigma}_w)$, for any $n > d_w + 1$, we have*

$$\mathbf{Var}(f_{\mathrm{w2s}}) = \frac{\sigma^2}{n - d_w - 1}\left(d_{s \wedge w} + \frac{d_s}{N}(d_w - d_{s \wedge w})\right).$$

*Proof of Theorem 3.1 and Theorem B.2.* We first observe that the solution of (1) as $\alpha_w \to 0$ is given by

$$\boldsymbol{\theta}_w = \widetilde{\boldsymbol{\Phi}}_w^\dagger \widetilde{\mathbf{y}} = \widetilde{\boldsymbol{\Phi}}_w^\dagger (\widetilde{\mathbf{f}}_* + \widetilde{\mathbf{z}}),$$

where $\widetilde{\mathbf{z}} \sim \mathcal{N}(\mathbf{0}_n, \sigma^2 \mathbf{I}_n)$. Meanwhile, the solution of (2) as $\alpha_{\mathrm{w2s}} \to 0$ is given by

$$\boldsymbol{\theta}_{\mathrm{w2s}} = \boldsymbol{\Phi}_s^\dagger \boldsymbol{\Phi}_w \boldsymbol{\theta}_w = \boldsymbol{\Phi}_s^\dagger \boldsymbol{\Phi}_w \widetilde{\boldsymbol{\Phi}}_w^\dagger (\widetilde{\mathbf{f}}_* + \widetilde{\mathbf{z}}).$$

Then, the excess risk of $f_{\mathrm{w2s}}$ can be decomposed into variance and bias as follows:

$$\begin{aligned}
\mathbf{ER}(f_{\mathrm{w2s}}) &= \mathbb{E}_{\mathcal{S}_x, \widetilde{\mathcal{S}}}\left[\frac{1}{N}\|\boldsymbol{\Phi}_s \boldsymbol{\theta}_{\mathrm{w2s}} - \mathbf{f}_*\|_2^2\right] \\
&= \mathbb{E}_{\mathcal{S}_x, \widetilde{\mathcal{S}}}\left[\frac{1}{N}\left\|(\boldsymbol{\Phi}_s \boldsymbol{\Phi}_s^\dagger \boldsymbol{\Phi}_w \widetilde{\boldsymbol{\Phi}}_w^\dagger \widetilde{\mathbf{f}}_* - \mathbf{f}_*) + \boldsymbol{\Phi}_s \boldsymbol{\Phi}_s^\dagger \boldsymbol{\Phi}_w \widetilde{\boldsymbol{\Phi}}_w^\dagger \widetilde{\mathbf{z}}\right\|_2^2\right] \\
&= \underbrace{\frac{1}{N}\mathbb{E}_{\mathcal{S}_x, \widetilde{\mathcal{S}}}\left[\left\|\boldsymbol{\Phi}_s \boldsymbol{\Phi}_s^\dagger \boldsymbol{\Phi}_w \widetilde{\boldsymbol{\Phi}}_w^\dagger \widetilde{\mathbf{z}}\right\|_2^2\right]}_{\mathbf{Var}(f_{\mathrm{w2s}})} + \underbrace{\frac{1}{N}\mathbb{E}_{\mathcal{S}_x, \widetilde{\mathcal{S}}}\left[\left\|\boldsymbol{\Phi}_s \boldsymbol{\Phi}_s^\dagger \boldsymbol{\Phi}_w \widetilde{\boldsymbol{\Phi}}_w^\dagger \widetilde{\mathbf{f}}_* - \mathbf{f}_*\right\|_2^2\right]}_{\mathbf{Bias}(f_{\mathrm{w2s}})}.
\end{aligned}$$

Recall the spectral decomposition $\boldsymbol{\Sigma}_w = \mathbf{V}_w \boldsymbol{\Lambda}_w \mathbf{V}_w^\top$. Since $\mathbb{E}_{\mathbf{x} \sim \mathcal{D}}[\phi_w(\mathbf{x})\phi_w(\mathbf{x})^\top] = \boldsymbol{\Sigma}_w$, for each $\mathbf{x} \sim \mathcal{D}$, we can write $\phi_w(\mathbf{x}) = \boldsymbol{\Sigma}_w^{1/2}\boldsymbol{\gamma}$, where $\boldsymbol{\gamma} \in \mathbb{R}^d$ is an independent random vector that is zero-mean and isotropic (*i.e.* $\mathbb{E}[\boldsymbol{\gamma}] = \mathbf{0}_d$ and $\mathbb{E}[\boldsymbol{\gamma}\boldsymbol{\gamma}^\top] = \mathbf{I}_d$). The same holds for $\boldsymbol{\Sigma}_s = \mathbf{V}_s \boldsymbol{\Lambda}_s \mathbf{V}_s^\top$ and $\phi_s(\mathbf{x}) = \boldsymbol{\Sigma}_s^{1/2}\boldsymbol{\gamma}$.

Then, for $\mathcal{S}$ and $\widetilde{\mathcal{S}}$, there exist independent random matrices $\boldsymbol{\Gamma} = [\boldsymbol{\gamma}_1, \ldots, \boldsymbol{\gamma}_N]^\top \in \mathbb{R}^{N \times d}$ and $\widetilde{\boldsymbol{\Gamma}} = [\widetilde{\boldsymbol{\gamma}}_1, \ldots, \widetilde{\boldsymbol{\gamma}}_n]^\top \in \mathbb{R}^{n \times d}$ consisting of *i.i.d.* zero-mean isotropic rows such that

$$
\begin{aligned}
\boldsymbol{\Phi}_w \boldsymbol{\Sigma}_w^{-1/2} &= \boldsymbol{\Gamma}\boldsymbol{\Sigma}_w^{1/2}\boldsymbol{\Sigma}_w^{-1/2} = \boldsymbol{\Gamma}\mathbf{V}_w\mathbf{V}_w^\top, \\
\widetilde{\boldsymbol{\Phi}}_w \boldsymbol{\Sigma}_w^{-1/2} &= \widetilde{\boldsymbol{\Gamma}}\boldsymbol{\Sigma}_w^{1/2}\boldsymbol{\Sigma}_w^{-1/2} = \widetilde{\boldsymbol{\Gamma}}\mathbf{V}_w\mathbf{V}_w^\top, \\
\boldsymbol{\Phi}_s \boldsymbol{\Sigma}_s^{-1/2} &= \boldsymbol{\Gamma}\boldsymbol{\Sigma}_s^{1/2}\boldsymbol{\Sigma}_s^{-1/2} = \boldsymbol{\Gamma}\mathbf{V}_s\mathbf{V}_s^\top, \\
\widetilde{\boldsymbol{\Phi}}_s \boldsymbol{\Sigma}_s^{-1/2} &= \widetilde{\boldsymbol{\Gamma}}\boldsymbol{\Sigma}_s^{1/2}\boldsymbol{\Sigma}_s^{-1/2} = \widetilde{\boldsymbol{\Gamma}}\mathbf{V}_s\mathbf{V}_s^\top.
\end{aligned}
\tag{7}
$$

Let $\boldsymbol{\Gamma}_w = \boldsymbol{\Gamma}\mathbf{V}_w \in \mathbb{R}^{N \times d_w}$ and $\widetilde{\boldsymbol{\Gamma}}_w = \widetilde{\boldsymbol{\Gamma}}\mathbf{V}_w \in \mathbb{R}^{n \times d_w}$ throughout the proof.

**Bias.** For the bias term, by observing that $\mathbf{P}_s = \boldsymbol{\Phi}_s\boldsymbol{\Phi}_s^\dagger$ is an $N \times N$ orthogonal projection, we can decompose the bias term as

$$
\mathbf{Bias}(f_{\text{w2s}}) = \mathbb{E}_{\mathcal{S}_x, \widetilde{\mathcal{S}}}\left[\frac{1}{N}\left\|\mathbf{P}_s\left(\boldsymbol{\Phi}_w\widetilde{\boldsymbol{\Phi}}_w^\dagger\widetilde{\mathbf{f}}_* - \mathbf{f}_*\right)\right\|_2^2\right] + \frac{1}{N}\mathbb{E}_{\mathcal{S}_x}\left[\left\|(\mathbf{I}_N - \mathbf{P}_s)\,\mathbf{f}_*\right\|_2^2\right],
$$

where $\mathbb{E}_{\mathcal{S}_x}\left[\left\|(\mathbf{I}_N - \mathbf{P}_s)\,\mathbf{f}_*\right\|_2^2\right] = \rho_s(N)$ by Definition 2.2, and

$$
\frac{1}{N}\mathbb{E}_{\mathcal{S}_x}\left[\left\|(\mathbf{I}_N - \mathbf{P}_s)\,\mathbf{f}_*\right\|_2^2\right] = \frac{\rho_s(N)}{N} \leqslant \rho_s.
$$

For the first term, since $\mathbf{P}_s$ is an orthogonal projection, we have

$$
\mathbb{E}_{\mathcal{S}_x, \widetilde{\mathcal{S}}}\left[\frac{1}{N}\left\|\mathbf{P}_s\left(\boldsymbol{\Phi}_w\widetilde{\boldsymbol{\Phi}}_w^\dagger\widetilde{\mathbf{f}}_* - \mathbf{f}_*\right)\right\|_2^2\right] \leqslant \mathbb{E}_{\mathcal{S}_x, \widetilde{\mathcal{S}}}\left[\frac{1}{N}\left\|\boldsymbol{\Phi}_w\widetilde{\boldsymbol{\Phi}}_w^\dagger\widetilde{\mathbf{f}}_* - \mathbf{f}_*\right\|_2^2\right] = \mathbf{Bias}(f_w).
$$

Notice that when $\mathbf{Bias}(f_w) > 0$ and $d_s < d_w$ such that $\boldsymbol{\Phi}_w\widetilde{\boldsymbol{\Phi}}_w^\dagger\widetilde{\mathbf{f}}_* - \widetilde{\mathbf{f}}_* \notin \mathrm{Range}(\boldsymbol{\Phi}_s)$ almost surely, the inequality is strict, *i.e.*, $\mathbf{Bias}(f_{\text{w2s}}) < \mathbf{Bias}(f_w) + \rho_s$. Overall, we have

$$
\mathbf{Bias}(f_{\text{w2s}}) \leqslant \mathbf{Bias}(f_w) + \rho_s \leqslant O(\rho_w) + \rho_s,
$$

where the second inequality follows from Proposition 3.3.

**Variance.** For the variance term, we observe that

$$
\begin{aligned}
\mathbf{Var}(f_{\text{w2s}}) &= \frac{1}{N}\mathbb{E}_{\mathcal{S}_x, \widetilde{\mathcal{S}}}\left[\left\|\mathbf{P}_s\boldsymbol{\Phi}_w\widetilde{\boldsymbol{\Phi}}_w^\dagger\widetilde{\mathbf{z}}\right\|_2^2\right] \\
&= \frac{1}{N}\mathbb{E}_{\mathcal{S}_x, \widetilde{\mathcal{S}}}\left[\mathrm{tr}\left(\boldsymbol{\Phi}_w^\top\mathbf{P}_s\boldsymbol{\Phi}_w\widetilde{\boldsymbol{\Phi}}_w^\dagger\widetilde{\mathbf{z}}\widetilde{\mathbf{z}}^\top(\widetilde{\boldsymbol{\Phi}}_w^\dagger)^\top\right)\right] \\
&= \frac{\sigma^2}{N}\mathbb{E}_{\mathcal{S}_x, \widetilde{\mathcal{S}}}\left[\mathrm{tr}\left(\boldsymbol{\Phi}_w^\top\mathbf{P}_s\boldsymbol{\Phi}_w(\widetilde{\boldsymbol{\Phi}}_w^\top\widetilde{\boldsymbol{\Phi}}_w)^\dagger\right)\right],
\end{aligned}
$$

which implies

$$
\mathbf{Var}(f_{\text{w2s}}) = \frac{\sigma^2}{N}\mathrm{tr}\left(\mathbb{E}_{\mathcal{S}_x}\left[\boldsymbol{\Sigma}_w^{-1/2}\boldsymbol{\Phi}_w^\top\mathbf{P}_s\boldsymbol{\Phi}_w\boldsymbol{\Sigma}_w^{-1/2}\right]\mathbb{E}_{\widetilde{\mathcal{S}}}\left[\left(\boldsymbol{\Sigma}_w^{-1/2}\widetilde{\boldsymbol{\Phi}}_w^\top\widetilde{\boldsymbol{\Phi}}_w\boldsymbol{\Sigma}_w^{-1/2}\right)^\dagger\right]\right).
\tag{8}
$$

We observe that

$$
\mathbb{E}_{\widetilde{\mathcal{S}}}\left[\left(\boldsymbol{\Sigma}_w^{-1/2}\widetilde{\boldsymbol{\Phi}}_w^\top\widetilde{\boldsymbol{\Phi}}_w\boldsymbol{\Sigma}_w^{-1/2}\right)^\dagger\right] = \mathbb{E}_{\widetilde{\mathcal{S}}}\left[\left(\mathbf{V}_w\widetilde{\boldsymbol{\Gamma}}_w^\top\widetilde{\boldsymbol{\Gamma}}_w\mathbf{V}_w^\top\right)^\dagger\right] = \mathbf{V}_w\mathbb{E}_{\widetilde{\mathcal{S}}}\left[\left(\widetilde{\boldsymbol{\Gamma}}_w^\top\widetilde{\boldsymbol{\Gamma}}_w\right)^\dagger\right]\mathbf{V}_w^\top.
$$

Now, we consider the following two cases for the feature distribution of $\phi_w(\mathbf{x})$, corresponding to the distribution of $\boldsymbol{\Gamma}_w$ and $\widetilde{\boldsymbol{\Gamma}}_w$:

(a) **Gaussian features**: In Theorem 3.1, assuming $\phi_w(\mathbf{x}) \sim \mathcal{N}(\mathbf{0}_d, \boldsymbol{\Sigma}_w)$ such that $\widetilde{\boldsymbol{\Gamma}}_w$ consists of *i.i.d.* Gaussian rows, we have $\widetilde{\boldsymbol{\gamma}}_i \sim \mathcal{N}(\mathbf{0}_{d_w}, \mathbf{I}_{d_w})$. Notice that under the assumption $n > d_w + 1$, $\operatorname{rank}(\widetilde{\boldsymbol{\Gamma}}_w) = d_w$ almost surely, and therefore $\widetilde{\boldsymbol{\Gamma}}_w^\top \widetilde{\boldsymbol{\Gamma}}_w$ is invertible.

Meanwhile, with $\widetilde{\boldsymbol{\gamma}}_i \sim \mathcal{N}(\mathbf{0}_{d_w}, \mathbf{I}_{d_w})$ for all $i \in [n]$, $(\widetilde{\boldsymbol{\Gamma}}_w^\top \widetilde{\boldsymbol{\Gamma}}_w) \sim \mathcal{W}(\mathbf{I}_{d_w}, n)$ follows the Wishart distribution (Wishart, 1928, Definition 3.4.1) with $n$ degrees of freedom and scale matrix $\mathbf{I}_{d_w}$. Therefore, $(\widetilde{\boldsymbol{\Gamma}}_w^\top \widetilde{\boldsymbol{\Gamma}}_w)^{-1} \sim \mathcal{W}^{-1}(\mathbf{I}_{d_w}, n)$ follows the inverse Wishart distribution (Mardia et al., 2024, §3.8), whose mean takes the form (Mardia et al., 2024, (3.8.3))

$$\mathbb{E}_{\widetilde{\mathcal{S}}}\left[(\widetilde{\boldsymbol{\Gamma}}_w^\top \widetilde{\boldsymbol{\Gamma}}_w)^\dagger\right] = \frac{1}{n - d_w - 1}\mathbf{I}_{d_w}.$$

Then, we have

$$\mathbb{E}_{\widetilde{\mathcal{S}}}\left[\left(\boldsymbol{\Sigma}_w^{-1/2}\widetilde{\boldsymbol{\Phi}}_w^\top \widetilde{\boldsymbol{\Phi}}_w \boldsymbol{\Sigma}_w^{-1/2}\right)^\dagger\right] = \frac{1}{n - d_w - 1}\mathbf{V}_w \mathbf{V}_w^\top.$$

Therefore, (8) implies

$$
\begin{aligned}
\mathbf{Var}(f_{\mathrm{w2s}}) &= \frac{\sigma^2}{N}\frac{1}{n - d_w - 1}\operatorname{tr}\left(\mathbf{V}_w^\top \mathbb{E}_{\mathcal{S}_x}\left[\boldsymbol{\Sigma}_w^{-1/2}\boldsymbol{\Phi}_w^\top \mathbf{P}_s \boldsymbol{\Phi}_w \boldsymbol{\Sigma}_w^{-1/2}\right]\mathbf{V}_w\right) \\
&= \frac{\sigma^2}{N}\frac{1}{n - d_w - 1}\operatorname{tr}\left(\mathbb{E}_{\mathcal{S}_x}\left[\mathbf{V}_w^\top \mathbf{V}_w \boldsymbol{\Gamma}_w^\top \mathbf{P}_s \boldsymbol{\Gamma}_w \mathbf{V}_w^\top \mathbf{V}_w\right]\right) \\
&= \frac{\sigma^2}{N}\frac{1}{n - d_w - 1}\operatorname{tr}\left(\mathbb{E}_{\mathcal{S}_x}\left[\boldsymbol{\Gamma}_w^\top \mathbf{P}_s \boldsymbol{\Gamma}_w\right]\right).
\end{aligned}
\tag{9}
$$

Recall that $\mathbf{P}_s = \boldsymbol{\Phi}_s \boldsymbol{\Phi}_s^\dagger$. Let $\boldsymbol{\Gamma}_s = \boldsymbol{\Gamma}\mathbf{V}_s \in \mathbb{R}^{N \times d_s}$, and we can write

$$\mathbf{P}_s = (\boldsymbol{\Phi}_s \boldsymbol{\Sigma}_s^{-1/2})(\boldsymbol{\Phi}_s \boldsymbol{\Sigma}_s^{-1/2})^\dagger = (\boldsymbol{\Gamma}_s \mathbf{V}_s^\top)(\boldsymbol{\Gamma}_s \mathbf{V}_s^\top)^\dagger = \boldsymbol{\Gamma}_s \boldsymbol{\Gamma}_s^\dagger.$$

Therefore, with $\boldsymbol{\Gamma}_w = \boldsymbol{\Gamma}\mathbf{V}_w$ and $\boldsymbol{\Gamma}_s = \boldsymbol{\Gamma}\mathbf{V}_s$, we can decompose

$$
\begin{aligned}
\operatorname{tr}\left(\mathbb{E}_{\mathcal{S}_x}\left[\boldsymbol{\Gamma}_w^\top \mathbf{P}_s \boldsymbol{\Gamma}_w\right]\right) &= \mathbb{E}_{\mathcal{S}_x}\left[\operatorname{tr}\left(\boldsymbol{\Gamma}_w^\top \boldsymbol{\Gamma}_s \boldsymbol{\Gamma}_s^\dagger \boldsymbol{\Gamma}_w\right)\right] \\
&= \mathbb{E}_{\mathcal{S}_x}\left[\operatorname{tr}\left(\mathbf{V}_w^\top \mathbf{V}_s \mathbf{V}_s^\top \mathbf{V}_w \boldsymbol{\Gamma}_w^\top \boldsymbol{\Gamma}_s \boldsymbol{\Gamma}_s^\dagger \boldsymbol{\Gamma}_w\right)\right] \\
&\quad + \mathbb{E}_{\mathcal{S}_x}\left[\operatorname{tr}\left(\mathbf{V}_w^\top (\mathbf{I}_d - \mathbf{V}_s \mathbf{V}_s^\top)\mathbf{V}_w \boldsymbol{\Gamma}_w^\top \boldsymbol{\Gamma}_s \boldsymbol{\Gamma}_s^\dagger \boldsymbol{\Gamma}_w\right)\right].
\end{aligned}
$$

For the first term, since $\boldsymbol{\Gamma}_w \mathbf{V}_w^\top \mathbf{V}_s = \boldsymbol{\Gamma}\mathbf{V}_w \mathbf{V}_w^\top \mathbf{V}_s$ and $\boldsymbol{\Gamma}_s = \boldsymbol{\Gamma}\mathbf{V}_s$, the range of $\boldsymbol{\Gamma}_w \mathbf{V}_w^\top \mathbf{V}_s$ is a subspace of that of $\boldsymbol{\Gamma}_s$ and therefore,

$$
\begin{aligned}
\mathbb{E}_{\mathcal{S}_x}\left[\operatorname{tr}\left(\mathbf{V}_w^\top \mathbf{V}_s \mathbf{V}_s^\top \mathbf{V}_w \boldsymbol{\Gamma}_w^\top \boldsymbol{\Gamma}_s \boldsymbol{\Gamma}_s^\dagger \boldsymbol{\Gamma}_w\right)\right] &= \mathbb{E}_{\mathcal{S}_x}\left[\operatorname{tr}\left(\mathbf{V}_s^\top \mathbf{V}_w \boldsymbol{\Gamma}_w^\top \boldsymbol{\Gamma}_s \boldsymbol{\Gamma}_s^\dagger \boldsymbol{\Gamma}_w \mathbf{V}_w^\top \mathbf{V}_s\right)\right] \\
&= \mathbb{E}_{\mathcal{S}_x}\left[\operatorname{tr}\left(\mathbf{V}_s^\top \mathbf{V}_w \boldsymbol{\Gamma}_w^\top \boldsymbol{\Gamma}_w \mathbf{V}_w^\top \mathbf{V}_s\right)\right] \\
&= \operatorname{tr}\left(\mathbf{V}_s^\top \mathbf{V}_w \mathbb{E}_{\mathcal{S}_x}\left[\boldsymbol{\Gamma}_w^\top \boldsymbol{\Gamma}_w\right]\mathbf{V}_w^\top \mathbf{V}_s\right).
\end{aligned}
$$

Since $\mathbb{E}_{\mathcal{S}_x}\left[\boldsymbol{\Gamma}_w^\top \boldsymbol{\Gamma}_w\right] = N\mathbf{I}_{d_w}$, we have

$$
\begin{aligned}
\mathbb{E}_{\mathcal{S}_x}\left[\operatorname{tr}\left(\mathbf{V}_w^\top \mathbf{V}_s \mathbf{V}_s^\top \mathbf{V}_w \boldsymbol{\Gamma}_w^\top \boldsymbol{\Gamma}_s \boldsymbol{\Gamma}_s^\dagger \boldsymbol{\Gamma}_w\right)\right] &= N\operatorname{tr}\left(\mathbf{V}_s^\top \mathbf{V}_w \mathbf{V}_w^\top \mathbf{V}_s\right) \\
&= N\left\|\mathbf{V}_s^\top \mathbf{V}_w\right\|_F^2 \\
&= Nd_{s \wedge w}.
\end{aligned}
$$

For the second term, we first observe that the row space of $\boldsymbol{\Gamma}_w \mathbf{V}_w^\top (\mathbf{I}_d - \mathbf{V}_s \mathbf{V}_s^\top)$ is orthogonal to that of $\boldsymbol{\Gamma}_s = \boldsymbol{\Gamma}\mathbf{V}_s$, and therefore, $\boldsymbol{\Gamma}_w \mathbf{V}_w^\top (\mathbf{I}_d - \mathbf{V}_s \mathbf{V}_s^\top)$ and $\boldsymbol{\Gamma}_s$ are independent, which implies

$$\mathbb{E}_{\mathcal{S}_x}\left[\operatorname{tr}\left(\mathbf{V}_w^\top (\mathbf{I}_d - \mathbf{V}_s \mathbf{V}_s^\top)\mathbf{V}_w \boldsymbol{\Gamma}_w^\top \boldsymbol{\Gamma}_s \boldsymbol{\Gamma}_s^\dagger \boldsymbol{\Gamma}_w\right)\right] = \operatorname{tr}\left(\mathbb{E}\left[\boldsymbol{\Gamma}_w \mathbf{V}_w^\top (\mathbf{I}_d - \mathbf{V}_s \mathbf{V}_s^\top)\mathbf{V}_w \boldsymbol{\Gamma}_w^\top\right]\mathbb{E}\left[\boldsymbol{\Gamma}_s \boldsymbol{\Gamma}_s^\dagger\right]\right).$$

Since $\mathbf{\Gamma}$ consists of independent isotropic rows, so do $\mathbf{\Gamma}_s = \mathbf{\Gamma}\mathbf{V}_s \in \mathbb{R}^{N \times d_s}$ and $\mathbf{\Gamma}_w = \mathbf{\Gamma}\mathbf{V}_w \in \mathbb{R}^{N \times d_w}$, which implies

$$\mathbb{E}\left[\mathbf{\Gamma}_s \mathbf{\Gamma}_s^\dagger\right] = \frac{d_s}{N}\, \mathbf{I}_N \quad \text{and} \quad \mathbb{E}\left[\mathbf{\Gamma}_w^\top \mathbf{\Gamma}_w\right] = N\, \mathbf{I}_{d_w}.$$

Then, we have

$$\begin{aligned}
\mathbb{E}_{\mathcal{S}_x}\left[\operatorname{tr}\left(\mathbf{V}_w^\top(\mathbf{I}_d - \mathbf{V}_s\mathbf{V}_s^\top)\mathbf{V}_w \mathbf{\Gamma}_w^\top \mathbf{\Gamma}_s \mathbf{\Gamma}_s^\dagger \mathbf{\Gamma}_w\right)\right] &= \operatorname{tr}\left(\mathbb{E}\left[\mathbf{\Gamma}_w\mathbf{V}_w^\top(\mathbf{I}_d - \mathbf{V}_s\mathbf{V}_s^\top)\mathbf{V}_w\mathbf{\Gamma}_w^\top\right]\mathbb{E}\left[\mathbf{\Gamma}_s\mathbf{\Gamma}_s^\dagger\right]\right) \\
&= \frac{d_s}{N}\operatorname{tr}\left(\mathbb{E}\left[\mathbf{\Gamma}_w\mathbf{V}_w^\top(\mathbf{I}_d - \mathbf{V}_s\mathbf{V}_s^\top)\mathbf{V}_w\mathbf{\Gamma}_w^\top\right]\right) \\
&= \frac{d_s}{N}\operatorname{tr}\left(\mathbf{V}_w^\top(\mathbf{I}_d - \mathbf{V}_s\mathbf{V}_s^\top)\mathbf{V}_w\mathbb{E}\left[\mathbf{\Gamma}_w^\top\mathbf{\Gamma}_w\right]\right) \\
&= \frac{d_s}{N}N\operatorname{tr}\left(\mathbf{V}_w^\top(\mathbf{I}_d - \mathbf{V}_s\mathbf{V}_s^\top)\mathbf{V}_w\right) \\
&= d_s(d_w - d_{s\wedge w}).
\end{aligned}$$

Combining the two terms, we have

$$\operatorname{tr}\left(\mathbb{E}_{\mathcal{S}_x}\left[\mathbf{\Gamma}_w^\top \mathbf{P}_s \mathbf{\Gamma}_w\right]\right) = Nd_{s\wedge w} + d_s(d_w - d_{s\wedge w}).$$

Then, by (9), the variance is exactly characterized by

$$\begin{aligned}
\mathbf{Var}(f_{\mathrm{w2s}}) &= \frac{\sigma^2}{N}\frac{Nd_{s\wedge w} + d_s(d_w - d_{s\wedge w})}{n - d_w - 1} \\
&= \frac{\sigma^2}{n - d_w - 1}\left(d_{s\wedge w} + \frac{d_s}{N}(d_w - d_{s\wedge w})\right).
\end{aligned}$$

(b) **Sub-gaussian features**: Relaxing the Gaussian feature assumption, when $\widetilde{\mathbf{\Gamma}}_w$ consists of *i.i.d.* sub-gaussian random vectors that are zero-mean and isotropic (*i.e.* $\mathbb{E}[\widetilde{\gamma}_i] = \mathbf{0}_{d_w}$ and $\mathbb{E}[\widetilde{\gamma}_i\widetilde{\gamma}_i^\top] = \mathbf{I}_{d_w}$), with $n \geqslant \Omega(d_w)$, Lemma B.3 implies that

$$\mathbb{E}_{\widetilde{\mathcal{S}}}\left[(\widetilde{\mathbf{\Gamma}}_w^\top\widetilde{\mathbf{\Gamma}}_w)^\dagger\right] \preccurlyeq O\left(\frac{1}{n}\right)\mathbf{I}_{d_w},$$

and therefore,

$$\mathbb{E}_{\widetilde{\mathcal{S}}}\left[\left(\mathbf{\Sigma}_w^{-1/2}\widetilde{\mathbf{\Phi}}_w^\top\widetilde{\mathbf{\Phi}}_w\mathbf{\Sigma}_w^{-1/2}\right)^\dagger\right] \preccurlyeq O\left(\frac{1}{n}\right)\mathbf{V}_w\mathbf{V}_w^\top.$$

Then, via an analogous argument as (9), (8) implies that

$$\mathbf{Var}(f_{\mathrm{w2s}}) \leqslant \frac{\sigma^2}{N}O\left(\frac{1}{n}\right)\operatorname{tr}\left(\mathbb{E}_{\mathcal{S}_x}\left[\mathbf{\Gamma}_w^\top\mathbf{P}_s\mathbf{\Gamma}_w\right]\right). \tag{10}$$

We observe that in the analysis of the Gaussian feature case, the characterization

$$\operatorname{tr}\left(\mathbb{E}_{\mathcal{S}_x}\left[\mathbf{\Gamma}_w^\top\mathbf{P}_s\mathbf{\Gamma}_w\right]\right) = (N - d_s)d_{s\wedge w} + d_s d_w$$

does not involve the Gaussianity of $\mathbf{\Gamma}$ and therefore holds for general subgaussian features. This leads to an upper bound on the variance:

$$\begin{aligned}
\mathbf{Var}(f_{\mathrm{w2s}}) &\leqslant \frac{\sigma^2}{N}O\left(\frac{1}{n}\right)(Nd_{s\wedge w} + d_s(d_w - d_{s\wedge w})) \\
&\lesssim \frac{\sigma^2}{n}\left(d_{s\wedge w} + \frac{d_s}{N}(d_w - d_{s\wedge w})\right).
\end{aligned}$$

$\square$

**Lemma B.3** (Adapting (Vershynin, 2010) Theorem 5.39). *Let $\widetilde{\mathbf{\Gamma}}_w = [\widetilde{\gamma}_1, \ldots, \widetilde{\gamma}_n]^\top$ be an $n \times d_w$ matrix whose rows $\widetilde{\gamma}_1, \ldots, \widetilde{\gamma}_n$ consist of i.i.d. sub-gaussian random vectors that are zero-mean and isotropic (i.e. $\mathbb{E}[\widetilde{\gamma}_i] = \mathbf{0}_{d_w}$ and $\mathbb{E}[\widetilde{\gamma}_i \widetilde{\gamma}_i^\top] = \mathbf{I}_{d_w}$). When $n \geqslant \Omega(d_w)$, we have*

$$\mathbb{E}\left[\left\|\left(\widetilde{\mathbf{\Gamma}}_w^\top \widetilde{\mathbf{\Gamma}}_w\right)^\dagger\right\|_2\right] \leqslant O\left(\frac{1}{n}\right),$$

*where $\Omega(\cdot)$ and $O(\cdot)$ suppresses constants that depend only on the sub-gaussian norm $\|\widetilde{\gamma}_i\|_{\psi_2} = \sup_{\mathbf{v} \in \mathbb{S}^{d_w - 1}} \sup_{p \geqslant 1} (\mathbb{E}[|\widetilde{\gamma}_i^\top \mathbf{v}|^p])^{1/p}/\sqrt{p}$, independent of $n, d_w$.*

*Proof of Lemma B.3.* Let $\sigma_{\min}(\widetilde{\mathbf{\Gamma}}_w^\top \widetilde{\mathbf{\Gamma}}_w)$ be the smallest singular value of $\widetilde{\mathbf{\Gamma}}_w^\top \widetilde{\mathbf{\Gamma}}_w$. Leveraging (Vershynin, 2010) Theorem 5.39, we notice that for $n \geqslant \Omega(d_w)$, there exist constants $c_1, c_2 > 0$ that depend only on the sub-gaussian norm $\|\widetilde{\gamma}_i\|_{\psi_2}$ such that

$$\Pr\left[\sigma_{\min}(\widetilde{\mathbf{\Gamma}}_w^\top \widetilde{\mathbf{\Gamma}}_w) < \left(\sqrt{n} - c_1\sqrt{d_w} - t\right)^2\right] \leqslant \exp\left(-c_2 t^2\right).$$

Therefore, we have

$$\Pr\left[\frac{1}{\sigma_{\min}(\widetilde{\mathbf{\Gamma}}_w^\top \widetilde{\mathbf{\Gamma}}_w)} > t\right] \leqslant \exp\left(-c_2\left(\sqrt{n} - c_1\sqrt{d_w} - \sqrt{\frac{1}{t}}\right)^2\right).$$

Notice that for any non-negative random variable $Z$ with a cumulative density function $F_Z(z)$,

$$\begin{aligned}\mathbb{E}[Z] &= \int_0^\infty z dF_Z(z) = -\int_0^\infty z d\left(1 - F_Z(z)\right) \\ &= \left[z\left(1 - F_Z(z)\right)\right]_0^\infty + \int_0^\infty \left(1 - F_Z(z)\right) dz \\ &= \int_0^\infty \Pr\left[Z > z\right] dz.\end{aligned}$$

Therefore, we have

$$\mathbb{E}\left[\frac{1}{\sigma_{\min}(\widetilde{\mathbf{\Gamma}}_w^\top \widetilde{\mathbf{\Gamma}}_w)}\right] \leqslant \int_0^\infty \exp\left(-c_2\left(\sqrt{n} - c_1\sqrt{d_w} - \sqrt{\frac{1}{t}}\right)^2\right) dt.$$

Let $t_0 = 1/\left(\sqrt{n} - c_1\sqrt{d_w}\right)^2$ such that $\sqrt{n} - c_1\sqrt{d_w} - \sqrt{\frac{1}{t}} = 0$ and

$$\int_0^{t_0} \exp\left(-c_2\left(\sqrt{n} - c_1\sqrt{d_w} - \sqrt{\frac{1}{t}}\right)^2\right) dt \leqslant t_0$$

Then, we have

$$\mathbb{E}\left[\frac{1}{\sigma_{\min}(\widetilde{\mathbf{\Gamma}}_w^\top\widetilde{\mathbf{\Gamma}}_w)}\right] \leqslant \int_0^\infty \exp\left(-c_2\left(\sqrt{n} - c_1\sqrt{d_w} - \sqrt{\frac{1}{t}}\right)^2\right) dt$$

$$\leqslant t_0 + \int_{t_0}^\infty \exp\left(-c_2\left(\sqrt{n} - c_1\sqrt{d_w} - \sqrt{\frac{1}{t}}\right)^2\right) dt$$

$$= t_0 + 2\int_0^{\sqrt{n}-c_1\sqrt{d_w}} \exp\left(-c_2 u^2\right)\left(\sqrt{n} - c_1\sqrt{d_w} - u\right)^{-3} du$$

$$= t_0 + \frac{2}{\left(\sqrt{n} - c_1\sqrt{d_w}\right)^2}\int_0^1 \exp\left(-c_2\left(\sqrt{n} - c_1\sqrt{d_w}\right)^2 u^2\right)(1-u)^{-3} du$$

$$= \frac{1}{\left(\sqrt{n} - c_1\sqrt{d_w}\right)^2} + \frac{2}{\left(\sqrt{n} - c_1\sqrt{d_w}\right)^2}\left(\int_0^1 \exp\left(-\Omega\left(u^2\right)\right)(1-u)^{-3} du\right)$$

$$= O\left(\frac{1}{\left(\sqrt{n} - c_1\sqrt{d_w}\right)^2}\right).$$

When $n \geqslant \Omega(d_w)$, we have $\sqrt{n} - c_1\sqrt{d_w} \geqslant \Omega(\sqrt{n})$, and therefore ,

$$\mathbb{E}\left[\left\|\left(\widetilde{\mathbf{\Gamma}}_w^\top\widetilde{\mathbf{\Gamma}}_w\right)^\dagger\right\|_2\right] \leqslant \mathbb{E}\left[\frac{1}{\sigma_{\min}(\widetilde{\mathbf{\Gamma}}_w^\top\widetilde{\mathbf{\Gamma}}_w)}\right] \leqslant O\left(\frac{1}{n}\right).$$

$\square$

## B.2. Proof of Proposition 3.3 and Corollary 3.4

*Proof of Proposition 3.3 and Corollary 3.4.* The excess risk of the finetuned weak teacher $f_w(\mathbf{x}) = \phi_w(\mathbf{x})^\top\boldsymbol{\theta}_w$ can be expressed as

$$\mathbf{ER}(f_w) = \mathbb{E}_{\mathcal{S}_x,\widetilde{\mathcal{S}}}\left[\frac{1}{N}\|\mathbf{\Phi}_w\boldsymbol{\theta}_w - \mathbf{f}_*\|_2^2\right],$$

where $\mathbf{f}_* = [\mathbf{f}_*(\mathbf{x}_1), \ldots, \mathbf{f}_*(\mathbf{x}_N)]^\top \in \mathbb{R}^N$; and we recall that $\mathbf{\Phi}_w = [\phi_w(\mathbf{x}_1), \ldots, \phi_w(\mathbf{x}_N)]^\top$. Notice that the randomness of $\boldsymbol{\theta}_w$ comes from the SFT samples $\widetilde{\mathcal{S}} \sim \mathcal{D}(f_*)^n$.

Observe that the solution of (1) as $\alpha_w \to 0$ is given by $\boldsymbol{\theta}_w = \widetilde{\mathbf{\Phi}}_w^\dagger\widetilde{\mathbf{y}}$, where $\widetilde{\mathbf{y}} = \widetilde{\mathbf{f}}_* + \widetilde{\mathbf{z}}$ is the noisy label vector with $\widetilde{\mathbf{z}} \sim \mathcal{N}(\mathbf{0}_n, \sigma^2\mathbf{I}_n)$. Therefore, with the randomness over $\widetilde{\mathcal{S}} \sim \mathcal{D}(f_*)^n$, we have

$$\mathbf{ER}(f_w) = \mathbb{E}\left[\frac{1}{N}\left\|\mathbf{\Phi}_w\widetilde{\mathbf{\Phi}}_w^\dagger\widetilde{\mathbf{y}} - \mathbf{f}_*\right\|_2^2\right]$$

$$= \mathbb{E}\left[\frac{1}{N}\left\|\mathbf{\Phi}_w\widetilde{\mathbf{\Phi}}_w^\dagger\widetilde{\mathbf{z}} + \left(\mathbf{\Phi}_w\widetilde{\mathbf{\Phi}}_w^\dagger\widetilde{\mathbf{f}}_* - \mathbf{f}_*\right)\right\|_2^2\right]$$

$$= \underbrace{\mathbb{E}\left[\frac{1}{N}\left\|\mathbf{\Phi}_w\widetilde{\mathbf{\Phi}}_w^\dagger\widetilde{\mathbf{z}}\right\|_2^2\right]}_{\mathbf{Var}(f_w)} + \underbrace{\mathbb{E}\left[\frac{1}{N}\left\|\mathbf{\Phi}_w\widetilde{\mathbf{\Phi}}_w^\dagger\widetilde{\mathbf{f}}_* - \mathbf{f}_*\right\|_2^2\right]}_{\mathbf{Bias}(f_w)}.$$

For bias, leveraging Lemma B.3 and by the definition of finetuning capacity (see Definition 2.2), we have

$$\mathbf{Bias}(f_w) = \mathbb{E}\left[\frac{1}{N}\left\|\mathbf{\Phi}_w\widetilde{\mathbf{\Phi}}_w^\dagger\widetilde{\mathbf{f}}_* - \mathbf{f}_*\right\|_2^2\right]$$

$$= \mathbb{E}_{\widetilde{\mathcal{S}}}\left[\mathbb{E}_{\mathbf{x}\sim\mathcal{D}}\left[\left(\phi_w(\mathbf{x})^\top\widetilde{\mathbf{\Phi}}_w^\dagger\widetilde{\mathbf{f}}_* - f_*(\mathbf{x})\right)^2\right]\right]$$

$$\lesssim \mathbb{E}_{\widetilde{\mathcal{S}}}\left[\frac{1}{n}\left\|\widetilde{\mathbf{\Phi}}_w\widetilde{\mathbf{\Phi}}_w^\dagger\widetilde{\mathbf{f}}_* - \widetilde{\mathbf{f}}_*\right\|_2^2\right] = \frac{\rho_w(n)}{n},$$

We observe that $\mathbf{Bias}(f_w) \lesssim \frac{\rho_w(n)}{n} \leqslant \rho_w$ by Lemma B.1.

For variance, we observe that

$$\mathbf{Var}(f_w) = \frac{1}{N}\mathbb{E}\left[\left\|\mathbf{\Phi}_w\widetilde{\mathbf{\Phi}}_w^\dagger\widetilde{\mathbf{z}}\right\|_2^2\right] = \mathbb{E}\left[\mathrm{tr}\left(\mathbf{\Sigma}_w\widetilde{\mathbf{\Phi}}_w^\dagger\widetilde{\mathbf{z}}\widetilde{\mathbf{z}}^\top(\widetilde{\mathbf{\Phi}}_w^\dagger)^\top\right)\right]$$

$$= \sigma^2\mathbb{E}\left[\mathrm{tr}\left((\mathbf{\Sigma}_w^{-1/2}\widetilde{\mathbf{\Phi}}_w^\top\widetilde{\mathbf{\Phi}}_w\mathbf{\Sigma}_w^{-1/2})^\dagger\right)\right] = \sigma^2\mathbb{E}\left[\mathrm{tr}\left((\widetilde{\mathbf{\Gamma}}_w^\top\widetilde{\mathbf{\Gamma}}_w)^{-1}\right)\right].$$

Assuming $\phi_w(\mathbf{x}) \sim \mathcal{N}(\mathbf{0}_d, \mathbf{\Sigma}_w)$, we then have $\mathbb{E}\left[\mathrm{tr}\left((\widetilde{\mathbf{\Gamma}}_w^\top\widetilde{\mathbf{\Gamma}}_w)^{-1}\right)\right] = \frac{d_w}{n - d_w - 1}$ and

$$\mathbf{Var}(f_w) = \frac{\sigma^2 d_w}{n - d_w - 1}.$$

Meanwhile, assuming $\phi_w(\mathbf{x})$ is sub-gaussian, Lemma B.3 implies that $\mathbb{E}\left[\mathrm{tr}\left((\widetilde{\mathbf{\Gamma}}_w^\top\widetilde{\mathbf{\Gamma}}_w)^{-1}\right)\right] \lesssim O\left(\frac{d_w}{n}\right)$, and therefore,

$$\mathbf{Var}(f_w) \lesssim \frac{\sigma^2 d_w}{n}.$$

The analogous results hold for the strong SFT model: $\mathbf{ER}(f_s) = \mathbf{Bias}(f_s) + \mathbf{Var}(f_s)$ satisfies

$$\mathbf{Bias}(f_s) = \mathbb{E}\left[\frac{1}{N}\left\|\mathbf{\Phi}_s\widetilde{\mathbf{\Phi}}_s^\dagger\widetilde{\mathbf{f}}_* - \mathbf{f}_*\right\|_2^2\right] \lesssim \frac{\rho_s(n)}{n} \leqslant \rho_s,$$

and

$$\mathbf{Var}(f_s) = \frac{\sigma^2 d_s}{n - d_s - 1} \quad \text{or} \quad \mathbf{Var}(f_s) \lesssim \frac{\sigma^2 d_s}{n},$$

when $\phi_s(\mathbf{x}) \sim \mathcal{N}(\mathbf{0}_d, \mathbf{\Sigma}_s)$ or $\phi_s(\mathbf{x})$ is sub-gaussian, respectively.

For the strong ceiling model $f_c$, the variance, $\mathbf{Var}(f_c)$, follows from the standard generalization analysis for fixed design linear regression, and the bias, $\mathbf{Bias}(f_c)$, follows directly from the definition of $\rho_s(n + N)$. $\qquad\square$

### B.3. Proof of Proposition 3.5

*Proof of Proposition 3.5.* Noticing that with $\mathrm{rank}(\widetilde{\mathbf{\Phi}}_w) = d_w$ and $\mathrm{rank}(\widetilde{\mathbf{\Phi}}_s) = \mathrm{rank}(\mathbf{\Phi}_s) = d_s$ almost surely, the excess risks of $f_w, f_s, f_c$ are characterized exactly in Proposition 3.3 and Corollary 3.4, and $\mathbf{ER}(f_{\mathrm{w2s}})$ is upper bounded by Theorem 3.1. Therefore, by directly plugging in the excess risks to the definitions of PGR and OPR, we have

$$
\begin{aligned}
\mathbf{PGR} =& \frac{\mathbf{ER}(f_w) - \mathbf{ER}(f_{\mathrm{w2s}})}{\mathbf{ER}(f_w) - \mathbf{ER}(f_c)} \\
\geqslant& \left(\frac{\sigma^2 d_w}{n - d_w - 1} + \mathbf{Bias}(f_w) - \frac{\sigma^2}{n - d_w - 1}\left(d_{s\wedge w} + \frac{d_s}{N}(d_w - d_{s\wedge w})\right) - (\mathbf{Bias}(f_w) + \rho_s)\right) \\
& \left(\frac{\sigma^2 d_w}{n - d_w - 1} + \mathbf{Bias}(f_w) - \sigma^2\frac{d_s}{N + n} - \rho_s\right)^{-1} \\
\geqslant& \left(\frac{\sigma^2 d_w}{n - d_w - 1} - \sigma^2\frac{d_{s\wedge w} + (d_w - d_{s\wedge w})d_s/N}{n - d_w - 1} - \rho_s\right) \bigg/ \left(\frac{\sigma^2 d_w}{n - d_w - 1} + O\left(\rho_w\right)\right),
\end{aligned}
\tag{11}
$$

and

$$\mathbf{OPR} = \frac{\mathbf{ER}(f_s)}{\mathbf{ER}(f_{\mathrm{w2s}})} \geqslant \frac{\sigma^2 d_s}{n - d_s - 1} \bigg/ \left(\sigma^2\frac{d_{s\wedge w} + (d_w - d_{s\wedge w})d_s/N}{n - d_w - 1} + O(\rho_w) + \rho_s\right). \tag{12}$$

When taking $n = d_w + q + 1$ for some small constant $q \in \mathbb{N}$, we observe that

$$
\begin{aligned}
\mathbf{PGR} &\geqslant \left( \frac{\sigma^2 d_w}{n - d_w - 1} - \sigma^2 \frac{d_{s \wedge w} + (d_w - d_{s \wedge w})d_s/N}{n - d_w - 1} - \rho_s \right) \Big/ \left( \frac{\sigma^2 d_w}{n - d_w - 1} + O\left(\rho_w\right) \right) \\
&= \left( \frac{d_w}{q} - \frac{d_{s \wedge w}}{q} - \frac{d_s}{N} \frac{d_w - d_{s \wedge w}}{q} - \frac{\rho_s}{\sigma^2} \right) \Big/ \left( \frac{d_w}{q} + O\left(\frac{\rho_w}{\sigma^2}\right) \right) \\
&\geqslant \left( \frac{d_w}{q} - \frac{d_{s \wedge w}}{q} - \frac{d_s}{N} \frac{d_w - d_{s \wedge w}}{q} - \frac{\rho_s}{\sigma^2} - O\left(\frac{\rho_w}{\sigma^2}\right) \right) \Big/ \frac{d_w}{q} \\
&= 1 - \frac{d_{s \wedge w}}{d_w} - \frac{d_s}{N} \frac{d_w - d_{s \wedge w}}{d_w} - \frac{q}{d_w} \frac{O(\rho_w) + \rho_s}{\sigma^2},
\end{aligned}
$$

and

$$
\begin{aligned}
\mathbf{OPR} &\geqslant \frac{\sigma^2 d_s}{n - d_s - 1} \Big/ \left( \sigma^2 \frac{d_{s \wedge w} + (d_w - d_{s \wedge w})d_s/N}{n - d_w - 1} + O(\rho_w) + \rho_s \right) \\
&\geqslant \frac{d_s}{n} \Big/ \left( \frac{d_{s \wedge w} + (d_w - d_{s \wedge w})d_s/N}{q} + \frac{O(\rho_w) + \rho_s}{\sigma^2} \right) \\
&= \left( \frac{n}{q} \frac{d_{s \wedge w} + (d_w - d_{s \wedge w})d_s/N}{d_s} + \frac{n}{d_s} \frac{O(\rho_w) + \rho_s}{\sigma^2} \right)^{-1}.
\end{aligned}
$$

$\square$

### B.4. Proof of Corollary 3.6

*Proof of Corollary 3.6.* Recall the notations introduced for conciseness:

$$
d_{\mathrm{w2s}}(N) = d_{s \wedge w} + (d_w - d_{s \wedge w})\frac{d_s}{N}, \quad \varrho = \frac{O(\rho_w) + \rho_s}{\sigma^2}.
$$

Then, the lower bounds for $\mathbf{PGR}$ and $\mathbf{OPR}$ in Proposition 3.5 can be expressed in terms of $d_{\mathrm{w2s}}(N)$ and $\varrho$ as

$$
\mathbf{PGR} \geqslant 1 - \frac{d_{\mathrm{w2s}}(N)}{d_w} - \varrho \frac{q}{d_w},
$$

and

$$
\mathbf{OPR} \geqslant \left( \frac{d_{\mathrm{w2s}}(N)}{d_s} + \frac{d_w + 1}{d_s}\varrho + \frac{d_{\mathrm{w2s}}(N)}{d_s} \frac{d_w + 1}{q} + q\frac{\varrho}{d_s} \right)^{-1}.
$$

Both lower bound of $\mathbf{PGR}$ is maximized when the last term is minimized, *i.e.*, $q = 0$, which yields $\mathbf{PGR} \geqslant 1 - \frac{d_{\mathrm{w2s}}(N)}{d_w}$. Meanwhile, the lower bound of $\mathbf{OPR}$ is maximized when the last two terms in the expressions that involve $q$ are minimized, which is achieved when $q = \sqrt{(d_w + 1)\, d_{\mathrm{w2s}}(N)/\varrho}$. Substituting the optimal $q$ back into the expression yields the best lower bound for $\mathbf{OPR}$:

$$
\begin{aligned}
\mathbf{OPR} &\geqslant \left( \frac{d_{\mathrm{w2s}}(N)}{d_s} + \varrho\frac{d_w + 1}{d_s} + 2\sqrt{\varrho\frac{d_w + 1}{d_s} \frac{d_{\mathrm{w2s}}(N)}{d_s}} \right)^{-1} \\
&= \left( \sqrt{\frac{d_{\mathrm{w2s}}(N)}{d_s}} + \sqrt{\varrho\frac{d_w + 1}{d_s}} \right)^{-2}.
\end{aligned}
$$

$\square$

## C. Ridge regression analysis

In this section, we investigate the more realistic scenario where the weak and strong feature covariances are not exactly low-rank but admit a small number of dominating eigenvalues.

Concretely, we consider the same data distribution $(\mathbf{x}, y) \sim \mathcal{D}(f_*)$ with $y = f_*(\mathbf{x}) + z$ for some independent Gaussian label noise $z \sim \mathcal{N}(0, \sigma^2)$ and an unknown ground truth function $f_* : \mathcal{X} \to \mathbb{R}$ as in Section 2. Under the same sub-gaussian feature assumption as in Assumption 2.1, we adapt Definitions 2.3 and 2.4 to the ridge regression setting as follows.

**Assumption C.1** (Data distribution). Let $\phi_s : \mathcal{X} \to \mathbb{R}^d$ and $\phi_w : \mathcal{X} \to \mathbb{R}^d$ be the strong and weak pretrained models that take $\mathbf{x} \sim \mathcal{D}$ and output pretrained features $\phi_s(\mathbf{x}), \phi_w(\mathbf{x}) \in \mathbb{R}^d$, respectively.

(i) **Ground truth**: Assume $f_*$ can be expressed as a linear function over an unknown ground truth feature $\phi_* : \mathcal{X} \to \mathbb{R}^d$ such that $f_*(\cdot) = \phi_*(\cdot)^\top \boldsymbol{\theta}_*$ for some fixed $\boldsymbol{\theta}_* \in \mathbb{R}^d$.

(ii) **Sub-gaussian features** (Assumption 2.1): Let $\phi_w(\mathbf{x}), \phi_s(\mathbf{x}), \phi_*(\mathbf{x})$ be zero-mean sub-gaussian random vectors with $\mathbb{E}[\phi_w(\mathbf{x})] = \mathbb{E}[\phi_s(\mathbf{x})] = \mathbb{E}[\phi_*(\mathbf{x})] = \mathbf{0}_d$,

$$\mathbb{E}[\phi_w(\mathbf{x})\phi_w(\mathbf{x})^\top] = \boldsymbol{\Sigma}_w, \quad \mathbb{E}[\phi_s(\mathbf{x})\phi_s(\mathbf{x})^\top] = \boldsymbol{\Sigma}_s, \quad \mathbb{E}[\phi_*(\mathbf{x})\phi_*(\mathbf{x})^\top] = \boldsymbol{\Sigma}_*,$$

and $\boldsymbol{\Sigma}_s^{-1/2}\phi_s(\mathbf{x})$ satisfies for all canonical basis vectors $\mathbf{e}_i \in \mathbb{R}^d$ that

$$\mathbb{E}\left[\left(\mathbf{e}_i^\top \boldsymbol{\Sigma}_s^{-1/2}\phi_s(\mathbf{x})\right)^4\right] = 3,$$

(e.g., when $\phi_s(\mathbf{x})$ is a Gaussian random vector)[7]. We assume without loss of generality that these features are roughly normalized, i.e., $\|\boldsymbol{\Sigma}_w\|_2 \asymp 1$, $\|\boldsymbol{\Sigma}_s\|_2 \asymp 1$, and $\|\boldsymbol{\Sigma}_*\|_2 \asymp 1$.

(iii) **Low intrinsic dimension**: Let $\boldsymbol{\Sigma}_s$ and $\boldsymbol{\Sigma}_w$ both be **positive-definite** with spectral decompositions $\boldsymbol{\Sigma}_s = \mathbf{V}_s \boldsymbol{\Lambda}_s \mathbf{V}_s^\top$ and $\boldsymbol{\Sigma}_w = \mathbf{V}_w \boldsymbol{\Lambda}_w \mathbf{V}_w^\top$, where $\boldsymbol{\Lambda}_s, \boldsymbol{\Lambda}_w \in \mathbb{R}^{d \times d}$ are diagonal matrices with positive eigenvalues in decreasing order; while $\mathbf{V}_s \in \mathbb{R}^{d \times d}$ and $\mathbf{V}_w \in \mathbb{R}^{d \times d}$ are orthogonal matrices consisting of the corresponding orthonormal eigenvectors. The low intrinsic dimension of FT implies that $\boldsymbol{\Lambda}_s = \mathrm{diag}(\lambda_1^s, \cdots, \lambda_d^s)$ and $\boldsymbol{\Lambda}_w = \mathrm{diag}(\lambda_1^w, \cdots, \lambda_d^w)$ consist of a few dominating eigenvalues, while the rest of the eigenvalues are negligible, i.e., there exist $d_s, d_w \ll d$ such that $\sum_{i > d_s} \lambda_i^s \ll \mathrm{tr}(\boldsymbol{\Sigma}_s)$ and $\sum_{i > d_w} \lambda_i^w \ll \mathrm{tr}(\boldsymbol{\Sigma}_w)$. Here,

$$\mathrm{tr}(\boldsymbol{\Sigma}_s) \lesssim d_s \quad \text{and} \quad \mathrm{tr}(\boldsymbol{\Sigma}_w) \lesssim d_w$$

effectively measure the intrinsic dimensions of $\phi_s$ and $\phi_w$.

*Remark* C.2 (Weak-strong similarity). In place of correlation dimension (Definition 2.4) in the ridgeless setting, for the ridge regression analysis, we measure the similarity between the weak and strong models directly through $\mathrm{tr}(\boldsymbol{\Sigma}_s \boldsymbol{\Sigma}_w)$. Notice that

$$\mathrm{tr}(\boldsymbol{\Sigma}_s \boldsymbol{\Sigma}_w) \leqslant \min\left\{\mathrm{tr}(\boldsymbol{\Sigma}_s)\|\boldsymbol{\Sigma}_w\|_2, \mathrm{tr}(\boldsymbol{\Sigma}_w)\|\boldsymbol{\Sigma}_s\|_2\right\} \lesssim \min\left\{\mathrm{tr}(\boldsymbol{\Sigma}_s), \mathrm{tr}(\boldsymbol{\Sigma}_w)\right\}.$$

In particular, when $\boldsymbol{\Sigma}_s$ and $\boldsymbol{\Sigma}_w$ admit low intrinsic dimensions, $\mathrm{tr}(\boldsymbol{\Sigma}_s \boldsymbol{\Sigma}_w)$ can be much smaller than $\min\left\{\mathrm{tr}(\boldsymbol{\Sigma}_s), \mathrm{tr}(\boldsymbol{\Sigma}_w)\right\}$ if their eigenvectors corresponding to the dominating eigenvalues are almost orthogonal.

*Remark* C.3 (FT approximation errors). It is worth noting that under the ground truth and positive-definite covariance assumptions in Assumption C.1(i, iii), the FT approximation errors in Definition 2.2 satisfy

$$\begin{aligned}
\rho_s &= \min_{\boldsymbol{\theta} \in \mathbb{R}^d} \mathbb{E}_{\mathbf{x} \sim \mathcal{D}}\left[(\phi_s(\mathbf{x})^\top \boldsymbol{\theta} - f_*(\mathbf{x}))^2\right] = 0 \quad (\text{when } \boldsymbol{\theta} = \boldsymbol{\Sigma}_s^{-1}\boldsymbol{\Sigma}_* \boldsymbol{\theta}_*), \\
\rho_w &= \min_{\boldsymbol{\theta} \in \mathbb{R}^d} \mathbb{E}_{\mathbf{x} \sim \mathcal{D}}\left[(\phi_w(\mathbf{x})^\top \boldsymbol{\theta} - f_*(\mathbf{x}))^2\right] = 0 \quad (\text{when } \boldsymbol{\theta} = \boldsymbol{\Sigma}_w^{-1}\boldsymbol{\Sigma}_* \boldsymbol{\theta}_*).
\end{aligned} \tag{13}$$

In place of Definition 2.2, with positive-definite covariances in Assumption C.1, we measure the alignment between the ground truth feature $\phi_*$ and the weak/strong feature $\phi_w, \phi_s$ through

$$\varrho_s = \|\boldsymbol{\Sigma}_s^{-1/2}\boldsymbol{\Sigma}_*^{1/2}\boldsymbol{\theta}_*\|_2^2, \quad \varrho_w = \|\boldsymbol{\Sigma}_w^{-1/2}\boldsymbol{\Sigma}_*^{1/2}\boldsymbol{\theta}_*\|_2^2.$$

---

[7]We make this assumption only for the conciseness of final results. The assumption on the fourth moment can be relaxed to any $\mathbb{E}[(\mathbf{e}_i^\top \boldsymbol{\Sigma}_s^{-1/2}\phi_s(\mathbf{x}))^4] = 3 + \kappa$ ($\kappa \in \mathbb{R}$) at the cost of an additional term in the upper bound for variance:

$$\mathbf{Var}(f_{\mathrm{w2s}}) \leqslant \frac{\sigma^2}{4(\alpha_w n)(\alpha_{\mathrm{w2s}} N)}\left(\left(1 + \frac{1}{N}\right)\mathrm{tr}(\boldsymbol{\Sigma}_s \boldsymbol{\Sigma}_w) + \frac{1}{N}\left(\mathrm{tr}(\boldsymbol{\Sigma}_s)\mathrm{tr}(\boldsymbol{\Sigma}_w) + \kappa\,\mathrm{tr}(\mathrm{diag}(\boldsymbol{\Sigma}_s)\boldsymbol{\Sigma}_w)\right)\right),$$

which has negligible impact on the sample complexity when $N$ is large.

Intuitively, for $\boldsymbol{\Sigma}_s$ and $\boldsymbol{\Sigma}_w$ with a few dominating eigenvalues (Assumption C.1(iii)), $\varrho_s$ and $\varrho_w$ are small if the eigensubspace associated with non-negligible eigenvalues of $\boldsymbol{\Sigma}_*$ is fully covered by the eigensubspaces associated with the dominating eigenvalues of $\boldsymbol{\Sigma}_s$ and $\boldsymbol{\Sigma}_w$, respectively.

The W2S FT under ridge regression consists of two steps.

(a) First, the weak teacher $f_w(\mathbf{x}) = \phi_w(\mathbf{x})^\top \boldsymbol{\theta}_w$ is supervisedly finetuned over $\widetilde{\mathcal{S}}$:

$$\boldsymbol{\theta}_w = \underset{\boldsymbol{\theta} \in \mathbb{R}^d}{\operatorname{argmin}} \frac{1}{n} \left\| \widetilde{\boldsymbol{\Phi}}_w \boldsymbol{\theta} - \widetilde{\mathbf{y}} \right\|_2^2 + \alpha_w \left\| \boldsymbol{\theta} \right\|_2^2, \quad \alpha_w > 0. \tag{14}$$

(b) Then, the W2S model $f_{\text{w2s}}(\mathbf{x}) = \phi_s(\mathbf{x})^\top \boldsymbol{\theta}_{\text{w2s}}$ is finetuned over the strong feature $\phi_s$ through the unlabeled samples $\mathcal{S}_x$ and their pseudo-labels generated by the weak teacher model:

$$\boldsymbol{\theta}_{\text{w2s}} = \underset{\boldsymbol{\theta} \in \mathbb{R}^d}{\operatorname{argmin}} \frac{1}{N} \left\| \boldsymbol{\Phi}_s \boldsymbol{\theta} - \boldsymbol{\Phi}_w \boldsymbol{\theta}_w \right\|_2^2 + \alpha_{\text{w2s}} \left\| \boldsymbol{\theta} \right\|_2^2, \quad \alpha_{\text{w2s}} > 0. \tag{15}$$

**Theorem C.4** (W2S under ridge regression). *Let* $\varrho_w = \left\| \boldsymbol{\Sigma}_w^{-1/2} \boldsymbol{\Sigma}_*^{1/2} \boldsymbol{\theta}_* \right\|_2^2$ *and* $\varrho_s = \left\| \boldsymbol{\Sigma}_s^{-1/2} \boldsymbol{\Sigma}_*^{1/2} \boldsymbol{\theta}_* \right\|_2^2$. *Under Assumption C.1, the generalization error of W2S FT via ridge regression with fixed* $\alpha_w, \alpha_{\text{w2s}} > 0$, $\mathbf{ER}(f_{\text{w2s}}) = \mathbf{Var}(f_{\text{w2s}}) + \mathbf{Bias}(f_{\text{w2s}})$, *is upper bounded by*

$$\mathbf{Var}(f_{\text{w2s}}) \leqslant \frac{\sigma^2}{4(\alpha_w n)(\alpha_{\text{w2s}} N)} \left( \left( 1 + \frac{1}{N} \right) \operatorname{tr}\left( \boldsymbol{\Sigma}_s \boldsymbol{\Sigma}_w \right) + \frac{1}{N} \operatorname{tr}(\boldsymbol{\Sigma}_s) \operatorname{tr}(\boldsymbol{\Sigma}_w) \right),$$

$$\mathbf{Bias}(f_{\text{w2s}}) \leqslant \alpha_w \varrho_w + \alpha_{\text{w2s}} \varrho_s.$$

*In particular, when taking*

$$\alpha_w = \left( \frac{\sigma^2 \operatorname{tr}\left( \boldsymbol{\Sigma}_s \boldsymbol{\Sigma}_w \right)}{4nN} \frac{\varrho_s}{\varrho_w^2} \right)^{1/3}, \quad \alpha_{\text{w2s}} = \left( \frac{\sigma^2 \operatorname{tr}\left( \boldsymbol{\Sigma}_s \boldsymbol{\Sigma}_w \right)}{4nN} \frac{\varrho_w}{\varrho_s^2} \right)^{1/3},$$

*the excess risk of W2S FT is upper bounded by*

$$\mathbf{ER}(f_{\text{w2s}}) \leqslant 3 \left( \frac{\sigma^2}{4nN} \varrho_s \varrho_w \left( \left( 1 + \frac{1}{N} \right) \operatorname{tr}\left( \boldsymbol{\Sigma}_s \boldsymbol{\Sigma}_w \right) + \frac{1}{N} \operatorname{tr}(\boldsymbol{\Sigma}_s) \operatorname{tr}(\boldsymbol{\Sigma}_w) \right) \right)^{1/3}.$$

Notice that for a large enough $N \geqslant \frac{\operatorname{tr}(\boldsymbol{\Sigma}_s) \operatorname{tr}(\boldsymbol{\Sigma}_w)}{\operatorname{tr}(\boldsymbol{\Sigma}_s \boldsymbol{\Sigma}_w)}$, Theorem C.4 implies that

$$\mathbf{ER}(f_{\text{w2s}}) \leqslant 3 \left( \frac{3\sigma^2}{4nN} \varrho_s \varrho_w \operatorname{tr}\left( \boldsymbol{\Sigma}_s \boldsymbol{\Sigma}_w \right) \right)^{1/3}.$$

Theorem C.4 conveys a similar high-level intuition as in Theorem 3.1 regarding the effect of the weak-strong similarity on the generalization error of W2S FT. In particular, the larger discrepancy between $\phi_s$ and $\phi_w$ (corresponding to the smaller $\operatorname{tr}\left( \boldsymbol{\Sigma}_s \boldsymbol{\Sigma}_w \right)$) leads to lower variance and therefore better W2S generalization.

Meanwhile, a key difference in W2S between the ridge and ridgeless settings (Theorem C.4 versus Theorem 3.1) is that the FT approximation errors in Theorem C.4, reflected by $\varrho_s = \|\boldsymbol{\Sigma}_s^{-1/2} \boldsymbol{\Sigma}_*^{1/2} \boldsymbol{\theta}_*\|_2^2$ and $\varrho_w = \|\boldsymbol{\Sigma}_w^{-1/2} \boldsymbol{\Sigma}_*^{1/2} \boldsymbol{\theta}_*\|_2^2$, can be compensated by larger sample sizes $n, N$ and directly affect the sample complexity:

$$nN \asymp \sigma^2 \operatorname{tr}\left( \boldsymbol{\Sigma}_s \boldsymbol{\Sigma}_w \right) \varrho_s \varrho_w.$$

Such difference is a result of optimizing the regularization hyperparameters $\alpha_w, \alpha_{\text{w2s}}$ in ridge regression that control the variance-bias tradeoff.

*Proof of Theorem C.4.* We first formalize some useful facts on the features and labels as in (7). In particular, the sub-gaussian assumption in Assumption C.1(ii) implies that for each $\mathbf{x} \sim \mathcal{D}$, the corresponding strong/weak feature $\phi_s(\mathbf{x}), \phi_w(\mathbf{x}) \in \mathbb{R}^d$,

and the ground truth $f_*(\mathbf{x}) \in \mathbb{R}$ are simultaneously characterized by an independent subgaussian random vector $\boldsymbol{\gamma} \in \mathbb{R}^d$ with $\mathbb{E}[\boldsymbol{\gamma}] = \mathbf{0}_d$ and $\mathbb{E}[\boldsymbol{\gamma}\boldsymbol{\gamma}^\top] = \mathbf{I}_d$, *i.e.*,

$$\phi_s(\mathbf{x}) = \boldsymbol{\Sigma}_s^{1/2}\boldsymbol{\gamma}, \quad \phi_w(\mathbf{x}) = \boldsymbol{\Sigma}_w^{1/2}\boldsymbol{\gamma}, \quad f_*(\mathbf{x}) = \phi_*(\mathbf{x})^\top\boldsymbol{\theta}_* = \boldsymbol{\gamma}^\top\boldsymbol{\Sigma}_*^{1/2}\boldsymbol{\theta}_*.$$

Then, for $\mathcal{S}$ and $\widetilde{\mathcal{S}}$, there exist independent random matrices $\boldsymbol{\Gamma} = [\boldsymbol{\gamma}_1, \ldots, \boldsymbol{\gamma}_N]^\top \in \mathbb{R}^{N \times d}$ and $\widetilde{\boldsymbol{\Gamma}} = [\widetilde{\boldsymbol{\gamma}}_1, \ldots, \widetilde{\boldsymbol{\gamma}}_n]^\top \in \mathbb{R}^{n \times d}$ consisting of *i.i.d.* zero-mean isotropic rows such that

$$
\begin{aligned}
\boldsymbol{\Phi}_s &= \boldsymbol{\Gamma}\boldsymbol{\Sigma}_s^{1/2} = \boldsymbol{\Gamma}_s\boldsymbol{\Lambda}_s^{1/2}\mathbf{V}_s^\top, \\
\boldsymbol{\Phi}_w &= \boldsymbol{\Gamma}\boldsymbol{\Sigma}_w^{1/2} = \boldsymbol{\Gamma}_w\boldsymbol{\Lambda}_w^{1/2}\mathbf{V}_w^\top, \\
\mathbf{y} &= \mathbf{f}_* + \mathbf{z}, \quad \mathbf{f}_* = \boldsymbol{\Gamma}\boldsymbol{\Sigma}_*^{1/2}\boldsymbol{\theta}_*, \quad \mathbf{z} \sim \mathcal{N}(\mathbf{0}_N, \sigma^2\mathbf{I}_N), \\
\widetilde{\boldsymbol{\Phi}}_w &= \widetilde{\boldsymbol{\Gamma}}\boldsymbol{\Sigma}_w^{1/2} = \widetilde{\boldsymbol{\Gamma}}_w\boldsymbol{\Lambda}_w^{1/2}\mathbf{V}_w^\top, \\
\widetilde{\mathbf{y}} &= \widetilde{\mathbf{f}}_* + \widetilde{\mathbf{z}}, \quad \widetilde{\mathbf{f}}_* = \widetilde{\boldsymbol{\Gamma}}\boldsymbol{\Sigma}_*^{1/2}\boldsymbol{\theta}_*, \quad \widetilde{\mathbf{z}} \sim \mathcal{N}(\mathbf{0}_n, \sigma^2\mathbf{I}_n),
\end{aligned}
\tag{16}
$$

where $\boldsymbol{\Gamma}_s = \boldsymbol{\Gamma}\mathbf{V}_s$, $\boldsymbol{\Gamma}_w = \boldsymbol{\Gamma}\mathbf{V}_w$, and $\widetilde{\boldsymbol{\Gamma}}_w = \widetilde{\boldsymbol{\Gamma}}\mathbf{V}_w$.

**Variance-bias decomposition.** With $\alpha_w > 0$, (14) yields a weak teacher model $f_w(\mathbf{x}) = \phi_w(\mathbf{x})^\top\boldsymbol{\theta}_w$ with

$$\boldsymbol{\theta}_w = \left(\widetilde{\boldsymbol{\Phi}}_w^\top\widetilde{\boldsymbol{\Phi}}_w + \alpha_w n\mathbf{I}_d\right)^{-1}\widetilde{\boldsymbol{\Phi}}_w^\top\left(\widetilde{\mathbf{f}}_8 + \widetilde{\mathbf{z}}\right).$$

Then, the W2S model $f_{\mathrm{w2s}}(\mathbf{x}) = \phi_s(\mathbf{x})^\top\boldsymbol{\theta}_{\mathrm{w2s}}$ is given by (15) with $\alpha_{\mathrm{w2s}} > 0$:

$$
\begin{aligned}
\boldsymbol{\theta}_{\mathrm{w2s}} &= \left(\boldsymbol{\Phi}_s^\top\boldsymbol{\Phi}_s + \alpha_{\mathrm{w2s}}N\mathbf{I}_d\right)^{-1}\boldsymbol{\Phi}_s^\top\boldsymbol{\Phi}_w\boldsymbol{\theta}_w \\
&= \left(\boldsymbol{\Phi}_s^\top\boldsymbol{\Phi}_s + \alpha_{\mathrm{w2s}}N\mathbf{I}_d\right)^{-1}\boldsymbol{\Phi}_s^\top\boldsymbol{\Phi}_w\left(\widetilde{\boldsymbol{\Phi}}_w^\top\widetilde{\boldsymbol{\Phi}}_w + \alpha_w n\mathbf{I}_d\right)^{-1}\widetilde{\boldsymbol{\Phi}}_w^\top\left(\widetilde{\mathbf{f}}_* + \widetilde{\mathbf{z}}\right),
\end{aligned}
$$

which implies

$$\mathbb{E}_{\mathcal{S}_x, \widetilde{\mathcal{S}}}[\boldsymbol{\theta}_{\mathrm{w2s}}] = \left(\boldsymbol{\Phi}_s^\top\boldsymbol{\Phi}_s + \alpha_{\mathrm{w2s}}N\mathbf{I}_d\right)^{-1}\boldsymbol{\Phi}_s^\top\boldsymbol{\Phi}_w\left(\widetilde{\boldsymbol{\Phi}}_w^\top\widetilde{\boldsymbol{\Phi}}_w + \alpha_w n\mathbf{I}_d\right)^{-1}\widetilde{\boldsymbol{\Phi}}_w^\top\widetilde{\mathbf{f}}_*.$$

Then, we can concretize the variance and bias terms as:

$$
\begin{aligned}
\mathbf{Var}(f_{\mathrm{w2s}}) &= \mathbb{E}_{\mathcal{S}_x, \widetilde{\mathcal{S}}}\left[\frac{1}{N}\left\|\boldsymbol{\Phi}_s\left(\boldsymbol{\theta}_{\mathrm{w2s}} - \mathbb{E}_{\mathcal{S}_x, \widetilde{\mathcal{S}}}[\boldsymbol{\theta}_{\mathrm{w2s}}]\right)\right\|_2^2\right] \\
&= \mathbb{E}_{\mathcal{S}_x, \widetilde{\mathcal{S}}}\left[\frac{1}{N}\left\|\boldsymbol{\Phi}_s\left(\boldsymbol{\Phi}_s^\top\boldsymbol{\Phi}_s + \alpha_{\mathrm{w2s}}N\mathbf{I}_d\right)^{-1}\boldsymbol{\Phi}_s^\top\boldsymbol{\Phi}_w\left(\widetilde{\boldsymbol{\Phi}}_w^\top\widetilde{\boldsymbol{\Phi}}_w + \alpha_w n\mathbf{I}_d\right)^{-1}\widetilde{\boldsymbol{\Phi}}_w^\top\widetilde{\mathbf{z}}\right\|_2^2\right],
\end{aligned}
\tag{17}
$$

and

$$
\begin{aligned}
\mathbf{Bias}(f_{\mathrm{w2s}}) &= \mathbb{E}_{\mathcal{S}_x, \widetilde{\mathcal{S}}}\left[\frac{1}{N}\left\|\boldsymbol{\Phi}_s\mathbb{E}_{\mathcal{S}_x, \widetilde{\mathcal{S}}}[\boldsymbol{\theta}_{\mathrm{w2s}}] - \mathbf{f}_*\right\|_2^2\right] \\
&= \mathbb{E}_{\mathcal{S}_x, \widetilde{\mathcal{S}}}\left[\frac{1}{N}\left\|\boldsymbol{\Phi}_s\left(\boldsymbol{\Phi}_s^\top\boldsymbol{\Phi}_s + \alpha_{\mathrm{w2s}}N\mathbf{I}_d\right)^{-1}\boldsymbol{\Phi}_s^\top\boldsymbol{\Phi}_w\left(\widetilde{\boldsymbol{\Phi}}_w^\top\widetilde{\boldsymbol{\Phi}}_w + \alpha_w n\mathbf{I}_d\right)^{-1}\widetilde{\boldsymbol{\Phi}}_w^\top\widetilde{\mathbf{f}}_* - \mathbf{f}_*\right\|_2^2\right].
\end{aligned}
\tag{18}
$$

Now, we are ready to upper-bound the variance and bias terms separately.

**Variance.** Denote $\boldsymbol{\zeta} = \boldsymbol{\Lambda}_w^{1/2}\mathbf{V}_w^\top\left(\widetilde{\boldsymbol{\Phi}}_w^\top\widetilde{\boldsymbol{\Phi}}_w + \alpha_w n\mathbf{I}_d\right)^{-1}\widetilde{\boldsymbol{\Phi}}_w^\top\widetilde{\mathbf{z}} \in \mathbb{R}^d$, whose randomness comes from $\widetilde{\mathcal{S}}$ only, independent of $\mathcal{S}_x$. Then, the variance term (17) can be expressed as

$$
\begin{aligned}
\mathbf{Var}(f_{\mathrm{w2s}}) &= \mathbb{E}_{\mathcal{S}_x, \widetilde{\mathcal{S}}}\left[\frac{1}{N}\left\|\boldsymbol{\Phi}_s\left(\boldsymbol{\Phi}_s^\top\boldsymbol{\Phi}_s + \alpha_{\mathrm{w2s}}N\mathbf{I}_d\right)^{-1}\boldsymbol{\Phi}_s^\top\boldsymbol{\Phi}_w\boldsymbol{\zeta}\right\|_2^2\right] \\
&= \frac{1}{N}\mathrm{tr}\left(\mathbb{E}_{\mathcal{S}_x}\left[\boldsymbol{\Gamma}_w^\top\boldsymbol{\Phi}_s\left(\boldsymbol{\Phi}_s^\top\boldsymbol{\Phi}_s + \alpha_{\mathrm{w2s}}N\mathbf{I}_d\right)^{-1}\boldsymbol{\Phi}_s^\top\boldsymbol{\Phi}_s\left(\boldsymbol{\Phi}_s^\top\boldsymbol{\Phi}_s + \alpha_{\mathrm{w2s}}N\mathbf{I}_d\right)^{-1}\boldsymbol{\Phi}_s^\top\boldsymbol{\Gamma}_w\right]\mathbb{E}_{\widetilde{\mathcal{S}}}\left[\boldsymbol{\zeta}\boldsymbol{\zeta}^\top\right]\right).
\end{aligned}
$$

Notice that

$$\left(\mathbf{\Phi}_s^\top \mathbf{\Phi}_s + \alpha_{\text{w2s}} N \mathbf{I}_d\right)^{-1} \mathbf{\Phi}_s^\top \mathbf{\Phi}_s \left(\mathbf{\Phi}_s^\top \mathbf{\Phi}_s + \alpha_{\text{w2s}} N \mathbf{I}_d\right)^{-1}$$
$$= \frac{1}{N} \left(\frac{1}{N} \mathbf{\Phi}_s^\top \mathbf{\Phi}_s + \alpha_{\text{w2s}} \mathbf{I}_d\right)^{-1} \left(\frac{1}{N} \mathbf{\Phi}_s^\top \mathbf{\Phi}_s\right) \left(\frac{1}{N} \mathbf{\Phi}_s^\top \mathbf{\Phi}_s + \alpha_{\text{w2s}} \mathbf{I}_d\right)^{-1} \preceq \frac{1}{2\alpha_{\text{w2s}} N^2} \mathbf{I}_d.$$

Therefore, we have

$$\mathbf{Var}(f_{\text{w2s}}) \leqslant \frac{1}{2\alpha_{\text{w2s}} N} \operatorname{tr}\left(\frac{1}{N^2} \mathbb{E}_{\mathcal{S}_x}\left[\mathbf{\Gamma}^\top \mathbf{\Phi}_s \mathbf{\Phi}_s^\top \mathbf{\Gamma}\right] \mathbb{E}_{\widetilde{\mathcal{S}}}\left[\mathbf{V}_w \boldsymbol{\zeta} \boldsymbol{\zeta}^\top \mathbf{V}_w^\top\right]\right)$$
$$= \frac{1}{2\alpha_{\text{w2s}} N} \operatorname{tr}\left(\frac{1}{N^2} \mathbb{E}_{\mathcal{S}_x}\left[\mathbf{\Gamma}^\top \mathbf{\Gamma} \mathbf{\Sigma}_s \mathbf{\Gamma}^\top \mathbf{\Gamma}\right] \mathbb{E}_{\widetilde{\mathcal{S}}}\left[\mathbf{V}_w \boldsymbol{\zeta} \boldsymbol{\zeta}^\top \mathbf{V}_w^\top\right]\right).$$

Since Assumption C.1 (ii) implies that

$$\frac{1}{N^2} \mathbb{E}_{\mathcal{S}_x}\left[\mathbf{\Gamma}^\top \mathbf{\Gamma} \mathbf{\Sigma}_s \mathbf{\Gamma}^\top \mathbf{\Gamma}\right] = \left(1 + \frac{1}{N}\right) \mathbf{\Sigma}_s + \frac{1}{N} \operatorname{tr}(\mathbf{\Sigma}_s) \mathbf{I}_d,$$

we then have

$$\mathbf{Var}(f_{\text{w2s}}) \leqslant \frac{1}{2\alpha_{\text{w2s}} N} \operatorname{tr}\left(\left(\left(1 + \frac{1}{N}\right) \mathbf{\Sigma}_s + \frac{1}{N} \operatorname{tr}(\mathbf{\Sigma}_s) \mathbf{I}_d\right) \mathbb{E}_{\widetilde{\mathcal{S}}}\left[\mathbf{V}_w \boldsymbol{\zeta} \boldsymbol{\zeta}^\top \mathbf{V}_w^\top\right]\right).$$

Meanwhile, we observe that

$$\mathbb{E}_{\widetilde{\mathcal{S}}}\left[\mathbf{V}_w \boldsymbol{\zeta} \boldsymbol{\zeta}^\top \mathbf{V}_w^\top\right] = \mathbb{E}_{\widetilde{\mathcal{S}}}\left[\mathbf{\Sigma}_w^{1/2} \left(\widetilde{\mathbf{\Phi}}_w^\top \widetilde{\mathbf{\Phi}}_w + \alpha_w n \mathbf{I}_d\right)^{-1} \widetilde{\mathbf{\Phi}}_w^\top \widetilde{\mathbf{z}} \widetilde{\mathbf{z}}^\top \widetilde{\mathbf{\Phi}}_w \left(\widetilde{\mathbf{\Phi}}_w^\top \widetilde{\mathbf{\Phi}}_w + \alpha_w n \mathbf{I}_d\right)^{-1} \mathbf{\Sigma}_w^{1/2}\right]$$
$$= \sigma^2 \mathbb{E}_{\widetilde{\mathcal{S}}}\left[\mathbf{\Sigma}_w^{1/2} \left(\widetilde{\mathbf{\Phi}}_w^\top \widetilde{\mathbf{\Phi}}_w + \alpha_w n \mathbf{I}_d\right)^{-1} \widetilde{\mathbf{\Phi}}_w^\top \widetilde{\mathbf{\Phi}}_w \left(\widetilde{\mathbf{\Phi}}_w^\top \widetilde{\mathbf{\Phi}}_w + \alpha_w n \mathbf{I}_d\right)^{-1} \mathbf{\Sigma}_w^{1/2}\right],$$

where

$$\left(\widetilde{\mathbf{\Phi}}_w^\top \widetilde{\mathbf{\Phi}}_w + \alpha_w n \mathbf{I}_d\right)^{-1} \widetilde{\mathbf{\Phi}}_w^\top \widetilde{\mathbf{\Phi}}_w \left(\widetilde{\mathbf{\Phi}}_w^\top \widetilde{\mathbf{\Phi}}_w + \alpha_w n \mathbf{I}_d\right)^{-1} \preceq \frac{1}{2\alpha_w n} \mathbf{I}_d.$$

Therefore, we have

$$\mathbb{E}_{\widetilde{\mathcal{S}}}\left[\mathbf{V}_w \boldsymbol{\zeta} \boldsymbol{\zeta}^\top \mathbf{V}_w^\top\right] \preceq \sigma^2 \mathbb{E}_{\widetilde{\mathcal{S}}}\left[\mathbf{\Sigma}_w^{1/2} \left(\frac{1}{2\alpha_w n} \mathbf{I}_d\right) \mathbf{\Sigma}_w^{1/2}\right] = \frac{\sigma^2}{2\alpha_w n} \mathbf{\Sigma}_w.$$

Overall, the variance of $f_{\text{w2s}}$ can be upper bounded as

$$\mathbf{Var}(f_{\text{w2s}}) \leqslant \frac{\sigma^2}{4(\alpha_w n)(\alpha_{\text{w2s}} N)} \left(\left(1 + \frac{1}{N}\right) \operatorname{tr}\left(\mathbf{\Sigma}_s \mathbf{\Sigma}_w\right) + \frac{1}{N} \operatorname{tr}(\mathbf{\Sigma}_s) \operatorname{tr}(\mathbf{\Sigma}_w)\right). \tag{19}$$

**Bias.** Let $\boldsymbol{\xi} = \mathbf{\Sigma}_w^{1/2} \left(\widetilde{\mathbf{\Phi}}_w^\top \widetilde{\mathbf{\Phi}}_w + \alpha_w n \mathbf{I}_d\right)^{-1} \widetilde{\mathbf{\Phi}}_w^\top \widetilde{\mathbf{f}}_* \in \mathbb{R}^d$, whose randomness comes from $\widetilde{\mathcal{S}}$ only, independent of $\mathcal{S}_x$. Recall from (18), the bias term (18) can be decomposed as

$$\mathbf{Bias}(f_{\text{w2s}}) = \mathbb{E}_{\mathcal{S}_x, \widetilde{\mathcal{S}}}\left[\frac{1}{N} \left\|\mathbf{\Phi}_s \left(\mathbf{\Phi}_s^\top \mathbf{\Phi}_s + \alpha_{\text{w2s}} N \mathbf{I}_d\right)^{-1} \mathbf{\Phi}_s^\top \mathbf{\Phi}_w \left(\widetilde{\mathbf{\Phi}}_w^\top \widetilde{\mathbf{\Phi}}_w + \alpha_w n \mathbf{I}_d\right)^{-1} \widetilde{\mathbf{\Phi}}_w^\top \widetilde{\mathbf{f}}_* - \mathbf{f}_*\right\|_2^2\right]$$
$$= \mathbb{E}_{\mathcal{S}_x, \widetilde{\mathcal{S}}}\left[\frac{1}{N} \left(\left\|\mathbf{\Phi}_s \left(\mathbf{\Phi}_s^\top \mathbf{\Phi}_s + \alpha_{\text{w2s}} N \mathbf{I}_d\right)^{-1} \mathbf{\Phi}_s^\top \mathbf{\Gamma} \boldsymbol{\xi} - \mathbf{\Phi}_s \mathbf{\Phi}_s^\dagger \mathbf{f}_*\right\|_2^2 + \left\|\left(\mathbf{I}_N - \mathbf{\Phi}_s \mathbf{\Phi}_s^\dagger\right) \mathbf{f}_*\right\|_2^2\right)\right],$$

where by Lemma B.1 and (13)

$$\mathbb{E}_{\mathcal{S}_x}\left[\frac{1}{N} \left\|\left(\mathbf{I}_N - \mathbf{\Phi}_s \mathbf{\Phi}_s^\dagger\right) \mathbf{f}_*\right\|_2^2\right] = \frac{\rho_s(N)}{N} \leqslant \rho_s = 0.$$

Therefore, with $\boldsymbol{\xi} = \boldsymbol{\Sigma}_w^{1/2} \left( \widetilde{\boldsymbol{\Phi}}_w^\top \widetilde{\boldsymbol{\Phi}}_w + \alpha_w n \mathbf{I}_d \right)^{-1} \widetilde{\boldsymbol{\Phi}}_w^\top \widetilde{\mathbf{f}}_*$, we have

$$\mathbf{Bias}(f_{\mathrm{w2s}}) = \mathbb{E}_{\mathcal{S}_x, \widetilde{\mathcal{S}}} \left[ \frac{1}{N} \left\| \boldsymbol{\Phi}_s \left( \boldsymbol{\Phi}_s^\top \boldsymbol{\Phi}_s + \alpha_{\mathrm{w2s}} N \mathbf{I}_d \right)^{-1} \boldsymbol{\Phi}_s^\top \boldsymbol{\Gamma} \boldsymbol{\xi} - \boldsymbol{\Phi}_s \boldsymbol{\Phi}_s^\dagger \mathbf{f}_* \right\|_2^2 \right].$$

Recall that $\mathbf{f}_* = \boldsymbol{\Gamma} \boldsymbol{\Sigma}_*^{1/2} \boldsymbol{\theta}_*$ and $\boldsymbol{\Phi}_s = \boldsymbol{\Gamma} \boldsymbol{\Sigma}_s^{1/2} = \boldsymbol{\Gamma}_s \boldsymbol{\Lambda}_s^{1/2} \mathbf{V}_s^\top$. Then, we can express the bias term as

$$\begin{aligned}
\mathbf{Bias}(f_{\mathrm{w2s}}) =& \mathbb{E}_{\mathcal{S}_x, \widetilde{\mathcal{S}}} \left[ \frac{1}{N} \left\| \boldsymbol{\Gamma} \left( \boldsymbol{\Gamma}^\top \boldsymbol{\Gamma} + \alpha_{\mathrm{w2s}} N \boldsymbol{\Sigma}_s^{-1} \right)^{-1} \boldsymbol{\Gamma}^\top \boldsymbol{\Gamma} \boldsymbol{\xi} - \boldsymbol{\Gamma} \boldsymbol{\Gamma}^\dagger \mathbf{f}_* \right\|_2^2 \right] \\
=& \mathbb{E}_{\mathcal{S}_x, \widetilde{\mathcal{S}}} \left[ \frac{1}{N} \left\| \boldsymbol{\Gamma} \boldsymbol{\Sigma}_*^{1/2} \boldsymbol{\theta}_* - \boldsymbol{\Gamma} \left( \boldsymbol{\Gamma}^\top \boldsymbol{\Gamma} + \alpha_{\mathrm{w2s}} N \boldsymbol{\Sigma}_s^{-1} \right)^{-1} \boldsymbol{\Gamma}^\top \boldsymbol{\Gamma} \boldsymbol{\xi} \right\|_2^2 \right] \\
=& \mathbb{E}_{\mathcal{S}_x, \widetilde{\mathcal{S}}} \left[ \frac{1}{N} \left\| \boldsymbol{\Gamma} \left( \boldsymbol{\Sigma}_*^{1/2} \boldsymbol{\theta}_* - \boldsymbol{\xi} \right) + \boldsymbol{\Gamma} \left( \mathbf{I}_d - \left( \boldsymbol{\Gamma}^\top \boldsymbol{\Gamma} + \alpha_{\mathrm{w2s}} N \boldsymbol{\Sigma}_s^{-1} \right)^{-1} \boldsymbol{\Gamma}^\top \boldsymbol{\Gamma} \right) \boldsymbol{\xi} \right\|_2^2 \right]
\end{aligned}$$

By the Woodbury matrix identity, we have

$$\mathbf{I}_d - \left( \boldsymbol{\Gamma}^\top \boldsymbol{\Gamma} + \alpha_{\mathrm{w2s}} N \boldsymbol{\Sigma}_s^{-1} \right)^{-1} \boldsymbol{\Gamma}^\top \boldsymbol{\Gamma} = \left( \mathbf{I}_d + \frac{1}{\alpha_{\mathrm{w2s}} N} \boldsymbol{\Sigma}_s \boldsymbol{\Gamma}^\top \boldsymbol{\Gamma} \right)^{-1}. \tag{20}$$

Therefore, we have

$$\mathbf{Bias}(f_{\mathrm{w2s}}) = \mathbb{E}_{\mathcal{S}_x, \widetilde{\mathcal{S}}} \left[ \frac{1}{N} \left\| \underbrace{\boldsymbol{\Gamma} \left( \boldsymbol{\Sigma}_*^{1/2} \boldsymbol{\theta}_* - \boldsymbol{\xi} \right)}_{\text{Term A}} + \underbrace{\boldsymbol{\Gamma} \left( \mathbf{I}_d + \frac{1}{\alpha_{\mathrm{w2s}} N} \boldsymbol{\Sigma}_s \boldsymbol{\Gamma}^\top \boldsymbol{\Gamma} \right)^{-1} \boldsymbol{\xi}}_{\text{Term B}} \right\|_2^2 \right]. \tag{21}$$

For Term A, notice that $\boldsymbol{\xi} = \boldsymbol{\Sigma}_w^{1/2} \left( \widetilde{\boldsymbol{\Phi}}_w^\top \widetilde{\boldsymbol{\Phi}}_w + \alpha_w n \mathbf{I}_d \right)^{-1} \widetilde{\boldsymbol{\Phi}}_w^\top \widetilde{\mathbf{f}}_*$ implies

$$\begin{aligned}
\boldsymbol{\Sigma}_*^{1/2} \boldsymbol{\theta}_* - \boldsymbol{\xi} =& \boldsymbol{\Sigma}_*^{1/2} \boldsymbol{\theta}_* - \boldsymbol{\Sigma}_w^{1/2} \left( \widetilde{\boldsymbol{\Phi}}_w^\top \widetilde{\boldsymbol{\Phi}}_w + \alpha_w n \mathbf{I}_d \right)^{-1} \widetilde{\boldsymbol{\Phi}}_w^\top \widetilde{\mathbf{f}}_* \\
=& \boldsymbol{\Sigma}_*^{1/2} \boldsymbol{\theta}_* - \left( \widetilde{\boldsymbol{\Gamma}}^\top \widetilde{\boldsymbol{\Gamma}} + \alpha_w n \boldsymbol{\Sigma}_w^{-1} \right)^{-1} \widetilde{\boldsymbol{\Gamma}}^\top \widetilde{\boldsymbol{\Gamma}} \boldsymbol{\Sigma}_*^{1/2} \boldsymbol{\theta}_* \\
=& \left( \mathbf{I}_d - \left( \widetilde{\boldsymbol{\Gamma}}^\top \widetilde{\boldsymbol{\Gamma}} + \alpha_w n \boldsymbol{\Sigma}_w^{-1} \right)^{-1} \widetilde{\boldsymbol{\Gamma}}^\top \widetilde{\boldsymbol{\Gamma}} \right) \boldsymbol{\Sigma}_*^{1/2} \boldsymbol{\theta}_* \\
=& \left( \mathbf{I}_d + \frac{1}{\alpha_w n} \boldsymbol{\Sigma}_w \widetilde{\boldsymbol{\Gamma}}^\top \widetilde{\boldsymbol{\Gamma}} \right)^{-1} \boldsymbol{\Sigma}_*^{1/2} \boldsymbol{\theta}_*,
\end{aligned}$$

where the last equality follows from Woodbury matrix identity as in (20). Therefore,

$$\begin{aligned}
\mathbb{E}_{\mathcal{S}_x, \widetilde{\mathcal{S}}} \left[ \frac{1}{N} \left\| \boldsymbol{\Gamma} \left( \boldsymbol{\Sigma}_*^{1/2} \boldsymbol{\theta}_* - \boldsymbol{\xi} \right) \right\|_2^2 \right] =& \mathbb{E}_{\widetilde{\mathcal{S}}} \left[ \frac{1}{n} \left\| \widetilde{\boldsymbol{\Gamma}} \left( \boldsymbol{\Sigma}_*^{1/2} \boldsymbol{\theta}_* - \boldsymbol{\xi} \right) \right\|_2^2 \right] \\
=& \mathbb{E}_{\widetilde{\mathcal{S}}} \left[ \frac{1}{n} \left\| \widetilde{\boldsymbol{\Gamma}} \left( \mathbf{I}_d + \frac{1}{\alpha_w n} \boldsymbol{\Sigma}_w \widetilde{\boldsymbol{\Gamma}}^\top \widetilde{\boldsymbol{\Gamma}} \right)^{-1} \boldsymbol{\Sigma}_*^{1/2} \boldsymbol{\theta}_* \right\|_2^2 \right].
\end{aligned}$$

Since

$$\left( \mathbf{I}_d + \frac{1}{\alpha_w n} \boldsymbol{\Sigma}_w \widetilde{\boldsymbol{\Gamma}}^\top \widetilde{\boldsymbol{\Gamma}} \right)^{-1} \widetilde{\boldsymbol{\Gamma}}^\top \widetilde{\boldsymbol{\Gamma}} \left( \mathbf{I}_d + \frac{1}{\alpha_w n} \boldsymbol{\Sigma}_w \widetilde{\boldsymbol{\Gamma}}^\top \widetilde{\boldsymbol{\Gamma}} \right)^{-1} \preceq \frac{\alpha_w n}{2} \boldsymbol{\Sigma}_w^{-1},$$

we have

$$\begin{aligned}
\mathbb{E}_{\mathcal{S}_x, \widetilde{\mathcal{S}}} \left[ \frac{1}{N} \left\| \boldsymbol{\Gamma} \left( \boldsymbol{\Sigma}_*^{1/2} \boldsymbol{\theta}_* - \boldsymbol{\xi} \right) \right\|_2^2 \right] \leqslant& \frac{1}{n} \mathrm{tr} \left( \frac{\alpha_w n}{2} \boldsymbol{\Sigma}_w^{-1} \boldsymbol{\Sigma}_*^{1/2} \boldsymbol{\theta}_* \boldsymbol{\theta}_*^\top \boldsymbol{\Sigma}_*^{1/2} \right) \\
=& \frac{\alpha_w}{2} \left\| \boldsymbol{\Sigma}_w^{-1/2} \boldsymbol{\Sigma}_*^{1/2} \boldsymbol{\theta}_* \right\|_2^2.
\end{aligned} \tag{22}$$

For Term B, leveraging Woodbury matrix identity as in (20), we notice that

$$
\mathbb{E}_{\mathcal{S}_x,\widetilde{\mathcal{S}}}\left[\frac{1}{N}\left\|\mathbf{\Gamma}\left(\mathbf{I}_d+\frac{1}{\alpha_{\mathrm{w2s}}N}\mathbf{\Sigma}_s\mathbf{\Gamma}^\top\mathbf{\Gamma}\right)^{-1}\boldsymbol{\xi}\right\|_2^2\right] \leqslant \mathbb{E}_{\mathcal{S}_x,\widetilde{\mathcal{S}}}\left[\frac{1}{N}\operatorname{tr}\left(\frac{\alpha_{\mathrm{w2s}}N}{2}\mathbf{\Sigma}_s^{-1}\boldsymbol{\xi}\boldsymbol{\xi}^\top\right)\right]
$$

$$
=\frac{\alpha_{\mathrm{w2s}}}{2}\mathbb{E}_{\mathcal{S}_x,\widetilde{\mathcal{S}}}\left[\left\|\mathbf{\Sigma}_s^{-1/2}\mathbf{\Sigma}_w^{1/2}\left(\widetilde{\mathbf{\Phi}}_w^\top\widetilde{\mathbf{\Phi}}_w+\alpha_w n\mathbf{I}_d\right)^{-1}\widetilde{\mathbf{\Phi}}_w^\top\widetilde{\mathbf{f}}_*\right\|_2^2\right]
$$

$$
=\frac{\alpha_{\mathrm{w2s}}}{2}\mathbb{E}_{\mathcal{S}_x,\widetilde{\mathcal{S}}}\left[\left\|\mathbf{\Sigma}_s^{-1/2}\left(\widetilde{\mathbf{\Gamma}}^\top\widetilde{\mathbf{\Gamma}}+\alpha_w n\mathbf{\Sigma}_w^{-1}\right)^{-1}\widetilde{\mathbf{\Gamma}}^\top\widetilde{\mathbf{\Gamma}}\mathbf{\Sigma}_*^{1/2}\boldsymbol{\theta}_*\right\|_2^2\right]
$$

Since $\left(\widetilde{\mathbf{\Gamma}}^\top\widetilde{\mathbf{\Gamma}}+\alpha_w n\mathbf{\Sigma}_w^{-1}\right)^{-1}\widetilde{\mathbf{\Gamma}}^\top\widetilde{\mathbf{\Gamma}}\preceq\mathbf{I}_d$, we know that

$$
\mathbb{E}_{\mathcal{S}_x,\widetilde{\mathcal{S}}}\left[\frac{1}{N}\left\|\mathbf{\Gamma}\left(\mathbf{I}_d+\frac{1}{\alpha_{\mathrm{w2s}}N}\mathbf{\Sigma}_s\mathbf{\Gamma}^\top\mathbf{\Gamma}\right)^{-1}\boldsymbol{\xi}\right\|_2^2\right] \leqslant \frac{\alpha_{\mathrm{w2s}}}{2}\left\|\mathbf{\Sigma}_s^{-1/2}\mathbf{\Sigma}_*^{1/2}\boldsymbol{\theta}_*\right\|_2^2. \tag{23}
$$

Combining (21), (22), and (23), we can upper bound the bias term as

$$
\mathbf{Bias}(f_{\mathrm{w2s}})=\mathbb{E}_{\mathcal{S}_x,\widetilde{\mathcal{S}}}\left[\frac{1}{N}\left\|\underbrace{\mathbf{\Gamma}\left(\mathbf{\Sigma}_*^{1/2}\boldsymbol{\theta}_*-\boldsymbol{\xi}\right)}_{\text{Term A}}+\underbrace{\mathbf{\Gamma}\left(\mathbf{I}_d+\frac{1}{\alpha_{\mathrm{w2s}}N}\mathbf{\Sigma}_s\mathbf{\Gamma}^\top\mathbf{\Gamma}\right)^{-1}\boldsymbol{\xi}}_{\text{Term B}}\right\|_2^2\right]
$$

$$
\leqslant 2\mathbb{E}_{\mathcal{S}_x,\widetilde{\mathcal{S}}}\left[\frac{1}{N}\left\|\mathbf{\Gamma}\left(\mathbf{\Sigma}_*^{1/2}\boldsymbol{\theta}_*-\boldsymbol{\xi}\right)\right\|_2^2\right]+2\mathbb{E}_{\mathcal{S}_x,\widetilde{\mathcal{S}}}\left[\frac{1}{N}\left\|\mathbf{\Gamma}\left(\mathbf{I}_d+\frac{1}{\alpha_{\mathrm{w2s}}N}\mathbf{\Sigma}_s\mathbf{\Gamma}^\top\mathbf{\Gamma}\right)^{-1}\boldsymbol{\xi}\right\|_2^2\right] \tag{24}
$$

$$
\leqslant \alpha_w\left\|\mathbf{\Sigma}_w^{-1/2}\mathbf{\Sigma}_*^{1/2}\boldsymbol{\theta}_*\right\|_2^2+\alpha_{\mathrm{w2s}}\left\|\mathbf{\Sigma}_s^{-1/2}\mathbf{\Sigma}_*^{1/2}\boldsymbol{\theta}_*\right\|_2^2.
$$

**Variance-bias tradeoff.** Recall $\varrho_s=\|\mathbf{\Sigma}_s^{-1/2}\mathbf{\Sigma}_*^{1/2}\boldsymbol{\theta}_*\|_2^2$ and $\varrho_w=\|\mathbf{\Sigma}_w^{-1/2}\mathbf{\Sigma}_*^{1/2}\boldsymbol{\theta}_*\|_2^2$ from Remark C.3. Overall, by (19) and (24), we have

$$
\mathbf{Var}(f_{\mathrm{w2s}})\leqslant\frac{\sigma^2}{4(\alpha_w n)(\alpha_{\mathrm{w2s}}N)}\left(\left(1+\frac{1}{N}\right)\operatorname{tr}\left(\mathbf{\Sigma}_s\mathbf{\Sigma}_w\right)+\frac{1}{N}\operatorname{tr}(\mathbf{\Sigma}_s)\operatorname{tr}(\mathbf{\Sigma}_w)\right),
$$

$$
\mathbf{Bias}(f_{\mathrm{w2s}})\leqslant\alpha_w\left\|\mathbf{\Sigma}_w^{-1/2}\mathbf{\Sigma}_*^{1/2}\boldsymbol{\theta}_*\right\|_2^2+\alpha_{\mathrm{w2s}}\left\|\mathbf{\Sigma}_s^{-1/2}\mathbf{\Sigma}_*^{1/2}\boldsymbol{\theta}_*\right\|_2^2\leqslant\alpha_w\varrho_w+\alpha_{\mathrm{w2s}}\varrho_s.
$$

The upper bound the excess risk $\mathbf{ER}(f_{\mathrm{w2s}})=\mathbf{Var}(f_{\mathrm{w2s}})+\mathbf{Bias}(f_{\mathrm{w2s}})$ is minimized by taking

$$
\alpha_w=\left(\frac{\sigma^2}{4nN}\frac{\varrho_s}{\varrho_w^2}\left(\left(1+\frac{1}{N}\right)\operatorname{tr}\left(\mathbf{\Sigma}_s\mathbf{\Sigma}_w\right)+\frac{1}{N}\operatorname{tr}(\mathbf{\Sigma}_s)\operatorname{tr}(\mathbf{\Sigma}_w)\right)\right)^{1/3},
$$

$$
\alpha_{\mathrm{w2s}}=\left(\frac{\sigma^2}{4nN}\frac{\varrho_w}{\varrho_s^2}\left(\left(1+\frac{1}{N}\right)\operatorname{tr}\left(\mathbf{\Sigma}_s\mathbf{\Sigma}_w\right)+\frac{1}{N}\operatorname{tr}(\mathbf{\Sigma}_s)\operatorname{tr}(\mathbf{\Sigma}_w)\right)\right)^{1/3},
$$

which leads to the optimal upper bound for the excess risk:

$$
\mathbf{ER}(f_{\mathrm{w2s}})\leqslant 3\left(\frac{\sigma^2}{4nN}\varrho_s\varrho_w\left(\left(1+\frac{1}{N}\right)\operatorname{tr}\left(\mathbf{\Sigma}_s\mathbf{\Sigma}_w\right)+\frac{1}{N}\operatorname{tr}(\mathbf{\Sigma}_s)\operatorname{tr}(\mathbf{\Sigma}_w)\right)\right)^{1/3}.
$$

$\square$

## D. Canonical angles

In this section, we review the concept of canonical angles between two subspaces that connect the formal definition of the correlation dimension $d_{s\wedge w}=\left\|\mathbf{V}_s^\top\mathbf{V}_w\right\|_F^2$ in Definition 2.4 to the intuitive notion of the alignment between the corresponding subspaces $\mathcal{V}_s$ and $\mathcal{V}_w$ in the introduction: $\sum\cos(\angle(\mathcal{V}_s,\mathcal{V}_w))=\left\|\mathbf{V}_s^\top\mathbf{V}_w\right\|_F^2$.

**Definition D.1** (Canonical angles (Golub & Van Loan, 2013), adapting from (Dong et al., 2024a))**.** Let $\mathcal{V}_s, \mathcal{V}_w \subseteq \mathbb{R}^d$ be two subspaces with dimensions $\dim(\mathcal{V}_s) = d_s$ and $\dim(\mathcal{V}_w) = d_w$ (assuming $d_w \geqslant d_s$ without loss of generality). The canonical angles $\angle(\mathcal{V}_s, \mathcal{V}_w) = \mathrm{diag}(\nu_1, \ldots, \nu_{d_s})$ are $d_s$ angles that jointly measure the alignment between $\mathcal{V}_s$ and $\mathcal{V}_w$, defined recursively as follows:

$$\mathbf{u}_i, \mathbf{v}_i \triangleq \mathrm{argmax}\ \mathbf{u}_i^* \mathbf{v}_i$$
$$\text{s.t. } \mathbf{u}_i \in \left(\mathcal{V}_s \setminus \mathrm{span}\{\mathbf{u}_\iota\}_{\iota=1}^{i-1}\right) \cap \mathbb{S}^{d-1},$$
$$\mathbf{v}_i \in \left(\mathcal{V}_w \setminus \mathrm{span}\{\mathbf{v}_\iota\}_{\iota=1}^{i-1}\right) \cap \mathbb{S}^{d-1}$$
$$\cos(\nu_i) = \mathbf{u}_i^* \mathbf{v}_i \quad \forall\, i = 1, \ldots, k,$$

such that $0 \leqslant \nu_1 \leqslant \ldots \leqslant \nu_k \leqslant \pi/2$.

Given two subspaces $\mathcal{V}_s, \mathcal{V}_w \subseteq \mathbb{R}^d$, let $\mathbf{V}_s \in \mathbb{R}^{d \times d_s}$ and $\mathbf{V}_w \in \mathbb{R}^{d \times d_w}$ be the matrices whose columns form orthonormal bases for $\mathcal{V}_s$ and $\mathcal{V}_w$, respectively. Then, the canonical angles $\angle(\mathcal{V}_s, \mathcal{V}_w)$ are determined by the singular values of $\mathbf{V}_s^\top \mathbf{V}_w$ (Björck & Golub, 1973, §3):

$$\cos(\angle_i(\mathcal{V}_s, \mathcal{V}_w)) = \sigma_i(\mathbf{V}_s^\top \mathbf{V}_w) \quad \forall\, i = 1, \ldots, d_s,$$

where $\sigma_i(\mathbf{V}_s^\top \mathbf{V}_w)$ denotes the $i$-th singular value of $\mathbf{V}_s^\top \mathbf{V}_w$.

In particular, since $\mathbf{V}_s, \mathbf{V}_w$ consist of orthonormal columns, the singular values of $\mathbf{V}_s^\top \mathbf{V}_w$ fall in $[0, 1]$, and therefore,

$$d_{s \wedge w} = \sum \cos(\angle(\mathcal{V}_s, \mathcal{V}_w)) = \left\| \mathbf{V}_s^\top \mathbf{V}_w \right\|_F^2 \in [0, \min\{d_s, d_w\}].$$

# E. Additional experiments
## E.1. Additional experiments and details on UTKFace regression
This section provides some additional details and results for the UTKFace regression experiments in Section 4.2.

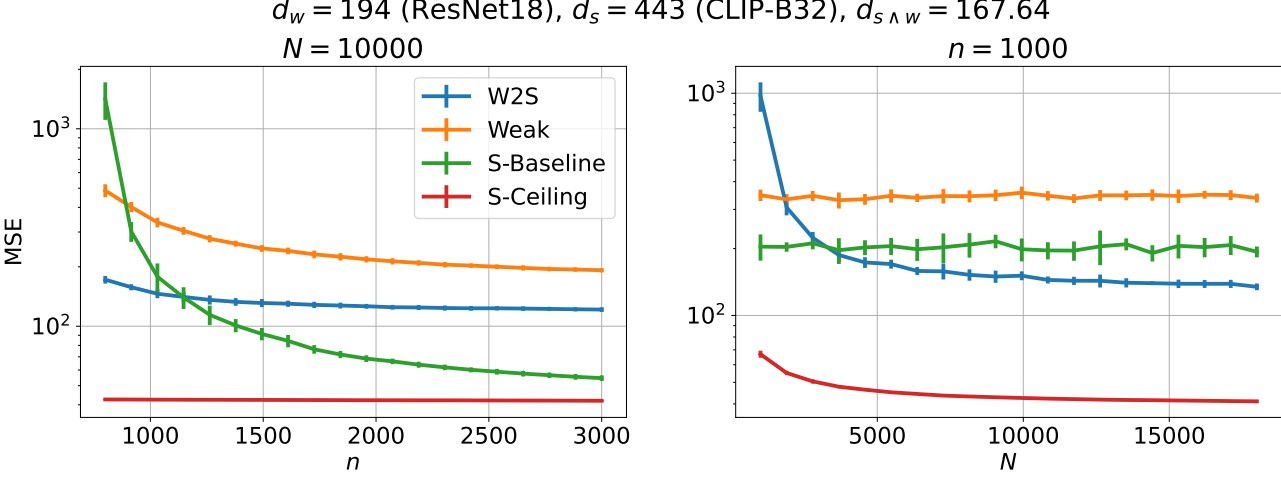

*Figure 7.* Scaling for MSE on UTKFace with `CLIP-B32` as the strong student and `ResNet18` as the weak teacher

We summarize the relevant dimensionality in Table 1. We observe the following:

- The intrinsic dimensions of the pretrained features are significantly smaller than the ambiance feature dimensions, which is consistent with our theoretical analysis and the empirical observations in (Aghajanyan et al., 2021).

- The correlation dimensions $d_{s \wedge w}$ are considerably smaller than the corresponding intrinsic dimensions, indicating that the subspaces spanned by the weak and strong features are not aligned in practice. As shown in Section 4.2, such discrepancies in the weak and strong features facilitate W2S generalization.

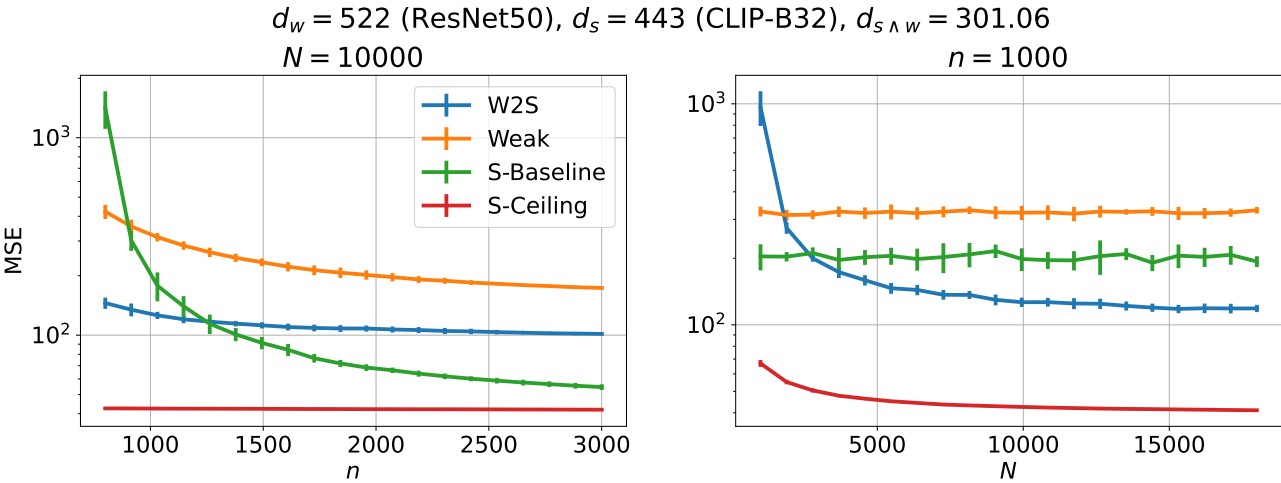

*Figure 8.* Scaling for MSE on UTKFace with `CLIP-B32` as the strong student and `ResNet50` as the weak teacher

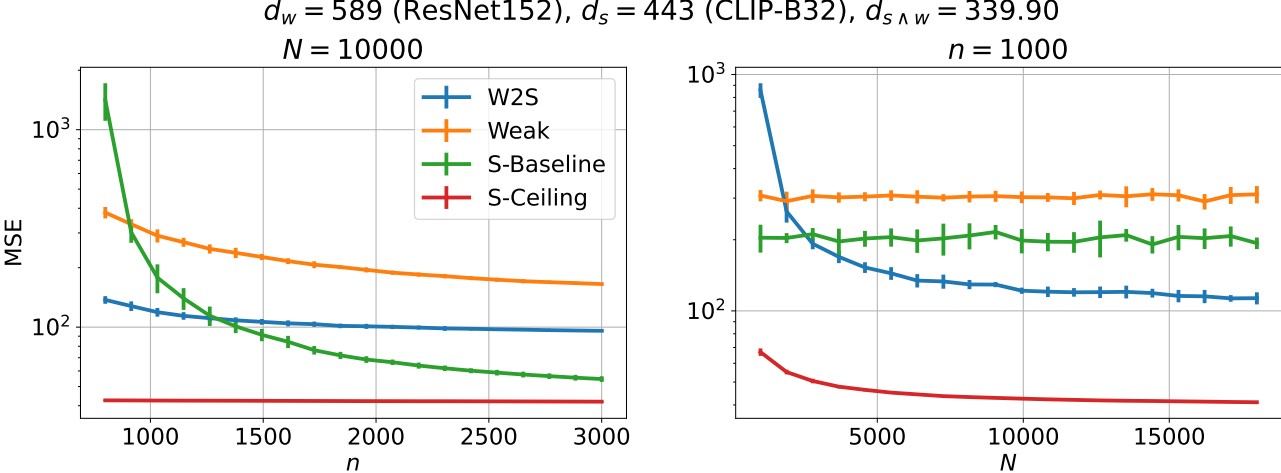

*Figure 9.* Scaling for MSE on UTKFace with `CLIP-B32` as the strong student and `ResNet152` as the weak teacher

*Table 1.* Summary of the pretrained feature dimensions, along with the intrinsic dimensions $d_s, d_w$ and correlation dimensions $d_{s \wedge w}$ (with respect to the strong student `CLIP-B32`) computed over the entire UTKFace dataset (including training and testing).

| Pretrained Model | Feature Dimension | Intrinsic Dimension ($\tau = 0.01$) | Correlation Dimension |
|---|---|---|---|
| ResNet18 | 512 | 194 | 167.64 |
| ResNet34 | 512 | 150 | 129.97 |
| ResNet50 | 2048 | 522 | 301.06 |
| ResNet101 | 2048 | 615 | 354.52 |
| ResNet152 | 2048 | 589 | 339.90 |
| CLIP-B32 | 768 | 443 | $\times$ |

For reference, we provide the scaling for MSE losses of three representative teacher-student pairs in Figures 7 to 9.

- It is worth highlighting that while the MSE loss of $f_{\text{w2s}}$ monotonically decreases with respect to both sample sizes $n, N$, the different rates of convergence compared to $f_w$, $f_s$, and $f_c$ lead to the distinct scaling behavior of the relative W2S performance (**PGR** and **OPR**) with respect to $n$ versus $N$ in Figures 5 and 6.

- When the strong student has a lower intrinsic dimension than the weak teacher (*cf.* Figure 7 versus Figures 8 and 9), $d_s < d_w$, the W2S model $f_{\text{w2s}}$ tends to achieve better generalization in terms of the test MSE. This is consistent with our analysis in Section 3.1.

- When $d_s < d_w$, the W2S model $f_{\text{w2s}}$ tends to achieve (slightly) better generalization for (slightly) smaller correlation dimension $d_{s \wedge w}$ (*cf.* Figure 8 versus Figure 9), again coinciding with our analysis in Section 3.1.

- W2S generalization generally happens (*i.e.* $f_{\text{w2s}}$ is able to outperform $f_w$) with sufficiently large sample sizes $n, N$. However, as the labeled sample size $n$ increases, the test MSE of $f_{\text{w2s}}$ converges slower than that of the strong baseline and ceiling models, $f_s$ and $f_c$, leading to the inverse scaling for **PGR** and **OPR** with respect to $n$ in Figures 5 and 6. When $n$ is too large, the W2S model $f_{\text{w2s}}$ may not be able to achieve better generalization than the strong baseline $f_s$.

### E.2. Experiments on image classification

**Dataset.** ColoredMNIST (Arjovsky et al., 2019) consists of groups of different colors and reassign the label to be binary (digits 0-4 vs 5-9). We pool together the groups to form one dataset. The choice is to bring diversity to the feature space with additional color features and thus potential feature discrepancies. We hold out a test set of 7000 samples and use the rest 63000 samples for training.

**Linear probing over pretrained features.** We fix a strong student as DINOv2-s14 (Oquab et al., 2024) and vary the weak teacher among the ResNet-d series and ResNet series (ResNet18D, ResNet34D, ResNet101, ResNet152) (He et al., 2019; 2016). We replace ResNet18 and ResNet34 used in Section 4.2 to experiment on weak models with similar intrinsic dimensions but different correlation dimensions. We treat the backbone of the models (excluding the classification layer) as $\phi_s$ and $\phi_w$ and finetune them via linear probing. We train the models with cross-entropy loss and AdamW optimizer. We tune the training hyperparameters of weak and strong models using a validation set and train them for 800 steps with a learning rate 1e-3 and weight decay 1e-6.

*Table 2.* Summary of the pretrained feature dimensions, along with the intrinsic dimensions $d_s, d_w$ and correlation dimensions $d_{s \wedge w}$ (with respect to the strong student `DINOv2-S14`) computed over the entire ColoredMNIST dataset (including training and testing).

| Pretrained Model | Feature Dimension | Intrinsic Dimension ($\tau = 0.01$) | Correlation Dimension |
|---|---|---|---|
| ResNet-18-D | 512 | 117 | 6.23 |
| ResNet-34-D | 512 | 127 | 7.07 |
| ResNet101 | 2048 | 121 | 1.74 |
| ResNet152 | 2048 | 128 | 1.88 |
| DINOv2-S14 | 384 | 28 | × |

**Discrepancies lead to better W2S.** Figure 10 shows the scaling of **PGR** and **OPR** with respect to the sample sizes $n, N$ for different weak teachers in the ResNet series with respect to a fixed student, `CLIP-B32`. As in Section 4.2, we observe that with similar intrinsic dimensions $d_s, d_w$, W2S achieves better relative performance in terms of **PGR** and **OPR** when the correlation dimension $d_{s \wedge w}$ is smaller.

**Variance reduction is a key advantage of W2S.** We inject noise to the labels of the original ColoredMNIST training samples by randomly flipping the ground truth labels with probability $\varsigma \in [0, 1]$ (following (Arjovsky et al., 2019)). Figure 11 shows the scaling of **PGR** and **OPR** with respect to $n$ and $N$ when taking DINOv2-S14 as the strong student and ResNet101 as the weak teacher. We observe that the larger artificial label noise $\varsigma$ leads to higher **PGR** and **OPR**.

### E.3. Experiments on sentiment classification

**Dataset.** The Stanford Sentiment Treebank (Socher et al., 2013) is a corpus with fully labeled parse trees that allows for a complete analysis of the compositional effects of sentiment in language. The corpus is based on the dataset introduced by Pang & Lee (2005) and consists of 11,855 single sentences extracted from movie reviews. It was parsed with the Stanford parser and includes a total of 215,154 unique phrases from those parse trees, each annotated by 3 human judges. We conduct binary classification experiments on full sentences (negative or somewhat negative versus somewhat positive or positive,

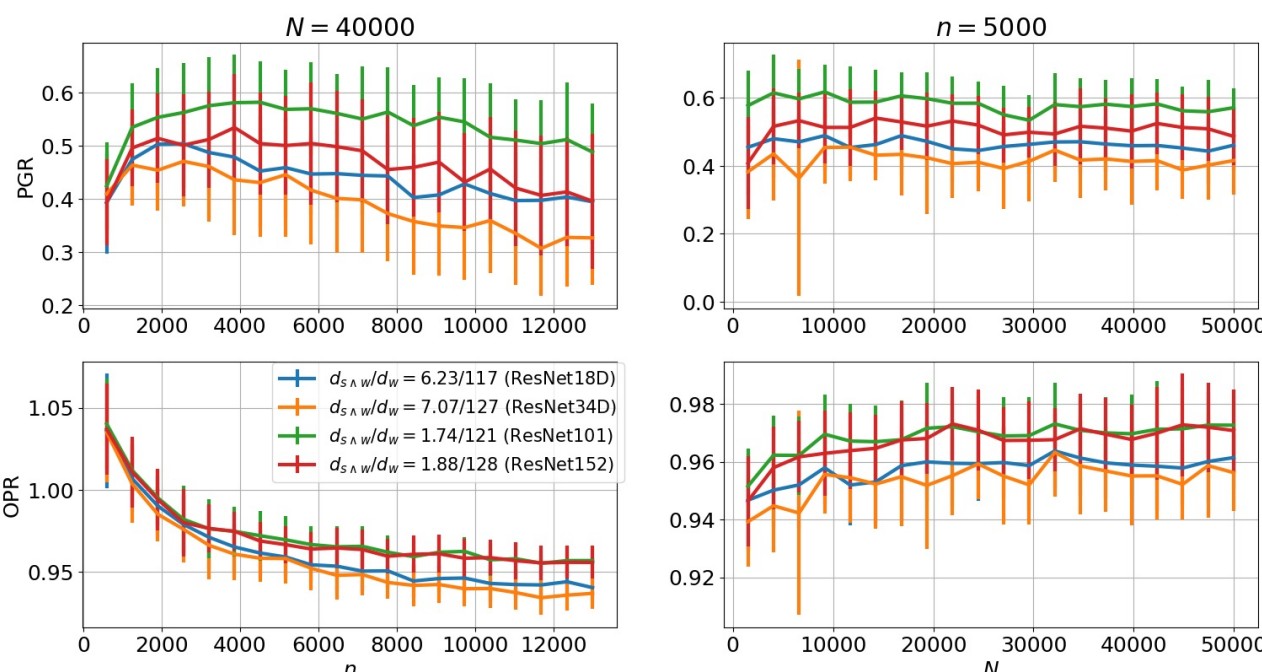

*Figure 10.* Scaling for **PGR** and **OPR** of different weak teachers with a fixed strong student on ColoredMNIST.

with neutral sentences discarded), the so-called SST-2 dataset, and split the dataset into training and testing sets of sizes 63000 and 4349. Generalization errors are estimated with the 0-1 loss over the test set.

**Full finetuning.** We fix the strong student as Electra-base-discriminator (Clark et al., 2020) and vary the weak teacher among the Bert series (Turc et al., 2019) (Bert-Tiny, Bert-Mini, Bert-Small, Bert-Medium). With manageable model sizes, we conduct full finetuning experiments following the setup in (Burns et al., 2024). We use the standard cross-entropy loss for supervised finetuning. When training strong students on weak labels (W2S), we use the confidence-weighted loss proposed by (Burns et al., 2024), which is suggested to be able to improve weak-to-strong generalization on many NLP tasks. All training is conducted via Adam optimizers (Kingma & Ba, 2014) with a learning rate of 5e-5, a cosine learning rate schedule, and 40 warmup steps. We train for 3 epochs, which is sufficient for the train and validation losses to stabilize.

**Intrinsic dimension.** The intrinsic dimensions $d_w, d_s$ are measured based on the Structure-Aware Intrinsic Dimension (SAID) method proposed by (Aghajanyan et al., 2021). We first train the full models on the whole training set, and then train the models with only $d$ trainable parameters based on SAID transformation. The $d_w$ or $d_s$ are the smallest number of parameters under SAID that is necessary to retain 90% accuracy of the full models. Here, the 90% accuracy is a common threshold used to estimate intrinsic dimensions in the literature (Li et al., 2018).

**Correlation Dimension.** Let $D_s, D_w \in \mathbb{N}$ be the finetunable parameter counts of the strong and weak models, respectively. For full FT whose dynamics fall in the kernel regime, as explained in Remark 2.5, the strong and weak "features" become the gradients[8], $\mathbf{\Phi}_s = \nabla_{\boldsymbol{\theta}} f_s(\mathbf{X}|\theta_s^{(0)}) \in \mathbb{R}^{N \times D_s}$ and $\mathbf{\Phi}_w = \nabla_{\boldsymbol{\theta}} f_w(\mathbf{X}|\theta_w^{(0)}) \in \mathbb{R}^{N \times D_w}$, of the respective models at the pretrained initialization, $\theta_s^{(0)} \in \mathbb{R}^{D_s}$ and $\theta_w^{(0)} \in \mathbb{R}^{D_w}$.

A practical challenge is that $D_s, D_w, N$ are all huge for full FT on most NLP tasks, making it infeasible to compute the $D_s \times D_s$ and $D_w \times D_w$ Gram matrices and their spectral decompositions. As a remedy, we leverage the significantly lower

---

[8]Notice that $f_s, f_w$ are scalar-valued functions for binary classification tasks like SST-2, and thus the gradients $\nabla_{\boldsymbol{\theta}} f_s$ and $\nabla_{\boldsymbol{\theta}} f_w$ are row vectors. For multi-class classification tasks where $f_s, f_w$ output vectors of logits, a common heuristic to keep $\mathbf{\Phi}_s, \mathbf{\Phi}_w$ as matrices of manageable sizes (in constrast to tensors) is to replace gradients of the models, $\nabla_{\boldsymbol{\theta}} f_s$ and $\nabla_{\boldsymbol{\theta}} f_w$, with gradients of MSE losses at the pretrained initialization. The gradients of MSE can be viewed as a weighted sum of the model gradients for each class.

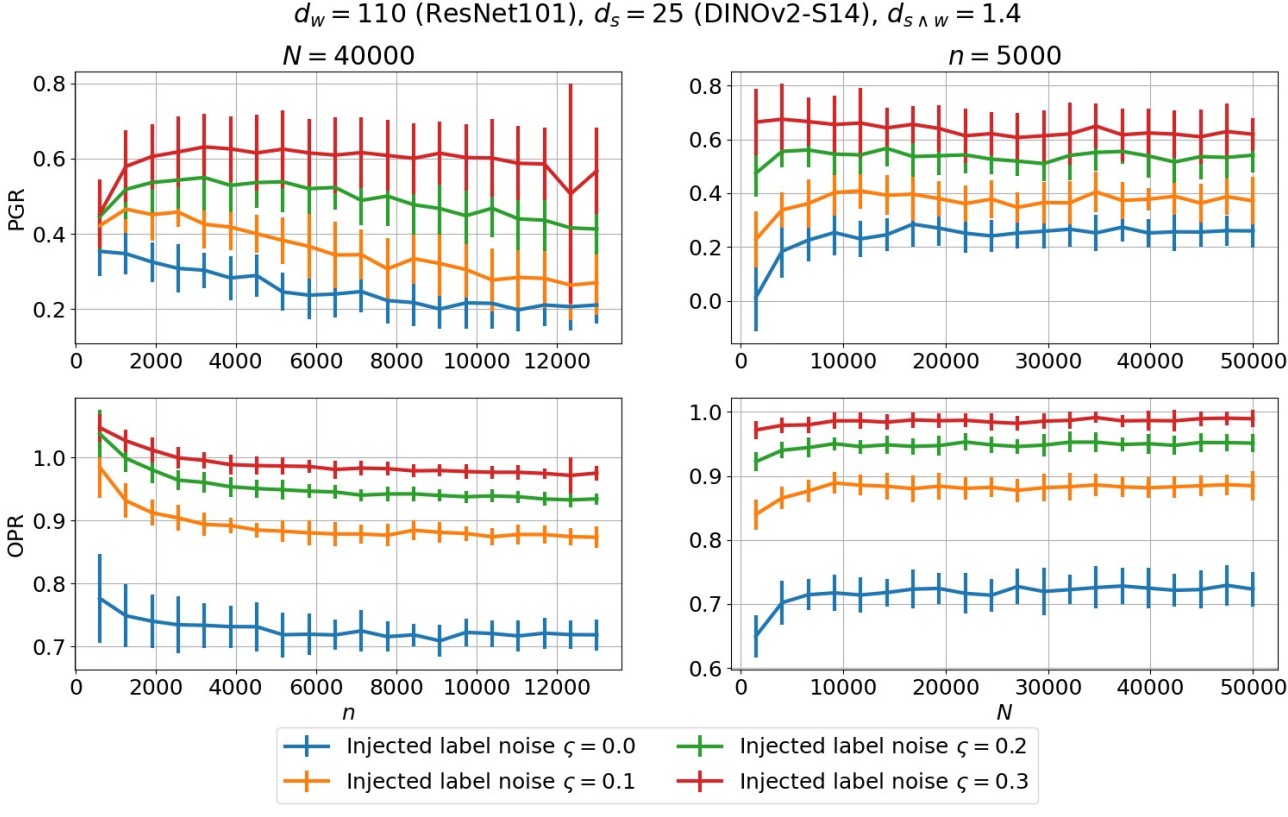

*Figure 11.* Scaling for **PGR** and **OPR** of W2S on ColoredMNIST with injected label noise.

intrinsic dimensions $d_s \ll D_s, d_w \ll D_w$ (see Table 2) to accelerate estimation of $d_{s \wedge w}$ via sketching (Halko et al., 2011; Woodruff et al., 2014).

(i) We first reduce both $D_s, D_w$ to the same lower dimension $D = 0.01 \min\{D_s, D_w\}$ (with $D \gg \max\{d_s, d_w\}$) by subsampling columns of $\mathbf{\Phi}_s, \mathbf{\Phi}_w$ (uniformly for efficiency, or adaptively via sketching-based interpolative decomposition (Dong & Martinsson, 2023) when affordable) to obtain $\mathbf{\Phi}'_s, \mathbf{\Phi}'_w \in \mathbb{R}^{N \times D}$.

(ii) Then, we use randomized rangefinder (Halko et al., 2011, Algorithm 4.1) to approximate the first $d_s, d_w$ right singular vectors, $\mathbf{V}_s \in \mathbb{R}^{D \times d_s}$ and $\mathbf{V}_w \in \mathbb{R}^{D \times d_w}$, of $\mathbf{\Phi}'_s, \mathbf{\Phi}'_w$. Taking the evaluation of $\mathbf{V}_s$ as an example, we draw a Gaussian random matrix $\mathbf{G}_s \in \mathbb{R}^{d_s \times D}$ and compute the orthornormalization $\mathbf{V}_s = \mathrm{ortho}(\mathbf{\Phi}'^{\top}_s \mathbf{G}_s)$ via QR decomposition.

(iii) Finally, we compute the correlation dimension $d_{s \wedge w} = \left\| \mathbf{V}_s^{\top} \mathbf{V}_w \right\|_F^2$.

*Table 3.* Summary of finetunable parameter counts $D_s, D_w$, intrinsic dimensions $d_s, d_w$, and correlation dimensions $d_{s \wedge w}$ (with respect to the strong student `Electra`) computed over the entire SST-2 dataset (including training and testing).

| Pretrained Model | $D_s, D_w$ | Intrinsic Dimension ($\tau = 0.01$) | Correlation Dimension |
|---|---|---|---|
| Bert-Tiny | 4.4M | 7000 | 81.13 |
| Bert-Mini | 11.2M | 8500 | 38.67 |
| Bert-Small | 28.8M | 8000 | 26.19 |
| Bert-Medium | 41.4M | 4000 | 8.52 |
| Electra | 109.5M | 700 | $\times$ |

**Discrepancies lead to better W2S.** Figure 12 shows the scaling of **PGR** and **OPR** with respect to $n$ and $N$ for different $d_{s \wedge w}$. As in Section 4.2 and Appendix E.2, we observe the better relative W2S performance in terms of **PGR** and **OPR**

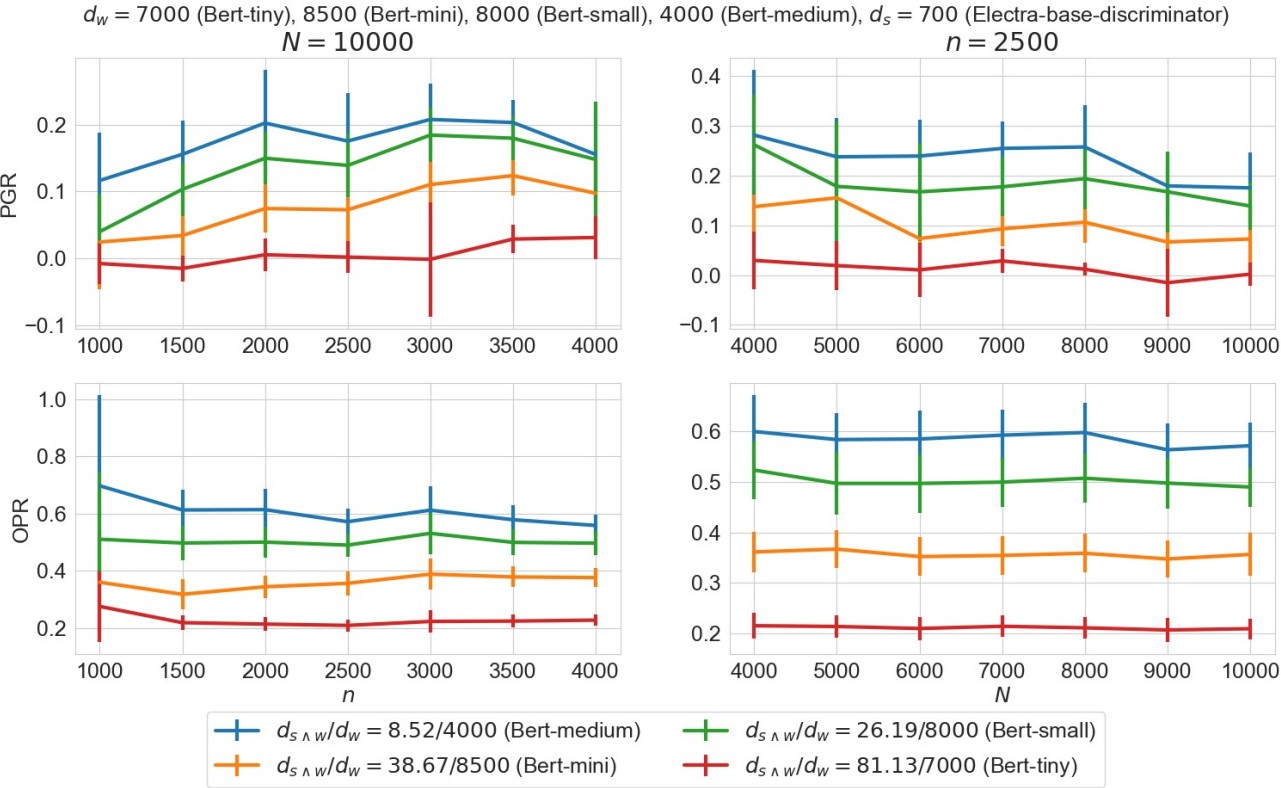

*Figure 12.* Scaling for **PGR** and **OPR** of different weak teachers with a fixed strong student on SST-2.

when $d_{s\wedge w}/d_w$ is smaller.

**Variance reduction is a key advantage of W2S.** We inject noise to the labels of training samples by randomly flipping labels with probability $\varsigma = 0, 0.1, 0.2, 0.3$. Figure 13 shows the scaling of **PGR** and **OPR** with respect to $n$ and $N$ when taking Electra as the strong student and Bert-Medium as the weak teacher. We observe that the larger artificial label noise $\varsigma$ leads to higher **PGR** and **OPR**.

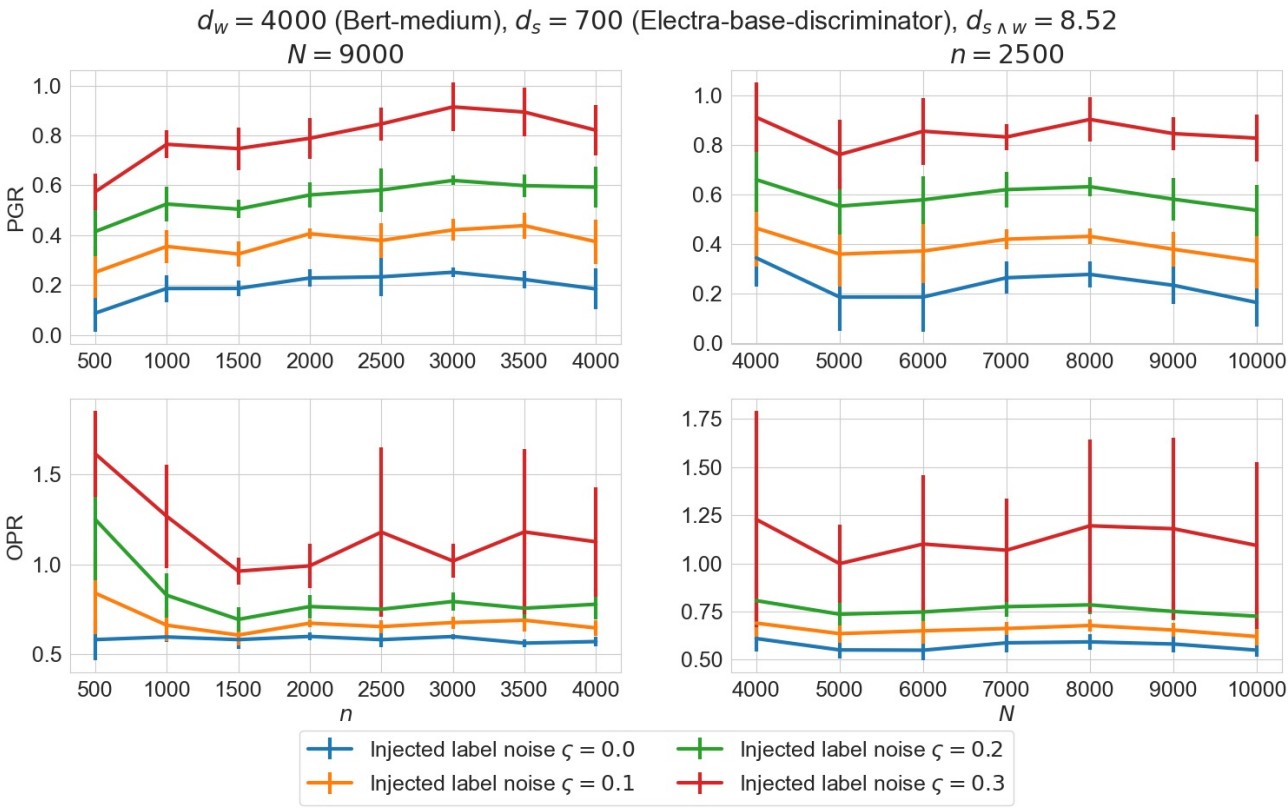

*Figure 13.* Scaling for **PGR** and **OPR** of W2S on SST-2 with injected label noise.

