# OpenReview forum: "Discrepancies are Virtue: Weak-to-Strong Generalization through Lens of Intrinsic Dimension"
_ICML.cc/2025/Conference — ICML 2025 poster_

### Official Review · Reviewer_W6Uk · 2025-03-07

**Overall Recommendation:** 3

**Summary:**

The paper considers the problem of weak-to-strong generalization, which is the study of how the performance of a strong model trained on pseudo labels of a weaker model generalize. More formally the following setup is consider. Two features transforms, $\phi_s$ the strong model and $\phi_w$ the weak model, are given where it is a assumed that the feature transform $\phi_s$ and $\phi_w$ has output in some spaces with dimension respectively $d_s$ and $d_w$.
Now a linear layer $\theta_w$ ontop of the weak model $\phi_w$ is trained on a dataset

$\tilde{S}=((\tilde{x}_1,\tilde{y}_1),\ldots,(\tilde{x}_n,\tilde{y}_n))$

 of i.i.d. examples where $\tilde{x}_1\sim \mathcal{D}$

 and ${\tilde{y}}_1=f^*(\tilde{x}_1)+z_1$,

where $z_1 \sim  N(0,\sigma^2)$, $f^*$ being uniformly bounded by 1. Now given a second dataset $S=((x_1,y_1),\ldots,(x_N,y_N))$, drawn i.i.d. in the same manner as $\tilde{S}$, where the learner is only given the unlabelled data $x_1,\ldots,x_N$ a linear layer $\theta_{w2s}$ ontop of the strong model $\phi_s$ is trained with the pseudo labels given by $\theta_w$, i.e. the data set $(x_1,x_1\theta_w),\ldots,(x_N,x_N\theta_w).$ The paper then want to show how the strong model generalize on new data examples $(x,y)$ drawn in the above manner. To this end the paper studies the expected generlization error of $\theta_{w2s},$ i.e. $err(\theta_{w2s})=E_x[E_f[(\theta_{w2s}\phi_s(x)-f*(x))^2]]$ via the bias variance decomposition. Under sufficient conditions, the paper shows that the bias can be bounded by the sum of the generalization error of the best fixed linear lay on top of $\phi_s$, $\rho_s= \min_\theta E_x[(\theta\phi_s(x)-f*(x))^2]$ and the best fixed linear lay on top of $\phi_w$,  $\rho_w= \min_\theta E_x[(\theta\phi_w(x)-f*(x))^2].$ The variance term of the strong model can be bounded by $\sigma/(n-d_w-1) (d_{sw}+d_s(d_w-d_{sw}/N),$ where $d_{sw}$ denotes the intersection of the subspaces that $\phi_s$ and $\phi_w$ spans, here the term $\sigma d_{sw}/(n-d_w-1)$ do not change with the number of pseudo labels where as the term $\sigma/(n-d_w-1)d_s(d_w-d_{sw}/N),$ does.
Thus, this theorem suggests that the generalization error of the stronger model trained with more pseudolabels improves this variance term, where as the other terms do not change with more pseudolabels. This variance-term can be thought of the discrepancy of the subspaces span by $\phi_s$ and $\phi_w$. For comparison of the performance of $\theta_s$, the paper also considers the model $\theta_f$ a linear layer trained ontop of the strong model $\phi_s$ with the dataset
$\tilde{S}$ and  the model $\theta_c$ a linear layer trained ontop of the strong model $\phi_s$ with the dataset $\tilde{S}\cup S.$
From these quantities, the paper defines the Performance Gap recovery, shorten PGR
$$PGR=\frac{err(\theta_w)-err(\theta_{w2s}}{err(\theta_w)-err(\theta_{w2s}}$$
and the Outperforming ratio (OPR)
$$OPR=\frac{err(\theta_{s})}{err(\theta_{w2s})}.$$
The paper then shows that under suitable conditions, which on a high levels says that the noise $\sigma$ is much larger than the approximations errors $\rho_s$ and $\rho_w,$ $(\rho_s+\rho_w) / \sigma \rightarrow 1$ and $n$ and $N$ large enough, then the PGR becomes lower bounded by $1-(d_{sw}/d_w)$ where it is argued that it have been observed in some experiments that $d_{sw}/d_w$ is small, since the weak model as a "large" dimension and the intersection of subspace spanned by the strong and the weak models $d_{sw}$ being "small". Furthermore, under these assumptions the OPR becomes $d_s/d_{sw}$, again being large since $d_{sw}\leq d_s.$ Furthermore when $(\rho_s+\rho_w)/\sigma \not\rightarrow 0$, the paper observes that the PGR and OPR is non-monotonic in $n$. Solving for the optimal $n$, the paper show that when $(\rho_s+\rho_w)/\sigma<<1$ and $N$ is large enough the $PGR$ and $OPR$ under this optimal $n$ becomes larger.
In general when the approximation errors $\rho_w+\rho_s$ is larger than $\sigma$ the lower bounds on PGR and OPR becomes not so meaningful.
The paper then examine there findings on an synthetic datasets and a real-world datasets. On the synthetic datasets where $\rho_w+\rho_s$ is smaller than $\sigma$, $\theta_{w2s}$ outperformes $\theta_s$ when $n<<N$, and in the case $\rho_w+\rho_s$ is close to $\sigma$, $\theta_{w2s}$'s performance is close to $\theta_w$ so giving little or no weak to strong generalization.
For the synthetic datasets with $\rho_w+\rho_s$ smaller than $\sigma$ the PGR as a function of $n$ shows a decreasing trend and as a function of $N$ shows a no monotonic trend, with an increase in the beginning and then turns and decreases, the increase is steeper than the decrease. The OPR as a function of $n$ shows a decreasing trend and the OPR as a function of N shows a non decreasing trend. For the real word dataset the picture is similar expect now PGR can go negative and PGR as a function of N is increasing.

**Claims And Evidence:**

There is no proofs in the main, which is deferred to the appendixes. The experiments are very small and I think should be seen as a pilot study giving some indications of what the theory showed.

**Essential References Not Discussed:**

To the best of my knowledge no.

**Experimental Designs Or Analyses:**

I did not check the code for the experiments.

**Methods And Evaluation Criteria:**

As my comment above I think the experiments should be seen as a complement to the theoretical works of this papers and there is room to further study the theoretical findings of this paper on more datasets to give a stronger indication on to what extend the theoretical frame work explains strong to weak generalization.

**Other Comments Or Suggestions:**

Here are the notes that I have collected while reading your nice paper. The first questions/notes(the enumarted ones) would be nice to get and answer on but it is very much not required. The last bullets points are notes i took while reading.

1. page 3 second column line 110, why can the boundedness over the whole distribution be assumed without loss of generality, i get it with normalization, but f* could be unbounded?

3. page 6 first column line 308. what is meant with $(\rho_s+\rho_w)/\sigma \rightarrow 0$ my understanding of theses quanties it not being dependent on neither n and N, so what is meant by going to 0?

4. page 6 second column line 308-312, i feel like the conclusion made in the paragraph is the same as in the paragraph page 6 first column line 318 to 276 in the second column, but no without any concrete lower bound, is this correct?

5. page 7 second column line 338, Is it that intriguingly that f_s outperfroms f_w2s when n\approx N? In this case f_ws was trained on N examples labelled by the weaker model, and f_s was trained on n examples from the true distribution?

6. page 8 first column line 417, why not report the spectrum such that one can get an idea of changing the cut off would change $d_s$ and $d_w$?

- Allot of assymptotical notation was defined what is meant by $<<$
- page 4 first column line 206 what is the gradient of $\nabla_\theta$ taken with respect to?

**Other Strengths And Weaknesses:**

The paper is well written, and explains nicely how they see their findings.

**Questions For Authors:**

1. It is a bit unclear to me what the technical contribution was? Do the proofs require new novel ideas, or is it similar to previous work of X, but the novelty is to put it in the context of weak to strong learning?

2. page 3 second column line 132, the embedings goes $\phi_s$ and $\phi_w$ to the same dimension, which my guess is because one wants to define the intersection of their covariance matrices later. What could be done without this assumption in terms of intrinsic dimension?

To the end of the above question

3. page 8 first column line 413-419, do ResNet50 and ResNet15 have the same output dimension?

Thanks for the response. I would like to keep my initial score. This decision is based on the response, which clarified my understanding of the paper and reading the other reviewers' comments and concerns.

**Relation To Broader Scientific Literature:**

The paper first examines the empirical literature, which is related to the paper in the sense that this was the ground for the interest in theoretically better understanding W2S, generalization that this paper studies. More specifically the W2S concept was introduced by Burns et. al. 2023. The theoretical work which the paper says is the closest is Ildiz et al, which also studies the ridgeless regressors, assumably also with the focus on variance reduction, but here the paper studies variance reduction from an intrinsic dimension perspective inspired by empirical observations by Aghajanyan et al. 2020.

**Theoretical Claims:**

There was no proofs in the main so i did not check the correctness of the proofs.

---

> ### Author Rebuttal · Authors · 2025-03-30
>
> We thank the reviewer for their constructive suggestions and supportive feedback. We are glad that they found our paper well-written and provided novel perspectives on W2S. Since we could not include figures in OpenReview rebuttal, we will synopsize our new experiments in text and present the formal results in an anonymous URL: https://anonymous.4open.science/r/W2S_IntrinsicDim_demo-FEAF/demo.pdf.
>
> ## Key clarifications
> 1. **Additional experiments on image regression and LLMs**: We appreciate the suggestion on strengthening the experimental validation. In the revision, we will include additional real experiments on image regression (UTKFace) based on CLIP and ResNet embeddings, as well as NLP task (SST-2) based on finetuning LLMs (see the URL for concrete plots). In particular, our new experiments provide strong empirical evidence for our theoretical results:
>     1. **Discrepancies lead to better W2S**: As Figures 4 and 5 in the submission, we plot the scaling of PGR/OPR against sample sizes $n,N$ for weak-strong pairs with various correlation dimensions on real vision and NLP tasks. The results show that the lower $d_{s \wedge w}/d_w$ (larger discrepancy) consistently brings better W2S.
>     2. **Variance reduction is a key advantage of W2S**: While the variance-dominated regime that we focus on is overlooked by most prior theoretical works on W2S, our new experiments confirm that this is a particularly important regime where strong W2S performance (in terms of PGR/OPR) occurs in practice. By injecting artificial label noise that leads to larger variance, we observe significant improvements in PGR and OPR, implying that variance reduction is a key advantage of W2S.
>
> 2. **Our analysis extends to weak-strong pairs with different feature dimensions**: Please see **R#2 (WLsf), Key#3** for detailed explanations.
>
> 3. **Empirical estimation of intrinsic and correlation dimensions**: Thanks for the question. In practice, the feature dimensions of the weak and strong models can be different. Please see **R#1 (5dnE), Key#2.1 and 2.2** for how we estimate $d_{s \wedge w}$ in practice.
>
> 4. **Our main technical contributions** are:
>     1. the theoretical framework based on low intrinsic dimensions of finetuning that provides **a theoretical underpinning for W2S in the variance-dominated regime**, along with
>     2. the insight on **how weak-strong discrepancy brings better W2S generalization**.
>
>     As pointed out in **R#2 (WLsf), Key#4**, W2S has shown appealing empirical performance when variance dominates (e.g., see Key#1.2). However, the variance-dominated regime is often overlooked in prior analyses on W2S. (We kindly point out the confusion that Ildiz et al. (2024) explain W2S from the bias instead of variance perspective. We will further clarify this in the revision.) To the best of our knowledge, this is **the first theoretical explanation for W2S in the important variance-dominated regime**. While our analysis is built upon various fundamental tools in random matrix theory, high-dimensional statistics, and linear algebra, the unique structure of the learning problem posed by W2S makes the combination of these tools highly non-trivial. Moreover, our framework is flexible enough to be extended to finetuning problems with similar structures like knowledge distillation and model collapse. We will clarify this in the revision.
>
> ## Detailed responses
> We appreciate the detailed suggestions on notations and presentation. Here, we address questions in "Other Comments Or Suggestions" (O):
> * O#1: We assume $f_*$ being bounded for conciseness of the analysis. This can be trivially relaxed by adding a constant $C = \max_x |f(x)|$ to the generalization error in our analysis.
> * O#2, 3: In Case I, $(\rho_s + \rho_w)/\sigma^2 \to 0$ essentially says that the FT approximation errors are small enough compared to the label noise so that we can treat them as zeros. In contrast, for Case II, we define a small constant $\varrho = (\rho_s + \rho_w)/\sigma^2 > 0$ to quantify the variance domination. This $\varrho$ affects the PGR and OPR lower bounds in Coro. 3.6. While the quantitative conclusions are different, it's correct that the qualitative takeaways for both cases are similar.
> * O#4: We agree. This is exactly a simple explanation for how $f_s$ eventually outperforms $f_{w2s}$ as $n$ increases. We will rephrase the sentence to avoid ambiguity.
> * O#5: The reason why we did not report the spectra is that they do not contain much relevant information. See the spectra of ResNet embeddings in our new UTKFace experiments for examples: https://anonymous.4open.science/r/W2S_IntrinsicDim_demo-FEAF/fig/utkface_svd_cutoff0.99.pdf.
>
> We are happy to answer any further questions you may have. If our responses above help address your concerns, we would truly appreciate a re-evaluation accordingly.

---

### Official Review · Reviewer_299q · 2025-03-11

**Overall Recommendation:** 3

**Summary:**

The paper studies weak-to-strong generalization in ridgeless regression with (sub-)Gaussian features. It reveals that weak-to-strong generalization arises from the discrepancy between the weak model's features and the strong model's features.

**Claims And Evidence:**

The main theorem statement is clearly presented, and the proof in the appendix  appear to be well-structured and understandable.

**Essential References Not Discussed:**

I believe most related works are cited in the manuscript, at least regarding the theory of weak-to-strong generalization, which I am familiar with. However, several relevant references have been released after the ICML submission deadline, and I hope the authors will cite and discuss them in the next revision. Here are some of them (but not limited to these):

[1] Medvedev, Marko, et al. "Weak-to-Strong Generalization Even in Random Feature Networks, Provably." *arXiv preprint arXiv:2503.02877* (2025).

[2] Yao, Wei, et al. "Understanding the Capabilities and Limitations of Weak-to-Strong Generalization." *arXiv preprint arXiv:2502.01458* (2025).

**Experimental Designs Or Analyses:**

The authors provide experimental results on synthetic regression and an image classification task. However, in the image classification task, I have concerns about the choice of intrinsic dimension. Why do the authors define the intrinsic dimension as 90% of the trace of the feature matrix? I believe it is essential to analyze the spectrum of the feature matrix, as the intrinsic dimension and trace can differ between the strong and weak models. Additionally, I am confused about the meaning of "threshold dimension $k$" in line 416 (left).

**Methods And Evaluation Criteria:**

In their main results and experimental evaluations, the authors use PGR and OPR, which are standard evaluation metrics in the weak-to-strong generalization literature.

**Other Comments Or Suggestions:**

Here are some minor comments on the manuscript:

- Typo: Line 142 (left): $p \in [0,1]^b$ → $p \in [0,1]^n$
- Clarification: Line 158, 162 (left): $\mathcal{D} : \mathcal{X} \times \mathcal{Y} \rightarrow [0,1]$. I believe it is not appropriate to represent the data distribution as a function mapping to [0,1].
- Clarification: Line 155–156 (right): Does $\dagger$ refer to the Moore–Penrose inverse? I think it would be better to clarify this.
- Figures: The figures in the paper have low resolution. I suggest the authors provide high-resolution versions (e.g., in .pdf format).

**Other Strengths And Weaknesses:**

One weakness of the paper is its simplified problem setting, specifically ridgeless regression and (sub-)Gaussian feature maps. However, the authors provide a strong motivation for this choice, and I believe that results derived from a well-motivated simple setting can also be a strength of the work.

**Questions For Authors:**

Can you provide a high-level intuition on why discrepancies between the features of the weak and strong models lead to weak-to-strong generalization? I believe the current version of the paper lacks this high-level intuition, even though the theoretical results are interesting. If the authors can provide a clearer high-level explanation of their findings and promise to emphasize this in the next revision, I am open to increasing my score.

**Relation To Broader Scientific Literature:**

This provides novel insights into understanding the weak-to-strong generalization phenomenon, particularly in relation to intrinsic dimension. This contribution enhances the broader understanding of weak-to-strong generalization.

**Theoretical Claims:**

I have checked the proof of the main results, and it appears to be correct.

---

> ### Author Rebuttal · Authors · 2025-03-30
>
> We thank the reviewer for their time and suggestions. We are glad that they found our paper well-written and provides novel understanding for W2S. Since we could not include figures in OpenReview rebuttal, we will synopsize our new experiments in text and present the formal results in an anonymous URL: https://anonymous.4open.science/r/W2S_IntrinsicDim_demo-FEAF/demo.pdf.
>
> ## Key clarifications
> 1. **The key intuitions on how discrepancies lead to better W2S are unrolled in two steps in the introduction**:
>     1. In Lines 68-79, we first break down our main theoretical result $$Var(f_{w2s}) \asymp \frac{d_{s \wedge w}}{n} + \frac{d_s}{N}\frac{d_w - d_{s \wedge w}}{n}$$ to explain **why a larger discrepancy (smaller $d_{s \wedge w}$) leads to better W2S**. That is, the strong student mimics variance of the weak teacher in the overlapped subspace of weak and strong features (with dimension $d_{s \wedge w}$). In contrast, **variance in the discrepancy subspace between weak and strong features (with dimension $d_w - d_{s \wedge w}$) is reduced by a factor of $\frac{d_s}{N}$ in W2S**. Recall that $d_s$ is the low intrinsic dimension, and the unlabeled W2S sample size $N$ is generally large in practice. Therefore, a multiplicative factor of $\frac{d_s}{N}$ significantly reduces the variance in the discrepancy subspace, leading to better W2S performance.
>     2. Then, in Lines 91-107, we use an example to provide high-level intuition for **how variance reduction in the discrepancy subspace happens**. In particular, we consider a downstream task on classifying the brands of vehicles based on their images.
>         - The weak features, spanning a low-dimensional subspace $\mathcal{V}_w$, capture the designs of vehicles that tend to be more complicated (with a higher intrinsic dimension $d_w$) but contain irrelevant/spurious information that makes the model weak for the downstream task.
>         - The strong features, spanning a low-dimensional subspace $\mathcal{V}_s$, capture the logos of vehicles that are often simpler (with a lower intrinsic dimension $d_s$) and more relevant for the downstream task.
>
>         Since the design and logo of a vehicle are typically irrelevant, $\mathcal{V}\_w$ and $\mathcal{V}\_s$ are likely to be almost orthogonal in a high-dimensional feature space, leading to a small $d_{s \wedge w}$. Then, **the weak teacher $f_w$ brought by noisy SFT labels will make mistakes that only correlate to the design features in $\mathcal{V}_w$, independent of the logo features in $\mathcal{V}_s$. Such errors in weak supervision can be viewed as independent label noise with respect to the strong features in $\mathcal{V}_s$**. With an intrinsic dimension $d_s$, the generalization error of strong student induced by such independent label noise vanishes at a rate of $\frac{d_s}{N}$ (following the intuition of classical regression analysis).
>
>     We will further clarify these key intuitions in the revision.
>
> 2. **Empirical estimation of intrinsic and correlation dimensions**: Please see **R#1 (5dnE), Key#2**.
>
> 3. **Additional experiments on image regression and LLMs**: In the URL, we included additional real experiments on image regression (UTKFace) based on CLIP and ResNet embeddings, as well as NLP task (SST-2) based on finetuning LLMs. In particular, our new experiments provide strong empirical evidence for our theoretical results by showing that (i) **model discrepancies lead to better W2S**; and (ii) **variance reduction is a key advantage of W2S**. Please see **R#4 (W6Uk), Key#1** for details.
>
> ## Detailed responses
> * Discussion on concurrent works: We are keeping track of relevant concurrent works on W2S and will include discussions on them in the revision.
> * We appreciate all the detailed suggestions in "Other Comments Or Suggestions" and will revise accordingly in the next version.
>
> We are happy to answer any further questions you may have. If our responses above help address your concerns, we would truly appreciate a re-evaluation accordingly.

---

> > ### Comment · Reviewer_299q · 2025-04-05
> >
> > Thank you for the response. It successfully addresses my concerns. I believe that adding further discussion, particularly regarding key intuition on findings, would further strengthen the manuscript. Accordingly, I have increased my score to 3 (weak accept).

---

### Official Review · Reviewer_WLsf · 2025-03-12

**Overall Recommendation:** 2

**Summary:**

This paper theoretically investigates the weak-to-strong (W2S) generalization phenomenon in the setting of ridgeless regression. From a bias-variance decomposition perspective, the authors utilize the intrinsic dimensionality of fine-tuned models to analyze the generalization performance of the weak teacher model, the W2S student model, the strong model, and the strong ceiling model.
When variance dominates the generalization error, the paper finds that the W2S student model behaves similarly to the weak teacher model within their overlapping feature space, but reduces variance in the other parts of weak teacher's feature subspace. This reduction in variance creates an opportunity for the W2S student to outperform the weak teacher. Additionally, the authors characterize the performance gap recovery metric and the outperforming ratio metric based on their bias-variance decomposition analysis.
Furthermore, the theoretical findings are empirically verified in both the ridgeless regression setting and a binary image classification task.

## update after rebuttal

I would like to thank the authors for their further responses. However, I still do not see how the explicit effect of early stopping is reflected in the results. As I understand it, in the main context of the paper, the parameter $\alpha_{w2s} \to 0$ is assumed, and all main results are derived under this setting. It is unclear to me (quantitatively) how choosing a suitable (non-zero) value of $\alpha_{w2s}$ would impact the theoretical results presented in the main text. At this point, I intend to maintain my original evaluation.

**Claims And Evidence:**

The theoretical results seem to be well supported by the empirical findings. However, I have some concerns regarding certain parts of the proofs, see the Theoretical Claims section for details.

**Essential References Not Discussed:**

The relevant works seem to be properly cited.

**Experimental Designs Or Analyses:**

The ridgeless regression experiments align with the theoretical analysis and appear to be sound. However, using a binary image classification task with MSE loss may not substantially strengthen the paper in terms of supporting the claims in a more realistic setting. Would similar observations hold for multi-class image classification with a cross-entropy loss? Exploring this could provide stronger empirical support for the theoretical results.

**Methods And Evaluation Criteria:**

The proposed methods and evaluation criteria seem reasonable.

**Other Comments Or Suggestions:**

Typo in Line 66 (left column): "both student and teach" ---> "both student and teacher"

**Other Strengths And Weaknesses:**

*Other Strengths:*

1. The paper is well written and easy to follow.

2. It introduces the use of intrinsic dimensionality to quantitatively characterize model capacity, an aspect that has not been explored in previous works on W2S generalization.

*Other Weaknesses:*

1. This paper mainly focuses on a setting where both the weak and strong models perform well, that is, both the weak teacher and the W2S student have relatively high model capacity and achieve low approximation error. This may differ from many scenarios in previous W2S generalization studies, where the weak teacher typically has limited capacity and performs poorly.

2. The analysis is largely restricted to the variance-dominated regime, which limits its general applicability. However, the authors explicitly acknowledge this limitation in the paper.

3. Another limitation is that the paper assumes both the weak model features and the strong model features exist in the same feature space (i.e. $\mathbb{R}^d$), which may not always hold in practical scenarios.

**Questions For Authors:**

1. In addition to the question raised in Theoretical Claims, could you clarify how the obtained variance and bias terms in Line 603 correspond to the terms defined in Lines 121–123 (right column)? Can these terms be directly derived from $\mathrm{Var}(f)$ and $\mathrm{Bias}(f)$ as formulated in Lines 121–123?

2. If the weak teacher model is not sufficiently strong (e.g., if $\rho_w$ is no longer small), will W2S generalization still occur in your analysis?

3. Will your analysis remain valid if  $\phi_w(x)\in\mathbb{R}^{d_1}$ and $\phi_s(x)\in\mathbb{R}^{d_2}$ with $d_1\leq d_2$? This would correspond to a scenario where the strong model has more parameters than the weak model.

4. Does your analysis provide any insights into why early stopping often benefits W2S generalization?

5. Based on Proposition 3.5, it seems that large label noise (i.e., large  $\sigma^2$) can facilitate W2S generalization. Does this suggest that artificially injecting independent noise into the labels when training the weak teacher model could make W2S generalization more likely?

6. Could you provide more details on how the intrinsic dimension was estimated in your image classification experiments?

**Relation To Broader Scientific Literature:**

This paper contributes to the broader literature on learning theory, with a particular focus on the weak-to-strong generalization phenomenon, first identified by Burns et al. (2023).

**Theoretical Claims:**

I checked the proofs in the appendix and have some concerns that need to be addressed. Specifically, in the proof of Theorem 3.1 (or Theorem A.2), the authors use the following equality in Lines 595–597:
$$
\mathbb{E}\_{\mathbf{x}\sim\mathcal{D}}[\mathbb{E}\_{f\_{w2s}}[(f\_{w2s}(x)-f\_{\*}(x))^2]]=\mathbb{E}\_{\mathcal{S}\_x}[\mathbb{E}\_{\theta\_{w2s}}[\frac{1}{N}||\Phi\_s\theta\_{w2s}(x)-f\_{\*}||^2]].
$$

On the LHS, the expression represents the definition of excess risk, where $\mathbf{x}\sim\mathcal{D}$ is an independently drawn test sample, which is independent of the random variable $f\_{w2s}$. However, on the RHS, the expectation over the test sample $\mathbf{x}\sim\mathcal{D}$  is replaced by an average over the training sample $\mathcal{S}\_x$, which is **not independent** of $\theta_{w2s}$. This suggests that the RHS might be the training error rather than the excess risk. This issue seems to exist in other parts of the proofs in the appendix. Could you clarify why this equality holds and whether it correctly represents the excess risk rather than the training error?

---

> ### Author Rebuttal · Authors · 2025-03-30
>
> We appreciate the constructive suggestions from the reviewer. We are glad that they found our paper well-presented and our perspective novel. Since we could not include figures in OpenReview rebuttal, we will synopsize our new experiments in text and present the formal results in an anonymous URL: https://anonymous.4open.science/r/W2S_IntrinsicDim_demo-FEAF/demo.pdf.
>
> ## Key clarifications
> 1. **Additional experiments on image regression and LLMs**: In the URL, we included additional real experiments on image regression (UTKFace) based on CLIP and ResNet embeddings, as well as NLP task (SST-2) based on finetuning LLMs. Please see **R#4 (W6Uk), Key#1** for details.
>
> 2. **Empirical estimation of intrinsic and correlation dimensions**: Please see **R#1 (5dnE), Key#2**.
>
> 3. **Our analysis extends to weak-strong pairs with different feature dimensions**: Notice that since the intrinsic dimensions are far lower than the feature dimensions, the larger feature dimension (e.g., $D_s$) can always be reduced to the smaller one ($D_w < D_s$) via random projection (Johnson-Lindenstrauss transforms) with a negligible information loss $\epsilon \asymp d_s / D_w$. This is the main idea of the empirical estimation in Key#2. Since the high feature dimensions are not essential in our setting, we consider $D_s = D_w = d$ for a clean analysis. We will clarify this in the revision.
>
> 4. **Variance-dominated regime is crucial for understanding W2S, focusing on which does not compromise our contributions**: We respectfully disagree with the reviewer and emphasize that our focus on variance-dominated regime is a strength rather than a weakness. This setting is empirically motivated and fills an important gap in existing theoretical understanding. As our motivation in Lines 23-26 right said, empirical evidence in (Burns et al. 2023) suggests that larger W2S gain tends to occur on easier tasks (i.e., variance-dominated tasks). This can also be observed in our synthetic experiments (cf. Fig. 2&3) and our new vision and NLP experiments (see **R#4 (W6Uk), Key#1.2**). **Despite the appealing empirical performance of W2S in variance-dominated regimes, to the best of our knowledge, this regime has never been rigorously studied in prior theories on W2S. Our work fills this gap by providing a descriptive yet clean theoretical underpinning for W2S in the variance-dominated regime under finite samples**.
>
> ## Detailed responses
> * Concern in "Theoretical Claims": Notice that taking expectation over the training dataset is a standard way to connect the excess risks (ER) of random and fixed designs: $$\text{random design ER}=\mathbb{E}\_{S_x \sim \mathcal{D}^N}[\text{fixed design ER over }S_x]$$ Thanks to the reviewer's question, we realized that the inner expectation $\mathbb{E}\_{\theta_{w2s}}$ with shared randomness between $S_x$ and $\theta_{w2s}$ could be misleading (although it is technically correct as $\theta_{w2s}$ inside is conditioned on $S_x$). A better expression for the right-hand side may be $\mathbb{E}\_{S_x,\widetilde{S}}[||\Phi_s\theta - f_*||^2/N]$. We will revise these notations in the proofs.
>
> * "Questions For Authors" QFA#3,5,6 are addressed in the Key clarifications.
>
> * QFA#1: The variance-bias decomposition in Lines 595-604 is equivalent to directly deriving $Var(f)$ and $Bias(f)$ in Lines 121-123, both following the standard variance-bias decomposition for linear regression (see e.g. Liang, (2016), Statistical learning theory, Sec 2.9). The key observation here is that when opening the square in Line 600, the cross term vanishes because $\widetilde{z}$ is an independent random vector with zero mean.
>
> * QFA#2: First, as the analysis in Sec 3.2 shows, even if the weak teacher lacks capacity (FT approximation error $\rho_w$ is large), as long as the label noise $\sigma^2$ dominates $\rho_w$ (intuitively, harder tasks are more likely to have noisy labels), the problem still falls in the variance-dominated regime. Second, beyond the variance-dominated regime, our theory suggests a degeneration of W2S performance due to the vanishing advantage in variance reduction. This is empirically confirmed in both synthetic and real experiments (see discussions in Key#4).
>
> * QFA#4: Our analysis is not directly related to early stopping, but there are shared insights. Early stopping has a known intuitive connection with weight decay, which translates to ridge regression in our setting. As footnote 3 explained, ridge regression effectively brings low intrinsic dimensions by filtering out small eigenvalues. In the revision, we formally extend our analysis to ridge regression (see **R#1 (5dnE), Key#4** or the URL). We show that a suitable choice of ridge parameter (or early stopping intuitively) can bring better W2S performance by revealing the underlying low intrinsic dimensions.
>
> We are happy to answer any further questions you may have. If our responses above help address your concerns, we would truly appreciate a re-evaluation accordingly.

---

> > ### Comment · Reviewer_WLsf · 2025-04-07
> >
> > I would like to thank the reviewer for their detailed responses and apologize for my late engagement.
> >
> > Regarding my QFA#5 in the initial review, I was unable to find a clear corresponding response in the "Key Clarifications" section. Could the authors kindly point it out or restate their response to QFA#5?
> >
> > In Key#4, the authors suggest that easier tasks correspond to variance-dominated regimes, which I agree with. However, in the response to my QFA#2, the authors argue that harder tasks, those more likely to have noisy labels, also fall within the variance-dominated regime. I may have misunderstood something here, but this seems contradictory. Could the authors clarify this point?
> >
> > Regarding early stopping, this is precisely where I feel the problem setup in the paper differs from practical W2S generalization. Intuitively, a W2S model should not reach an optimal solution during training, as doing so increases the risk of overfitting to the weak teacher's outputs, i.e. mimicking the weak teacher. In fact, most W2S models in practice do not converge to an optimal solution, as shown in experiments (e.g., Burns et al., 2023). Therefore, defining the W2S model as the optimal solution in Eq. (3) may not capture the core mystery behind W2S generalization. Of course, my viewpoint can be subjective and I know that some other theoretical works on W2S adopt a similar setup. I'm open to further discussion with the authors and other reviewers on this.

---

> > > ### Author Response · Authors · 2025-04-07
> > >
> > > We appreciate the additional questions from the reviewer.
> > >
> > > - QFA#5: Yes, artificially injecting independent label noise to SFT data does improve W2S in both synthetic and real tasks. Such improvement in the synthetic regression can be observed by comparing Fig.2&3 in the submission, as discussed in Line 347-348, right. For real tasks, we point the reviewer to the discussion on **additional image regression and LLM experiments** in **R#4 (W6Uk), Key#1.2**:
> > >     - **Variance reduction is a key advantage of W2S**: While the variance-dominated regime that we focus on is overlooked by most prior theoretical works on W2S, our new experiments confirm that this is a particularly important regime where strong W2S performance (in terms of PGR/OPR) occurs in practice. By injecting artificial label noise to the UTKFace and SST-2 training data that leads to larger variance, we observe significant improvements in PGR and OPR, implying that variance reduction is a key advantage of W2S (see https://anonymous.4open.science/r/W2S_IntrinsicDim_demo-FEAF/demo.pdf).
> > >
> > > - Key#4 & QFA#2: First, we would like to clarify that our point in QFA#2 is not "harder tasks, those more likely to have noisy labels, also fall within the variance-dominated regime". Instead, what we emphasize is that **the key factor that characterizes the variance-dominated regime is the relative magnitude between the label noise and the FT approximation error**, instead of the absolute "hardness" quantified by the FT approximation error. For harder tasks (e.g. Olympic math problems), while the FT approximation error $\rho_w$ may be larger, the label noise $\sigma^2$ also tends to be larger (e.g. the human labels may be less accurate). As a result, a hard task can still fall in the variance-dominated regime, $\rho_w + \rho_s \ll \sigma^2$.
> > >
> > > - We totally agree that **regularization (e.g. early stopping, weight decay, min-$\ell_2$-norm solution) is crucial for W2S, which is reflected by our ridgeless regression + low intrinsic dimension analysis**.
> > >     - First, we kindly clarify a misconception on Eq(3): the **regularized** optimal ridgeless regression solution learned from data with low intrinsic dimensions in our setting (or the ridge regression solution of in our new analysis discussed in **R#1 (5dnE), Key#4**) is fundamentally different from the **unregularized** optimal solution learned from all weak pseudolabels. As highlighted in our response to QFA#4, **early stopping, ridgeless, and ridge regressions can all be viewed as regularization on the locality of parameter updates**. Therefore, our ridgeless/ridge regression analysis provides intuitive explanations on why regularization like early stopping is essential for W2S in practice.
> > >     - A minor difference of ridgeless regression from early stopping is that the regularization posed by the low intrinsic dimension of data is fixed. Such subtle difference disappears when we extend our analysis to **ridge regression** (see **R#1 (5dnE), Key#4**) where **the low intrinsic dimensions are implicitly enforced by the $\ell_2$-regularization**. Our (informal) ridge regression result conveys **the same message as ridgeless regression**: assume $\Sigma_s, \Sigma_w \succ 0$ are full rank with fast decay eigenvalues; let $\varrho_s, \varrho_w \ge 0$ quantify the FT approximation error in the ridge regression setting (see Remark 2 in the URL appendix for formal definition); then, with a suitable choice of ridge parameters $\alpha_w,\alpha_{w2s}$, we have $$\mathrm{ER}(f_{w2s}) \le 3 (\frac{\sigma^2}{4 n N}tr(\Sigma_s \Sigma_w)\varrho_s\varrho_w)^{1/3},$$ where $tr(\Sigma_s \Sigma_w)$ is the analogy of $d_{s \wedge w}$ in the ridgeless setting that quantifies the correlation between the weak and strong features. Intuitively, **the suitable choice of $\alpha_w,\alpha_{w2s}$ in ridge regression corresponds to the suitable stopping time in early stopping, which plays an important role in W2S**.
> > >     - In the less common scenario where finetuning goes beyond the kernel regime, we agree that early stopping may bring different and potentially more interesting feature learning dynamics. We will include a discussion on this in the future direction.
> > >
> > > We hope the above responses, along with our initial rebuttal, address all the reviewer's questions and concerns. If so, we would greatly appreciate a timely acknowledgment and your support of our work.

---

### Official Review · Reviewer_5dnE · 2025-03-14

**Overall Recommendation:** 2

**Summary:**

This paper presents a theoretical analysis of weak-to-strong (W2S) generalization, a recently observed phenomenon where a strong student model outperforms a weak teacher model when trained on the teacher's pseudo-labels. The authors provide a variance reduction perspective by leveraging the concept of intrinsic dimensionality by analyzing kernel models.  The analysis is technically sound, and the findings offer some insights into the conditions under which W2S generalization may occur.  The paper is well-written and organized, making it accessible to readers familiar with machine learning theory.

**Claims And Evidence:**

The authors make several key claims, including:

- W2S generalization can be explained through the lens of intrinsic dimension.
- The discrepancy between strong and weak models in W2S has a positive effect on reducing variance.
- W2S generalization occurs in variance, with the student model inheriting the teacher's variance in the overlapped feature subspaces and reducing it in the discrepancy subspace.
- The student-teacher correlation influences W2S, with lower correlation benefiting W2S.

The authors somewhat support these claims:

- **Theoretical Framework**: The authors develop a theoretical framework using ridgeless regression and the concept of intrinsic dimension. They provide mathematical formulations and theorems (like Theorem 3.1) to characterize the variance and bias in W2S generalization.  The framework is built on established observations on fine-tuning and the concept of intrinsic dimension.

- **Variance Reduction Analysis**: The authors provide a detailed analysis of how variance is reduced in the discrepancy subspace, supported by Theorem 3.1 and related discussions.  They use the analogy of car logo vs. design to provide intuition.

- **Role of Student-Teacher Correlation**: The authors define and incorporate student-teacher correlation (using correlation dimension) into their framework.  They explain how lower correlation (greater discrepancy) can lead to better W2S generalization.

- **Experimental Validation**: The authors present experiments on synthetic regression tasks and real image classification tasks.  These experiments aim to validate their theoretical findings.

**Essential References Not Discussed:**

It seems like most key contributions have been discussed.

**Experimental Designs Or Analyses:**

Please see above.

**Methods And Evaluation Criteria:**

**Theoretical Framework**

- The authors propose a theoretical framework based on ridgeless regression and the concept of intrinsic dimension.
- They use mathematical tools and theorems to analyze the variance and bias components of W2S generalization.
- This theoretical approach is appropriate for gaining a deeper understanding of the underlying mechanisms driving W2S. Ridgeless regression, while simplified, allows for tractable analysis and can provide valuable insights.


**Experimental Validation**

- The authors use both synthetic regression tasks and real image classification tasks for experimental validation.

- Synthetic data allows them to control specific parameters and test the theoretical predictions in a controlled setting.

- Real image classification tasks (using CIFAR-10) demonstrate the relevance of their findings to practical applications.

---
**Suggestion**: While the current methods and evaluation criteria are generally sound, the experimental validation could be expanded. For example, exploring a wider range of datasets, model architectures, and training paradigms would provide a more comprehensive evaluation of the theory. Additional experiments that directly measure or manipulate the intrinsic dimension could further strengthen the connection between the theory and the empirical results.

**Other Comments Or Suggestions:**

The paper has merit in its theoretical analysis and the use of intrinsic dimension to explain W2S generalization. However, the strong assumptions, limited novelty, and the need for more extensive experimental validation and discussion of practical implications suggest that the paper is not yet ready for publication. I encourage the authors to address these concerns and resubmit the work in the future.

**Other Strengths And Weaknesses:**

**Weaknesses**

- **Idealized Assumptions**: The theoretical analysis relies on strong assumptions, such as the Gaussian feature assumption. While the authors mention that the results hold for sub-Gaussian features, the analysis in the main text is limited to the Gaussian case. It is unclear how sensitive the results are to these assumptions and how well they generalize to more realistic scenarios.

- **Practical Implications**: The theoretical results provide a good understanding of W2S, but their practical implications are not fully explored. The paper could benefit from a more detailed discussion on how these findings can be used to improve W2S training or model design in real-world applications.

- **Experimental Validation**: While the experiments support the theory, they are somewhat limited. Additional experiments on more diverse datasets and model architectures such as language models would strengthen the empirical validation of the proposed framework.

**Strengths**

- **Clearly written**: The paper is well-motivated, clearly written, and easy to follow.

- **New perspective on W2S**: The theoretical framework based on intrinsic dimension provides a novel perspective on W2S generalization. The analysis of variance reduction and the role of student-teacher discrepancy is insightful.

**Questions For Authors:**

- **Gaussian Feature Assumption**: The theoretical analysis relies heavily on the assumption of Gaussian features. While you mention that the results can be extended to sub-Gaussian features, could you provide more detail on how this extension is achieved and what the key differences in the results would be? Are there specific types of sub-Gaussian distributions where the results would be significantly weaker or not applicable.

- **Alternative distributions**: How does the Gaussian assumption influence the specific forms of the generalization error bounds derived in Theorem 3.1? Are there alternative distributional assumptions that would allow for a similar analysis, and what would be the trade-offs?

- **Estimate intrinsic dimension**: The concept of intrinsic dimension is central to your analysis. How do you propose to estimate or approximate the intrinsic dimension of a model in practice, especially for deep learning models where it's not straightforward? Are there practical methods or heuristics that can be used to guide the choice of student and teacher models based on their estimated intrinsic dimensions?

- **Potential downsides**: Your analysis suggests that a discrepancy in intrinsic dimensions between the student and teacher is beneficial for W2S. Are there any potential downsides to having a very large discrepancy? Could there be a point where the discrepancy becomes too large, and W2S is negatively affected

- **Correlation Mesaure**: The correlation dimension is used to quantify the similarity between the student and teacher models. How sensitive are your results to the specific way this correlation is measured? Are there other measures of student-teacher similarity that could be used, and would they lead to qualitatively different results?

- **Ridge case**: The paper focuses on ridgeless regression. How do you anticipate the results might change with the introduction of regularization, which is commonly used in practice? What are the key challenges in extending your analysis to the regularized setting?

- **Practical Implications**: What are the most important practical implications of your findings? How can practitioners use your theoretical insights to improve W2S training or design more effective student and teacher models?

**Relation To Broader Scientific Literature:**

That aspect seems fine.

**Theoretical Claims:**

I did not check the correctness of the theoretical proofs.

---

> ### Author Rebuttal · Authors · 2025-03-30
>
> We thank the reviewer for their time and suggestions. We are glad that they found this work well-presented and provided a good understanding of W2S. Since we could not include figures in the OpenReview rebuttal, we will synopsize our new experiments in text and present the formal results in an anonymous URL: https://anonymous.4open.science/r/W2S_IntrinsicDim_demo-FEAF/demo.pdf.
>
> ## Key clarifications
> 1. **Additional experiments on image regression and LLMs**: We appreciate the suggestion on strengthening the experimental validation. In the URL, we included additional real experiments on image regression (UTKFace) based on CLIP and ResNet embeddings, as well as NLP task (SST-2) based on finetuning LLMs. Please see **R#4 (W6Uk), Key#1** for detailed discussions.
>
> 2. **Empirical estimation of intrinsic and correlation dimensions**: We first highlight that with small finetunable parameter counts (e.g., linear probing or finetuning last few layers), we estimate intrinsic dimensions based on traces of data covariances (see Line 413-419, i.e., the minimum rank for the best low-rank approximation of $\Sigma_w, \Sigma_s$ with a relative error in trace less than $\tau=0.01$).
>     1. For correlation dimension, a practical challenge is that **the feature dimensions of the weak and strong model can be different**--$V_s, V_w$ of sizes $D_s \times d_s$ and $D_w \times d_w$ can have $D_s \ne D_w$. In this case, we gauge the correlation dimension by matching $V_s, V_w$ through a random unitary matrix $G \in \mathbb{R}^{D_s \times D_w}$ s.t. $d_{s \wedge w} = ||V_s^\top G V_w||^2_F$. This provides a good estimation for $d_{s \wedge w}$ because with low intrinsic dimensions $\max(d_s, d_w) \ll D_s, D_w$ in practice, mild dimension reduction through $G$ well preserves the essential information in $V_s, V_w$.
>     2. We appreciate the question on **empirical estimations of intrinsic and correlation dimensions for finetuning large models**. When finetuning LLMs, $D_s, D_w$ will be on the order of millions, making the covariance-based estimation infeasible. In this case, we use the SAID method proposed by Aghajanyan et al. (2020) to estimate intrinsic dimensions. Following Remark 2.5, for full FT, we take $\Phi_s, \Phi_w$ as the gradients of the strong and weak models at pretrained initializations. We use randomized rangefinder based on sparse JLTs and the random unitary projection trick above to estimate $d_{s \wedge w}$ efficiently (see Appendix C in the above URL for detailed procedures).
>
> 3. **Our assumptions are reasonable and generalizable**. We kindly highlight that footnote 4 and Theorem 3.1 provide explicit pointers to Theorem A.1, the formal version of Theorem 3.1 that rigorously extends the results to sub-Gaussian features. As explained in footnote 4, both theorems convey the same message. Due to page limit, we present the Gaussian results in the main text for clarity. Meanwhile, in the introduction, we provide strong motivations for studying FT in the kernel regime. The choice of subgaussian features is also well-justified by literature (Wei et al., 2022) and related works (Wu & Sahai, 2024; Ildiz et al., 2024) on W2S. Most importantly, the empirical evidence in our new experiments demonstrates the generalizability of our assumptions and results to real-world scenarios.
>
> 4. **Ridge regression analysis**: Our analysis can be extended to the ridge regression setting. As mentioned in footnote 1, when $\Sigma_s, \Sigma_w$ admit full ranks with fast decaying eigenvalues, ridge regression effectively brings low intrinsic dimensions by filtering out the small eigenvalues. The result for ridge regression again conveys the same message as Theorem 3.1, with the correlation dimension replaced by $tr(\Sigma_s \Sigma_w)$. Intuitively, a large discrepancy corresponds to a $\Sigma_s, \Sigma_w$ pair with approximately orthogonal leading eigenvectors associated with the large eigenvalues, bringing a small $tr(\Sigma_s \Sigma_w)$. (See Appendix A in the above URL for detailed statements and proofs.)
>
> ## Detailed responses
> * The **practical implications** of our theoretical insights on the choice of weak v.s. strong models and sample sizes in W2S are self-evident and discussed in detail in Sections 3.2 and 5. We will try to further emphasize them in the revision.
>
> * Assuming both weak and strong models have sufficient capacities to achieve **low FT approximation errors on the downstream task**, our theory and experiments show that the larger discrepancy between weak and strong models brings better W2S, with **no potential downsides**.
>
> We are happy to answer any further questions you may have. If our responses above have addressed your concerns, we would truly appreciate a re-evaluation accordingly.

---

### Decision · Program_Chairs · 2025-05-01

**Decision:**

Accept (poster)

**Comment:**

This paper provides a theoretical study of the weak-to-strong (W2S) generalization phenomenon highlighting the role of the intrinsic dimension. The theoretical analysis is carried out in the setting of ridgeless regression (as common in related literature), and the main result is a decomposition of the variance into three terms: one depending on the variance in the intersection of the two subspaces and on the number of training samples for the teacher, one depending on the dimension of the strong subspace and on the number of training samples for the student, and the last one depending on the variance in the difference between the two subspaces and on the number of training samples for the teacher. This implies that W2S occurs when the student has low intrinsic dimension, or student-teacher correlation is low -- something in agreement with existing experimental evidence.

The paper is currently borderline: I have disregarded Review 5dnE as LLM-generated, and the remaining three reviews express borderline evaluations even after the rebuttal of the authors and the discussion between AC and reviewers. On the positive side, the reviewers have praised the insights coming the theoretical analysis and, specifically, the use of intrinsic dimensionality to quantitatively characterize model capacity. On the negative side, criticisms about the simplified setting and practical relevance have been raised. The authors have provided additional experiments that have partially addressed such criticisms. One remaining issue concerns the restriction to the ridgeless regression setting. To address that, the authors have shared an additional appendix via an anonymous link. I do think that carefully proof-reading that goes well beyond what is possible in the ICML reviewing timeline. At the same time, after my own reading of the manuscript, my opinion is that (1) the extension to ridge regression is interesting but the paper is already strong enough to meet the bar for acceptance, and (2) the extension to ridge regression is already mentioned as a footnote in the original version of the paper and, based on my knowledge of the area, I find it likely that it can be carried out using similar techniques. For these reasons, I am leaning towards accepting the paper.

I do warmly encourage the authors to provide a careful revision to the camera ready incorporating the non-trivial amount of material that has been discussed during the rebuttal process.